# Minimax Optimal Two-Stage Algorithm For Moment Estimation Under Covariate Shift

**Zhen Zhang, Xin Liu,**[*] **Shaoli Wang, Jiaye Teng**[*]
School of Statistics and Data Science
Shanghai University of Finance and Economics
Shanghai 200433, P.R. China
`zhenzhang@stu.sufe.edu.cn, swang@shufe.edu.cn`

## Abstract

Covariate shift occurs when the distribution of input features differs between the training and testing phases. In covariate shift, estimating an unknown function's moment is a classical problem that remains under-explored, despite its common occurrence in real-world scenarios. In this paper, we investigate the minimax lower bound of the problem when the source and target distributions are known. To achieve the minimax optimal bound (up to a logarithmic factor), we propose a two-stage algorithm. Specifically, it first trains an optimal estimator for the function under the source distribution, and then uses a likelihood ratio reweighting procedure to calibrate the moment estimator. In practice, the source and target distributions are typically unknown, and estimating the likelihood ratio may be unstable. To solve this problem, we propose a truncated version of the estimator that ensures double robustness and provide the corresponding upper bound. Extensive numerical studies on synthetic examples confirm our theoretical findings and further illustrate the effectiveness of our proposed method.

## 1 Introduction

*Covariate shift* occurs when the marginal distribution of the input covariates differs between the training and test data, but the conditional distribution of the response given the covariates remains consistent. It is prevalent and crucial in real-world applications, such as natural language processing (Wang et al., 2017), computer vision (Tzeng et al., 2017), reinforcement learning (Chang et al., 2021), finance (Timper, 2021) and medical care (Guan & Liu, 2021). In addition to its practical significance, covariate shift has been extensively studied theoretically, particularly in the context of function estimation. For example, the linear function in linear regression (Ryan & Culp, 2015), the classifier in nonparametric classification (Kpotufe & Martinet, 2021), and the conditional mean function in nonparametric regression (Feng et al., 2024b).

However, in modern machine learning tasks, we are often more interested in the functionals of these estimated functions, such as the average treatment effect in causal inference or accuracy in classification problems. To address these issues, practitioners commonly use a two-stage approach: first, they estimate the functions, and then they use these estimators to calculate the desired functional (Oates et al., 2017; Lei & Candès, 2021; Blanchet et al., 2024). Yet, there are few theoretical works that analyze the efficiency of these two stages from a unified perspective. Consequently, this paper aims to explore the following question:

*Under covariate shift, can we calibrate an optimal function estimator under source distribution to maintain the optimality for the functional under target distribution, and if so, how?*

To this end, we consider the problem of estimating the moment of an unknown function under covariate shift. This is a common scenario in many fields, such as counterfactual inference in causal inference[1]. Specifically, we aim to estimate the $q$-th moment $I_f^q := \mathbb{E}_{\boldsymbol{X} \sim \mathbb{P}^*}[f^q(\boldsymbol{X})]$ of $f$ under target distribution $\mathbb{P}^*$ with p.d.f. $p^*(\boldsymbol{x})$, based on values $f(\boldsymbol{x}_1), \ldots, f(\boldsymbol{x}_n)$ observed on $n$ random samples

---

[*]Correspond to {`liu.xin,tengjiaye`}`@mail.shufe.edu.cn`.
[1]For a more detailed discussion, please refer to Appendix B.1.

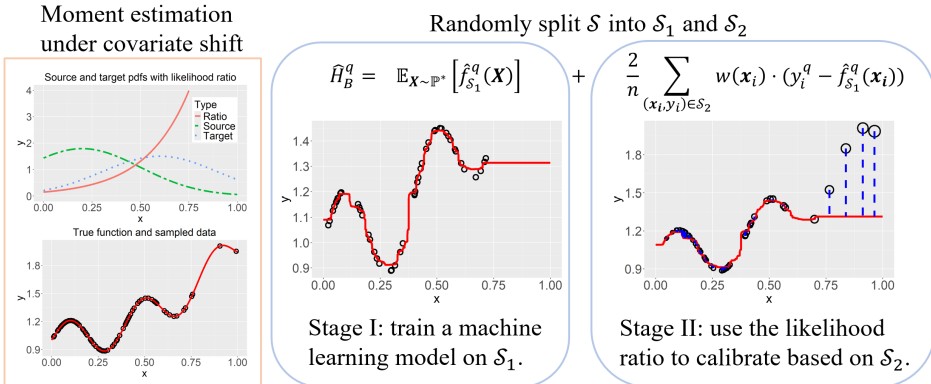

Figure 1: The top figure in the left box shows the source and target distributions along with the corresponding likelihood ratio. The bottom figure in the left box illustrates the underlying function and the sampled data. The middle and right boxes show the first and second stages, respectively. More details can be found in Section 4.2.

$\boldsymbol{x}_1, \ldots, \boldsymbol{x}_n \in \Omega$ drawn from source distribution $\mathbb{P}^\circ$ with p.d.f. $p^\circ(\boldsymbol{x})$. Here, the unknown function $f$ belongs to the Sobolev space $\mathcal{W}^{s,p}(\Omega)$ with $\Omega \subset \mathbb{R}^d$, where $s$ indicates the degree of smoothness and $p$ specifies the integrability condition of these derivatives. To understand the complexity of this nonparametric functional estimation problem under covariate shift, we restrict our problem in certain class defined as $\mathcal{V} := \{(f, p^\circ, p^*) : f \in \mathcal{W}^{s,p}(\Omega), \ \underline{b} \le p^\circ \le \bar{b}, \ \sup_{\boldsymbol{x} \in \Omega} w(\boldsymbol{x}) \le B\}$, where $w(\boldsymbol{x}) := \frac{p^*(\boldsymbol{x})}{p^\circ(\boldsymbol{x})}$ represents the likelihood ratio. The constants $\underline{b}, \bar{b}$ and $B$ are used to quantitatively compare the quality of different source distribution and the intensity of covariate shift. The problem we consider extends the work of Blanchet et al. (2024) to scenarios involving covariate shift.

Our contributions in this paper can be summarized as follows:

Contribution I: **The impact of covariate shift on the minimax lower bound when $p^\circ$ and $p^*$ are known**. We establish the minimax lower bound for any given target p.d.f. $p^*$ over $\mathcal{V}$ as follow[2]:

$$\inf_{\hat{H}^q \in \mathcal{H}} \sup_{(f, p^\circ) \in \mathcal{V}_{p^*}} \mathbb{E} \left| \hat{H}^q - I_f^q \right| \gtrsim B \cdot \bar{b} \cdot n^{\max\{-q(\frac{s}{d} - \frac{1}{p}) - 1, -\frac{1}{2} - \frac{s}{d}\}},$$

where $\mathcal{H} := \{\hat{H}^q : \Omega^n \times \mathbb{R}^n \to \mathbb{R}\}$ denotes the class that contains all estimators of the $q$-th moment $I_f^q$. The value $B \ge \sup_{\boldsymbol{x} \in \Omega} w(\boldsymbol{x})$ measures the cost we need when doing estimations under covariate shift. A larger $B$ indicates a greater extent of covariate shift, which leads to a more challenging estimation task. When there is no covariate shift, $B = 1$. The value $\bar{b}$ indicates the influence of the sampling strategy to the problem, a larger $\bar{b}$ implies a more irregular source distribution. When $\mathbb{P}^\circ$ is taken as uniform distribution, $\bar{b} = 1$. The term $n^{\max\{-q(\frac{s}{d} - \frac{1}{p}) - 1, -\frac{1}{2} - \frac{s}{d}\}}$ describes the difficulty caused by the combination of function class $\mathcal{W}^{s,p}(\Omega)$ and the order of the moment $q$.

Contribution II: **A two-stage algorithm that achieves the lower bound**. We then propose a two-stage algorithm which attains the minimax lower bound up to a logarithmic factor. Specifically, we split the data $\mathcal{S} = \{(\boldsymbol{x}_i, y_i = f(\boldsymbol{x}_i))\}_{i=1}^n$ into two parts with equal size, $\mathcal{S}_1$ and $\mathcal{S}_2$. We use $\mathcal{S}_1$ to fit an optimal nonparametric estimator $\hat{f}_{\mathcal{S}_1}$ of $f$, and then use $\mathcal{S}_2$ to do calibration. Our estimator is formulated as follows, and the framework is illustrated in Figure 1:

$$\hat{H}_B^q(\mathcal{S}) := \mathbb{E}_{\boldsymbol{X} \sim \mathbb{P}^*}[\hat{f}_{\mathcal{S}_1}^q(\boldsymbol{X})] + \sum_{(\boldsymbol{x}_i, y_i) \in \mathcal{S}_2} w(\boldsymbol{x}_i) \cdot (y_i^q - \hat{f}_{\mathcal{S}_1}^q(\boldsymbol{x}_i)). \tag{1}$$

Since $p^*(\boldsymbol{x})$ is known, we use a sampling method to estimate the first term. This estimator can be seen as importance sampling using $\hat{f}_{\mathcal{S}_1}$ as the control variate. The $\hat{f}_{\mathcal{S}_1}$ introduced in the first stage is used to reduce variance, while $w(\boldsymbol{x})$ in the second stage is used to debias residual. Furthermore, if $\hat{f}_{\mathcal{S}_1}$ is chosen to be optimal (Assumption 1) then it matches the minimax lower bound up to a

---

[2]We use the notations $\gtrsim$ and $\lesssim$ to hide log factors and define $\mathcal{V}_{p^*} := \{(f, p^\circ) : (f, p^\circ, p^*) \in \mathcal{V}\}$.

logarithmic factor, as follows:

$$\mathbb{E}_{\mathcal{S}}\left[\left|\hat{H}_B^q(\mathcal{S}) - I_f^q\right|\right] \lesssim B \cdot \bar{b} \cdot n^{\max\{-q(\frac{s}{d}-\frac{1}{p})-1, -\frac{1}{2}-\frac{s}{d}\}}.$$

Contribution III: **Stabilized algorithm with double robustness when $p^\circ$ and $p^*$ are unknown**. In practice, the source and target distributions are typically unknown. Therefore, we need to replace $w(\boldsymbol{x})$ in equation 1 with the estimator $\hat{w}(\boldsymbol{x})$. To estimate $w(\boldsymbol{x})$, it is necessary to have $m$ ($\gg n$)[3] unlabelled data points drawn from $\mathbb{P}^*$, denoted as $\mathcal{S}' = \{x_1', \ldots, x_m'\}$. Although there is extensive literature (Yu & Szepesvári, 2012; Bickel et al., 2007; Kpotufe, 2017) on methods for estimating $w(\boldsymbol{x})$, $\hat{w}(\boldsymbol{x})$ becomes unstable when we lack prior information about $\mathbb{P}^*$ and $\mathbb{P}^\circ$. To address this, we introduce the truncated version of equation 1 as follows:

$$\tilde{H}_T^q(\mathcal{S} \cup \mathcal{S}') := \frac{1}{m}\sum_{i=1}^{m}\hat{f}_{\mathcal{S}_1}^q(x_i') + \sum_{(\boldsymbol{x}_i, y_i) \in \mathcal{S}_2}\tau_T(\hat{w}(\boldsymbol{x}_i)) \cdot (y_i^q - \hat{f}_{\mathcal{S}_1}^q(\boldsymbol{x}_i)),$$

where $\tau_T(\cdot) := \min\{\cdot, T\}$. To understand how this affects the convergence rate, we assume that $g(T) := \mathbb{P}\left(\left\{x : \frac{p^*(\boldsymbol{x})}{p^\circ(\boldsymbol{x})} > T\right\}\right) \leq T^{-\alpha}$ and denote $r(n) = n^{\max\{-q(\frac{s}{d}-\frac{1}{p})-1, -\frac{1}{2}-\frac{s}{d}\}}$. By selecting $T$ to balance the trade-off between bias and variance, the convergence rate of the truncated version of the estimator is:

$$\mathbb{E}_{\mathcal{S}}\left[\left|\tilde{H}_T^q(\mathcal{S} \cup \mathcal{S}') - I_f^q\right|\right] \lesssim \bar{b} \cdot r(n)^{\frac{\alpha}{\alpha+1}} + m^{-\frac{1}{2}}.$$

Note that our estimator involves two models: one for estimating $f$ and the other for estimating $w(\boldsymbol{x})$. We demonstrate that our estimator possesses the double robust property, meaning it be consistent if at least one of the two estimates is consistent.

The paper is organized as follows. Section 2 provides a review of the relevant literature. Section 3 introduces the notation and problem setup. In Section 4, we establish the minimax rate of convergence for the moment estimation problem under covariate shift and demonstrate its attainability where $p^\circ$ and $p^*$ are known. In Section 5, we introduce a stabilized algorithm with double robustness for cases where $p^\circ$ and $p^*$ are unknown. In Section 6, we provide the simulation results to illustrate our theoretical findings. Finally, we conclude in Section 7 and leave most of proofs to the appendix.

## 2 RELATED WORK

**Covariate shift.** Extensive work has been done on covariate shift (Sugiyama & Kawanabe, 2012). The primary tool to address covariate shift is the reweighting method, as discussed by Shimodaira (2000). This method utilizes the likelihood ratio to reweight the loss function or estimator, thereby correcting for bias. Numerous studies have examined the statistical efficiency of this method in various contexts, such as nonparametric classification (Kpotufe & Martinet, 2021) and nonparametric regression (Ma et al., 2023; Feng et al., 2024a). Among them, Ma et al. (2023) is closely related to ours which studies the covariate shift problem in the context of nonparametric regression over a reproducing kernel Hilbert space and introduces the truncation trick on the reweighted loss for the case of unbounded likelihood ratios. However, while their focus is on estimating the unknown function itself, our work is concerned with the moments of the unknown function, thereby advancing one step beyond previous studies. The likelihood ratio is the key component of this method, but it is often unknown in practice. Many studies focus on estimating the likelihood ratio in covariate shift using techniques such as histogram-based methods (Kpotufe, 2017), kernel mean matching (Yu & Szepesvári, 2012; Li et al., 2020), and discriminative learning (Bickel et al., 2007; 2009).

There are also other methods that do not require estimating the likelihood ratio. For example, transformation-based methods (Flamary et al., 2016; Wang et al., 2024) involve learning a transformation from the source feature domain to the target feature domain and then using the transformed data to train the model. Domain-invariant methods (Robey et al., 2021; Blanchard et al., 2021) aim to learn transformations that map data between domains and subsequently enforce invariance to these transformations.

---

[3]The condition $m$ ($\gg n$) is not necessary in this paper but it is a common scenario in practice, as unlabeled data is generally more easily available than labeled data.

**Two-stage algorithm** is a class of method that divides the problem-solving process into two distinct phases, each with a specific purpose. These methods have numerous applications across various fields. For example, in the deep learning community, the training regime typically involves representation learning followed by downstream task fine-tuning (Devlin et al., 2018; Du et al., 2020; He et al., 2022). In computational statistics, regression-adjusted control variates are introduced in the first stage to reduce the variance for the estimation of the quantity of interest (Oates et al., 2017; Holzmüller & Bach, 2023). In uncertainty quantification, conformal inference uses a calibration set in the second stage to construct confidence intervals for any point estimator from the first stage (Romano et al., 2019; Tibshirani et al., 2019; Teng et al., 2021; Lei & Candès, 2021).

Recently, Blanchet et al. (2024) found that using optimal nonparametric machine learning algorithms to construct control variates is the best way to enhance Monte Carlo methods, but it remains underexplored how to adapt such a two-stage algorithm to ensure it remains optimal under covariate shift.

## 3   NOTATION AND SETUP

**Notation.** Let $\|\cdot\|$ denote the standard Euclidian norm, and let $\Omega = [0,1]^d$ denote the unit cube in $\mathbb{R}^d$ for any fixed $d \in \mathbb{N}$. Define $\mathbb{N}_0 := \mathbb{N} \cup \{0\}$ as the set of all non-negative integers. For any given $s \in \mathbb{N}_0$ and $1 \leq p \leq \infty$, the Sobolev space $\mathcal{W}^{s,p}(\Omega)$ is defined as follows:
$$\mathcal{W}^{s,p}(\Omega) := \{f \in \mathcal{L}^p(\Omega) : D^\alpha f \in \mathcal{L}^p(\Omega), \forall \alpha \in \mathbb{N}_0 \text{ satisfying } |\alpha| \leq s\}.$$

**Problem Setup.** Consider the problem of estimating a function's $q$-th moment under covariate shift. For any fixed $q \in \mathbb{N}$ and $f \in \mathcal{W}^{s,p}(\Omega)$, we want to estimate the $q$-th moment $I_f^q := \mathbb{E}_{\boldsymbol{X} \sim \mathbb{P}^*}[f^q(\boldsymbol{X})]$ under target distribution $\mathbb{P}^*$ where $p^*$ is the p.d.f. of $\mathbb{P}^*$. However, we only access to $n$ observations $\mathcal{S} = \{(\boldsymbol{x}_i, y_i)\}_{i=1}^n \subset \Omega^n \times \mathbb{R}^n$, where $\{\boldsymbol{x}_i\}_{i=1}^n$ are i.i.d. samples from source distribution $\mathbb{P}^\circ$ with p.d.f. $p^\circ$ and $y_i = f(\boldsymbol{x}_i)$. Our goal is to find an estimator $\hat{H}^q : \Omega^n \times \mathbb{R}^n \to \mathbb{R}$ to tackle with this nonparametric functional estimation problem under covariate shift.

Notably, under covariate shift, transferring knowledge from the source to the target is not straightforward without additional assumptions on $\mathbb{P}^\circ$ and $\mathbb{P}^*$. For example, consider the extreme case where $\mathbb{P}^\circ$ is a uniform distribution on $[0, \frac{1}{2}]^d$ and $\mathbb{P}^*$ is a uniform distribution on $[\frac{1}{2}, 1]^d$. In this scenario, anything learned on $[0, \frac{1}{2}]^d$ is independent of the information on $[\frac{1}{2}, 1]^d$. Therefore, the difficulty of this problem hinges on the notion of discrepancy between the source and target distributions. By imposing various conditions on this discrepancy, we can define different families of source-target pairs (Definition 1), denoted as $(\mathbb{P}^\circ, \mathbb{P}^*)$. In this paper, we focus on pairs defined on the likelihood ratio, a common choice under covariate shift (Ma et al., 2023; Feng et al., 2024b), as outlined below.

**Definition 1** (Uniformly $B$-bounded). *For any source-target pairs $(\mathbb{P}^\circ, \mathbb{P}^*)$ with p.d.f. $(p^\circ, p^*)$ and constant $B \geq 1$, we say that the pair is uniformly $B$-bounded if:*
$$\sup_{\boldsymbol{x} \in \Omega} w(\boldsymbol{x}) := \frac{p^*(\boldsymbol{x})}{p^\circ(\boldsymbol{x})} \leq B.$$

It is worth noting that $B = 1$ recovers the case without covariate shift, *i.e.*, $\mathbb{P}^\circ = \mathbb{P}^*$. At the same time, the source distribution also affects the difficulty of the problem. A well-shaped source distribution provides comprehensive information about the function $f$. To exclude extreme cases, we assume that $p^\circ$ is bounded below by $\underline{b}$ and above by $\bar{b}$, an assumption commonly used in transfer learning (Audibert & Tsybakov, 2007; Cai & Wei, 2021).

Putting the above together, we consider the following class in the rest of this paper:
$$\mathcal{V}(\underline{b}, \bar{b}, B, s, p) := \left\{(f, p^\circ, p^*) : f \in \mathcal{W}^{s,p}(\Omega), \ \underline{b} \leq p^\circ \leq \bar{b}, \ \sup_{\boldsymbol{x} \in \Omega} w(\boldsymbol{x}) \leq B\right\},$$
and investigate how the parameters within this family influence the difficulty of the estimation problem. We use the shorthand $\mathcal{V}$ when the context is clear.

## 4   MINIMAX LOWER BOUND AND ITS ATTAINABILITY

In this section, we establish the minimax rate of convergence for the moment estimation problem under covariate shift in Section 4.1, assuming that $p^\circ$ and $p^*$ are known. We then propose a two-

stage algorithm that achieves this bound up to a logarithmic factor in Section 4.2. Additionally, since a large covariate shift intensity $B$ leads to high variance, we introduce a biased estimator in Section 4.3 to achieve a variance-bias trade-off. This allows us to obtain a bound that is independent of $B$.

## 4.1 MINIMAX LOWER BOUND

In this subsection, we study the minimax lower bound for the problem above via the method of two fuzzy hypotheses (Tsybakov, 2009). We have the following minimax lower bound on the class $\mathcal{H}_n^{f,q}$ that contains all estimators $\hat{H}^q : \Omega^n \times \mathbb{R}^n \to \mathbb{R}$ of the $q$-th moment $I_f^q$.

**Theorem 1** (Minimax Lower Bound on Estimating the Moment under Covariate Shift). *Let $\mathcal{H}_n^{f,q}$ denote the class of all the estimators using $\mathcal{S} = \{(\boldsymbol{x}_i, y_i = f(\boldsymbol{x}_i))\}_{i=1}^n$ to estimate the $q$-th moment of $f$ under $\mathbb{P}^*$. For any given target p.d.f. $p^* \in \{p^* : (\cdot, \cdot, p^*) \in \mathcal{V}\}$, when $p \geq 2$, $q \leq p \leq 2q$ and $(f, p^\circ) \in \mathcal{V}_{p^*} := \{(f, p^\circ) : (f, p^\circ, p^*) \in \mathcal{V}\}$, it holds that*

$$\inf_{\hat{H}^q \in \mathcal{H}_n^{f,q}} \sup_{(f,p^\circ) \in \mathcal{V}_{p^*}} \mathbb{E}_{\mathcal{S}} \left[ \left| \hat{H}^q(\mathcal{S}) - I_f^q \right| \right] \gtrsim B \cdot \bar{b} \cdot n^{\max\{-q(\frac{s}{d} - \frac{1}{p}) - 1, -\frac{1}{2} - \frac{s}{d}\}}. \tag{2}$$

Theorem 1 indicates that the complexity of this problem is determined by three factors: covariate shift, sampling strategy, and the interplay between the function class and the moment order. We will provide a simulation study to illustrate these three factors in Section 6.

**Covariate Shift**: The value $B \geq \sup_{\boldsymbol{x} \in \Omega} w(\boldsymbol{x})$ represents the additional cost associated with making accurate estimations in the presence of covariate shift. A larger $B$ signifies a more pronounced covariate shift, which inherently complicates the estimation process due to the increased divergence between the source and target distributions. When there is no covariate shift, $B = 1$, meaning the distributions are aligned, and estimation is straightforward.

**Sampling Strategy**: The parameter $\bar{b}$ quantifies the impact of the sampling strategy on the problem. A larger $\bar{b}$ suggests a more irregular source distribution, which introduces complexities in the estimation process. For example, when $\mathbb{P}^\circ$ is a uniform distribution, $\bar{b} = 1$, indicating that the source distribution is regular and well-behaved.

**Function Class and Moment Order**: The term $n^{\max\{-q(\frac{s}{d} - \frac{1}{p}) - 1, -\frac{1}{2} - \frac{s}{d}\}}$ encapsulates the complexity arising from the interplay between the function class $\mathcal{W}^{s,p}(\Omega)$ and the order of the moment $q$. It delineates the function class into two distinct regimes based on error decomposition: $(f^q - \hat{f}^q)$ can be broken down into the semi-parametric influence part $f^{2q-2}(f - \hat{f})^2$ and the propagated estimation error $(f - \hat{f})^{2q}$ (Blanchet et al., 2024). When $s > \frac{d(2q-p)}{p(2q-2)}$, the smoothness parameter $s$ is sufficiently large to make the semi-parametric influence part the dominant term. Conversely, $s < \frac{d(2q-p)}{p(2q-2)}$, the semi-parametric influence no longer dominates, and the final convergence rate shifts from $n^{-\frac{1}{2} - \frac{s}{d}}$ to $n^{-q(\frac{s}{d} - \frac{1}{p}) - 1}$.

## 4.2 ATTAINABILITY

This subsection focuses on constructing minimax optimal estimators for the $q$-th moment under covariate shift. It addresses the question posed in Section 1: under covariate shift, if we can obtain an optimal function estimator $\hat{f}_{\mathcal{S}_1}$ that satisfies Assumption 1, then we can calibrate it as follows to derive the moment estimator $\hat{H}_B^q$, which preserves the optimality in functional estimation:

$$\hat{H}_B^q(\mathcal{S}) := \mathbb{E}_{\boldsymbol{X} \sim \mathbb{P}^*}[\hat{f}_{\mathcal{S}_1}^q(\boldsymbol{X})] + \frac{2}{n} \sum_{(\boldsymbol{x}_i, y_i) \in \mathcal{S}_2} w(\boldsymbol{x}_i) \cdot (y_i^q - \hat{f}_{\mathcal{S}_1}^q(\boldsymbol{x}_i)), \tag{3}$$

where $\mathcal{S}_1 = \{(\boldsymbol{x}_i, y_i)\}_{i=1}^{\frac{n}{2}}$ and $\mathcal{S}_2 = \{(\boldsymbol{x}_i, y_i)\}_{i=\frac{n}{2}+1}^n$ are two subsets of the sample $\mathcal{S}$.

**Assumption 1** (Optimal Function Estimator as an Oracle). *Given any function $f \in \mathcal{W}^{s,p}(\Omega)$ and $n \in \mathbb{N}$, let $\{\boldsymbol{x}_i\}_{i=1}^n$ be $n$ data points sampled independently and identically from the source distribution $\mathbb{P}^\circ$, whose p.d.f. $p^\circ$ is bounded bellow by $\underline{b}$ and above by $\bar{b}$ on $\Omega$. Assume that for $\frac{s}{d} > \frac{1}{p} - \frac{1}{2q}$, there exists an oracle $K_n : \Omega^n \times \mathbb{R}^n \to \mathcal{W}^{s,p}(\Omega)$ that estimates $f$ based on the $n$ points $\{\boldsymbol{x}_i\}_{i=1}^n$*

---

**Algorithm 1** Minimax Optimal Algorithm for Known Source and Target Distributions

1: **Input:** labeled data: $\mathcal{S} = \{(\boldsymbol{x}_i, y_i = f(\boldsymbol{x}_i))\}_{i=1}^n$.
2: Randomly split $\mathcal{S}$ into two parts $\mathcal{S}_1 = \{(\boldsymbol{x}_i, y_i)\}_{i=1}^{\frac{n}{2}}$ and $\mathcal{S}_2 = \{(\boldsymbol{x}_i, y_i)\}_{i=\frac{n}{2}+1}^n$.
3: Train a machine learning model on $\mathcal{S}_1$ to obtain $\hat{f}_{\mathcal{S}_1}(\boldsymbol{x})$.
4: Choose a sampling method based on $p^*(\boldsymbol{x})$ to compute $\mathbb{E}_{\boldsymbol{X} \sim \mathbb{P}^*}[\hat{f}_{\mathcal{S}_1}^q(\boldsymbol{X})]$.
5: Use $\mathcal{S}_2$ to estimate the $I_f^q$ based on equation 3.
6: **Output:** $q$-th moment under target: $I_f^q$.

---

*along with the $n$ observed function values $\{f(\boldsymbol{x}_i)\}_{i=1}^n$ and satisfies the following bound for any $r$ satisfying $\frac{1}{r} \in \left( \max\{\frac{d-sp}{pd}, 0\}, \max\{\frac{1}{p}, \mathbb{I}\{s > \frac{d}{p}\}\} \right]$:*

$$\left( \mathbb{E}_{\{\boldsymbol{x}_i\}_{i=1}^n} \left[ \left\| K_n(\{\boldsymbol{x}_i\}_{i=1}^n, \{f(\boldsymbol{x}_i)\}_{i=1}^n) - f \right\|_{\mathcal{L}^r(\Omega)}^r \right] \right)^{\frac{1}{r}} \lesssim \bar{b} \cdot n^{-\frac{s}{d} + (\frac{1}{p} - \frac{1}{r})_+}. \tag{4}$$

Assumption 1 illustrates that when the function is smooth enough, *i.e.*, $\frac{s}{d} > \frac{1}{p} - \frac{1}{2q}$, it becomes possible to construct an optimal function estimator, provided the source distribution is well-shaped. This assumption is also used in Blanchet et al. (2024), and many works (Wendland, 2004; Krieg & Sonnleitner, 2024; Blanchet et al., 2024) focus on constructing such estimators, notably through the moving least squares method, which achieves the convergence rate in equation 4. A construction of the desired oracle is deferred to Appendix B.3.

**Theorem 2** (Upper Bound on Moment Estimation with Smoothness). *Assume that $p \geq 2$, $q \leq p \leq 2q$ and $\frac{s}{d} > \frac{1}{p} - \frac{1}{2q}$. Let $(f, p^\circ, p^*) \in \mathcal{V}(\underline{b}, \bar{b}, B, s, p)$ and we have sample $\mathcal{S} = \{(\boldsymbol{x}_i, y_i = f(\boldsymbol{x}_i))\}_{i=1}^n$. If $\hat{f}_{\mathcal{S}_1}$ satisfies the Assumption 1, then the estimator $\hat{H}_B^q$ constructed in equation 3 above satisfies*

$$\mathbb{E}_{\mathcal{S}} \left[ \left| \hat{H}_B^q(\mathcal{S}) - I_f^q \right| \right] \lesssim B \cdot \bar{b} \cdot n^{\max\{-q(\frac{s}{d} - \frac{1}{p}) - 1, -\frac{1}{2} - \frac{s}{d}\}}. \tag{5}$$

**Construction Procedure.** We construct our moment estimator in a two-stage manner. First, we divide the sample $\mathcal{S}$ into two parts $\mathcal{S}_1 = \{(\boldsymbol{x}_i, y_i)\}_{i=1}^{\frac{n}{2}}$ and $\mathcal{S}_2 = \{(\boldsymbol{x}_i, y_i)\}_{i=\frac{n}{2}+1}^n$. Using a machine learning algorithm, we compute a nonparametric estimation $\hat{f}_{\mathcal{S}_1}$ of $f$ based on $\mathcal{S}_1$. In the second stage, we use $\mathcal{S}_2$ for calibration based on the following equation:

$$\mathbb{E}_{\boldsymbol{X} \sim \mathbb{P}^*}[f^q(\boldsymbol{X})] = \mathbb{E}_{\boldsymbol{X} \sim \mathbb{P}^*}[\hat{f}_{\mathcal{S}_1}^q(\boldsymbol{X})] + \mathbb{E}_{\boldsymbol{X} \sim \mathbb{P}^*}[f^q(\boldsymbol{X}) - \hat{f}_{\mathcal{S}_1}^q(\boldsymbol{X})]$$

$$= \mathbb{E}_{\boldsymbol{X} \sim \mathbb{P}^*}[\hat{f}_{\mathcal{S}_1}^q(\boldsymbol{X})] + \mathbb{E}_{X \sim \mathbb{P}^\circ}[w(\boldsymbol{X}) \cdot (f^q(\boldsymbol{X}) - \hat{f}_{\mathcal{S}_1}^q(\boldsymbol{X}))]. \tag{6}$$

Specifically, we estimate residual $\mathbb{E}_{\boldsymbol{X} \sim \mathbb{P}^*}[f^q(\boldsymbol{X})] - \mathbb{E}_{\boldsymbol{X} \sim \mathbb{P}^*}[\hat{f}_{\mathcal{S}_1}^q(\boldsymbol{X})]$, *i.e.*, the second term in equation 6 using samples from $\mathcal{S}_2$. This is given by $\frac{2}{n} \sum_{(\boldsymbol{x}_i, y_i) \in \mathcal{S}_2} w(\boldsymbol{x}_i) \cdot (y_i^q - \hat{f}_{\mathcal{S}_1}^q(\boldsymbol{x}_i))$. Since we assume that $p^*$ is known, the first term in equation 6 can be easily estimated. Finally, by combining these steps, we obtain our moment estimator, as summarized in Algorithm 1:

$$\hat{H}_B^q(\mathcal{S}) := \mathbb{E}_{\boldsymbol{X} \sim \mathbb{P}^*}[\hat{f}_{\mathcal{S}_1}^q(\boldsymbol{X})] + \frac{2}{n} \sum_{(\boldsymbol{x}_i, y_i) \in \mathcal{S}_2} w(\boldsymbol{x}_i) \cdot (y_i^q - \hat{f}_{\mathcal{S}_1}^q(\boldsymbol{x}_i)).$$

Note that the $\hat{f}_{\mathcal{S}_1}$ in equation 3 can be constructed using various method in practice (Künzel et al., 2019; Wager & Athey, 2018). Theorem 2 shows that if $\hat{f}_{\mathcal{S}_1}$ is optimal, *i.e.*, it satisfies Assumption 1 above, then the estimator $\hat{H}_B^q$ achieves the minimax lower bound up to a logarithmic factor.

### 4.3 $B$-INDEPENDENT BOUND

The bound derived in Theorem 2 depends on the intensity of the covariate shift. In some scenarios, the term $B \geq \sup_{\boldsymbol{x} \in \Omega} w(\boldsymbol{x})$, which measures this shift, can become exceedingly large even when the actual difference between $p^*$ and $p^\circ$ is relatively minor. This situation leads to the issue where large

values of the weighting function $w(\boldsymbol{x})$ in the estimator $\hat{H}_B^q$ introduce significant variance, despite the estimator maintaining its unbiased nature. High variance is problematic because it overshadows the benefits of unbiasedness, resulting in less reliable estimations. Consequently, it becomes crucial to balance bias and variance to achieve more stable estimators. To address this challenge, we propose using a truncated version of the estimator $\hat{H}_B^q$. The truncated estimator is defined as follows:

$$\hat{H}_T^q(\mathcal{S}) := \mathbb{E}_{\boldsymbol{X} \sim \mathbb{P}^*}[\hat{f}_{\mathcal{S}_1}^q(\boldsymbol{X})] + \frac{2}{n} \sum_{(\boldsymbol{x}_i, y_i) \in \mathcal{S}_2} \tau_T(w(\boldsymbol{x}_i)) \cdot (y_i^q - \hat{f}_{\mathcal{S}_1}^q(\boldsymbol{x}_i)), \tag{7}$$

where $\tau_T(\cdot) := \min\{\cdot, T\}$ is the truncation function, which limits the influence of large likelihood ratios by truncating values of $w(\boldsymbol{x})$ that exceed a predefined threshold $T > 0$.

**Theorem 3** (*B*-independent Bound). *Denote $\Omega_T^- := \{x : w(\boldsymbol{x}) > T\}$ as the area where the likelihood ratio exceeds the threshold $T$, and let $\mathbb{P}(\Omega_T^-) = g(T)$ denote the probability of this event. Under the conditions in Theorem 2, the upper bound for the variance and bias of the truncated estimator 7 are given as follows:*

$$Bias = \left| \mathbb{E}(\hat{H}_T^q) - I_f^q \right| \le \bar{b} \cdot \mathbb{P}(\Omega_T^-),$$

$$Variance = \mathbb{E}\left[ \hat{H}_T^q - \mathbb{E}(\hat{H}_T^q) \right]^2 \le \bar{b}^2 \cdot T^2 \cdot r(n)^2 + \bar{b}^2 \cdot \mathbb{P}(\Omega_T^-)^2.$$

*If we further assume that $g(T) \le T^{-\alpha}$, and pick $T = \mathcal{O}(r(n)^{-\frac{1}{\alpha+1}})$[4], then we obtain the following B-independent bound:*

$$\mathbb{E}_{\mathcal{S}} \left[ \left| \hat{H}_T^q(\mathcal{S}) - I_f^q \right| \right] \lesssim \bar{b} \cdot r(n)^{\frac{\alpha}{\alpha+1}}, \tag{8}$$

*where $r(n) = n^{\max\{-q(\frac{s}{d} - \frac{1}{p})-1, -\frac{1}{2} - \frac{s}{d}\}}$ is the same order in Theorem 2.*

Compared to the unbiased estimator $\hat{H}_B^q$ in Theorem 2, introducing the threshold $T$ adds a bias term when $T < B$. The threshold $T$ controls the trade-off between bias and variance: increasing $T$ reduces bias, while decreasing $T$ reduces variance. By carefully selecting the threshold $T$, we can mitigate the variance introduced by extreme likelihood ratios, thus achieving an optimal $B$-independent bound. However, eliminating the influence of $B$ comes at the cost of changing $r(n)$ to $r(n)^{\frac{\alpha}{\alpha+1}}$. This result is especially beneficial when $B$ is large, as it helps mitigate high variance, improving the stability of the algorithm. Even when $B$ is not large, the estimator $\hat{w}(\boldsymbol{x})$ may still be large, potentially introducing additional variance. To address this, we apply the truncated estimator $\hat{H}_T^q$, which we will discuss in Section 5.

**Remark.** *The additional assumption $g(T) \le T^{-\alpha}$ reduces the size of the class $\mathcal{V}$, potentially making the minimax lower bound in 2 less tight. Consequently, the upper bound in 8 may be better than the minimax lower bound 2 when $B$ is large. A more detailed discussion of this assumption is provided in Appendix B.2.*

## 5 STABILIZED ALGORITHM FOR UNKNOWN LIKELIHOOD RATIOS

In Section 4, we discuss the importance of the likelihood ratio $w(\boldsymbol{x})$ in debiasing estimators. However, the related discussion is limited to the ideal scenario where both the source p.d.f. $p^\circ(\boldsymbol{x})$ and the target p.d.f. $p^*(\boldsymbol{x})$ are known, allowing direct computation of $w(\boldsymbol{x})$. In practice, these distributions are usually unknown, making the calculation of the likelihood ratio challenging.

Fortunately, we can still gather a series of unlabeled data from the target domain when both $p^\circ(\boldsymbol{x})$ and $p^*(\boldsymbol{x})$ are unknown. Specifically, we have access to $n$ labeled samples from the source distribution, denoted as $\mathcal{S} = \{(\boldsymbol{x}_i, y_i = f(\boldsymbol{x}_i))\}_{i=1}^n$. Additionally, we have $m \ (\gg n)$ unlabeled samples from the target distribution, represented by $\mathcal{S}' = \{\boldsymbol{x}_i'\}_{i=1}^m$.

There are various methods available for estimating the likelihood ratio $w(\boldsymbol{x})$ (Yu & Szepesvári, 2012; Bickel et al., 2007; Kpotufe, 2017). For example, we employ a classification-based approach

---

[4]The term depends on the parameters defining the class $\mathcal{V}$, which brings insight about the relation between the choice of $T$ and the smoothness of function, intensity of covariate shift. Unfortunately, they are unknown in practice, we can only determine the appropriate truncation point through a grid search method.

to obtain the estimator $\hat{w}(\boldsymbol{x})$, with further details provided in Appendix A.7. However, it is inevitable that the estimation procedure introduces large values of $\hat{w}(\boldsymbol{x})$, even when the true value $w(\boldsymbol{x})$ is relatively small. To address this issue, we adopt the truncated version of the moment estimator from Section 4.3. Specifically, we replace the unknown weight function $w(\boldsymbol{x})$ in the estimator 7 with $\hat{w}(\boldsymbol{x})$. Furthermore, since the target distribution $p^*(\boldsymbol{x})$ is unknown, we cannot directly compute the first term $\mathbb{E}_{\boldsymbol{X} \sim \mathbb{P}^*}[\hat{f}_{\mathcal{S}_1}^q(\boldsymbol{X})]$. To address this, we use unlabeled data to approximate it. The modified estimator is given by equation 9 and the entire procedure is summarized in Algorithm 2:

$$\tilde{H}_T^q(\mathcal{S} \cup \mathcal{S}') := \frac{1}{m}\sum_{i=1}^m \hat{f}_{\mathcal{S}_1}^q(\boldsymbol{x}_i') + \frac{2}{n}\sum_{(\boldsymbol{x}_i, y_i) \in \mathcal{S}_2} \tau_T(\hat{w}(\boldsymbol{x}_i)) \cdot (y_i^q - \hat{f}_{\mathcal{S}_1}^q(\boldsymbol{x}_i)). \tag{9}$$

## 5.1 Convergence Rate and Double Robustness

In this subsection, we first examine the convergence rate of the estimator 9 under the same assumptions in Theorem 3. Additionally, we demonstrate the double robustness of the estimator 9.

The difference between estimators 7 and 9 lies in the need to estimate additional quantities when $p^\circ(\boldsymbol{x})$ and $p^*(\boldsymbol{x})$ are unknown. The following theorem shows how these unknown quantities affect the convergence rate of the estimator 9.

**Theorem 4** (Convergence Rate of Plug-in Truncated Estimator)**.** *Under the same assumptions in Theorem 3, the convergence rate of the plug-in truncated estimator $\tilde{H}_T^q$ is:*

$$\mathbb{E}\left[\left|\tilde{H}_T^q(\mathcal{S} \cup \mathcal{S}') - I_f^q\right|\right] \lesssim \mathbb{E}\left[\left|\hat{H}_T^q(\mathcal{S}) - I_f^q\right|\right] + m^{-\frac{1}{2}}. \tag{10}$$

Theorem 4 indicates that the only additional cost to the convergence rate for the unknown source and target distributions is $m^{-\frac{1}{2}}$. This term represent the Monte Carlo simulation error for estimating $\mathbb{E}_{\boldsymbol{X} \sim \mathbb{P}^*}[\hat{f}_{\mathcal{S}_1}^q(\boldsymbol{X})]$ using $m$ unlabeled data points $\mathcal{S}'$. The additional error from estimating $w(\boldsymbol{x})$ is bounded above by the error of the estimator $\hat{H}_T^q$, due to the truncation applied to the estimator $\hat{w}(\boldsymbol{x})$.

**Proof Sketch.** The relationship between the convergence rate of $\hat{H}_T^q$ and $\tilde{H}_T^q$ is established through the following error decomposition:

$$\mathbb{E}\left[\left|\tilde{H}_T^q - I_f^q\right|\right] \leq \underbrace{\mathbb{E}_{\mathcal{S} \cup \mathcal{S}'}\left[\left|\frac{2}{n}\sum_{(\boldsymbol{x}_i, y_i) \in \mathcal{S}_2}(\tau_T(\hat{w}(\boldsymbol{x}_i)) - \tau_T(w(\boldsymbol{x}_i))) \cdot (y_i^q - \hat{f}_{\mathcal{S}_1}^q(\boldsymbol{x}_i))\right|\right]}_{\text{additional error for estimating } w(\boldsymbol{x})} +$$

$$\underbrace{\mathbb{E}_{\mathcal{S} \cup \mathcal{S}'}\left[\left|\frac{1}{m}\sum_{i=1}^m \hat{f}_{\mathcal{S}_1}^q(\boldsymbol{x}_i') - \mathbb{E}_{\boldsymbol{X} \sim \mathbb{P}^*}[\hat{f}_{\mathcal{S}_1}^q(\boldsymbol{X})]\right|\right]}_{\text{Monte Carlo simulation error}} + \underbrace{\mathbb{E}_{\mathcal{S} \cup \mathcal{S}'}\left[\left|\hat{H}_T^q - I_f^q\right|\right]}_{\text{error of the case when } w(\boldsymbol{x}) \text{ is known}}.$$

The first term represents the additional error due to estimating the likelihood ratio $w(\boldsymbol{x})$. Due to truncation, this term is bounded above by $T$ and the estimation error of $\hat{f}_{\mathcal{S}_1}^q(\boldsymbol{x})$, which is further bounded by the third term. The second term accounts for the Monte Carlo simulation error in estimating $\mathbb{E}_{\boldsymbol{X} \sim \mathbb{P}^*}[\hat{f}_{\mathcal{S}_1}^q(\boldsymbol{X})]$ using $m$ unlabeled data $\mathcal{S}'$ form target domain, with a convergence rate $\mathcal{O}(m^{-\frac{1}{2}})$. The third term corresponds to the estimation error of $\hat{H}_T^q$ as discussed in Theorem 3.

In our approach, two models play a crucial role: one for estimating the likelihood ratio $w(\boldsymbol{x})$ and another for estimating the function $f(\boldsymbol{x})$ itself. So far, assumptions have been made about $\hat{f}_{\mathcal{S}_1}(\boldsymbol{x})$ but not about $\hat{w}(\boldsymbol{x})$. If assumptions are instead made only about $\hat{w}(\boldsymbol{x})$, the conclusion is as follows.

**Corollary 1** (Double Robustness of Plug-in Truncated Estimator)**.** *If $\mathbb{P}(\Omega_T^-) \to 0$ as $n \to \infty$, then the estimator $\tilde{H}_T^q$ exhibits double robustness. Specifically, this means that as long as either $\hat{w}(\boldsymbol{x})$ or $\hat{f}_{\mathcal{S}_1}(\boldsymbol{x})$ is consistent, the moment estimator $\tilde{H}_T^q$ is also consistent.*

Corollary 1 provides guidance on selecting $\hat{w}(\boldsymbol{x})$ and $\hat{f}_{\mathcal{S}_1}(\boldsymbol{x})$ to maintain consistency when achieving the optimal estimator of $f$ that satisfies Assumption 1 is not feasible in practice. The assumption

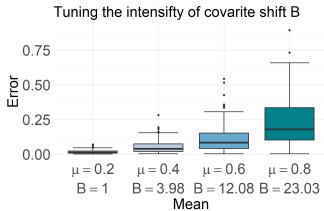 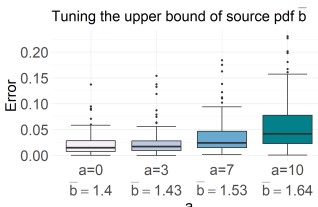 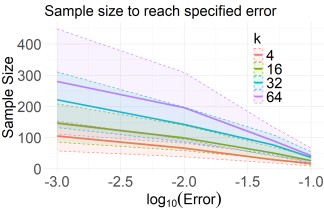

(a) Effect of covariate shift.  (b) Effect of sampling strategy.  (c) Effect of function class.

Figure 2: Impact of covariate shift, sampling strategy, and function class on the error of the estimator, measured by the $L_1$ norm between the true and estimated values. **(a)** Boxplot of the error across different levels of covariate shift intensity $B$, which is adjusted by fixing $\mathbb{P}^\circ$ as truncated normal distribution with a mean of $0.2$ and altering $\mathbb{P}^*$ be the same truncated normal with a shifted mean $\mu$ to vary $B$ while keeping $\bar{b}$ constant. As $B$ increases, the error increases and the stability of the algorithm decreases. **(b)** Boxplot of the error across different levels of the upper bound $\bar{b}$ of the $\mathbb{P}^\circ$, with fixed $\mathbb{P}^*$ as uniform distribution and modifying $\mathbb{P}^\circ$ with a hyperparameter $a$ that vary $\bar{b}$ while keeping $B$ constant. As $\bar{b}$ increases, the error increases and the stability of the algorithm decreases. **(c)** Required sample size to achieve a specified error for different smoothness levels, controlled by hyperparameter $k$. A larger $k$ indicates less smoothness, resulting in a slower convergence rate.

$\mathbb{P}(\Omega_T^-) \to 0$ as $n \to \infty$ indicates that the probability of the truncated region $\Omega_T^-$, diminishes as the sample size increases, thus ensuring that the influence of extreme values becomes negligible in large samples. It can be achieved if we follow the rule in Theorem 3.

## 6 NUMERICAL EXPERIMENTS

In this section, we generate synthetic data to illustrate three key factors that influence the problem discussed in Section 4.1 , as well as the effect of truncation covered in Section 4.3. We set $q = 2$ and the machine learning algorithm employed to learn $\hat{f}_{\mathcal{S}_1}$ is the random forest. Each experiment is repeated 100 times. To highlight the advantages of our proposed two-stage estimator under covariate shift, we compare it against the Monte Carlo estimator and the one-stage estimator on both synthetic data (Figure 3) and real-world data (Figure 5) with detailed explanations provided in Appendix C.

**Covariate Shift.** We set $\mathbb{P}^\circ$ as a truncated normal distribution on $[0, 1]$ with a mean of $0.2$ and a standard deviation of $0.3$, denoted by $\texttt{tnorm}([0, 1], 0.2, 0.3)$. To vary the intensity of covariate shift, we consider the target distributions $\texttt{tnorm}([0, 1], \mu, 0.3)$, adjusting the mean parameter $\mu$. We set $f(x) = 1 + x^2 + \frac{1}{5}\sin(16x)$ and select $\mu = 0.2, 0.4, 0.6, 0.8$, with the corresponding $B$ values being $1.00, 3.98, 12.08, 23.03$, respectively.

**Sampling Strategy.** We set $\mathbb{P}^*$ as uniform distribution over $[0, 1]$. To maintain the covariate shift, we consider source distributions with the following probability density functions:

$$p(x; a) = ax^3 + (-\frac{24}{5} - \frac{3}{2}a)x^2 + (\frac{1}{2}a + \frac{24}{5})x + \frac{1}{5}.$$

which ensures that $B = 5$. We set $f(x) = 1 + x^2 + \frac{1}{5}\sin(16x)$ and select $a = 0, 3, 7, 10$, with the corresponding $\bar{b}$ values $1.40, 1.43, 1.53, 1.64$, respectively.

**Function Class.** We set $\mathbb{P}^*$ as $\texttt{tnorm}([0, 1], 0.6, 0.6)$ and $\mathbb{P}^\circ$ as uniform distribution over $[0, 1]$. To examine the impact of the function class, we select functions $f$ from the following class:

$$f(x; k) = 1 + x^2 + \frac{1}{5}\sin(k \cdot x),$$

where increasing $k$ leads to a less smooth function. In our analysis, we use $k = 4, 16, 32, 64$. To determine the required sample size to achieve specified error levels of $0.1, 0.05, 0.01, 0.001$, we start with a sample size of $10$ and increment it by $10$ until the error falls below $0.001$. For each specified error level, we select the sample size for which the error is closest to the desired value.

The numerical experiments reveal that the error rates increase with the intensity of covariate shift, as illustrated in Figure 2(a). When varying the hyperparameter $B$, which controls the mean shift in the

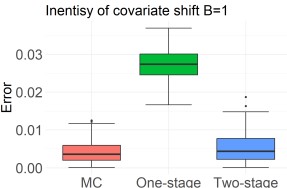 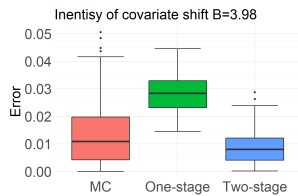 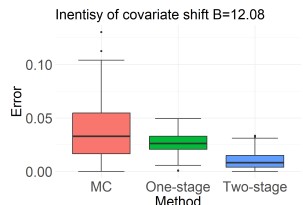

Figure 3: Comparison of the Monte Carlo estimator, one-stage estimator and two-stage estimator on synthetic data, with performance measured by the $L_1$ norm between true and estimated values. Each subfigure presents boxplots of the errors for different methods across varying levels of covariate shift $B$. The Monte Carlo estimator performs better than the two-stage methods when there is no covariate shift ($B = 1$). However, as $B$ increases, the two-stage estimator demonstrates greater accuracy and improved stability, highlighting its necessity under significant covariate shift.

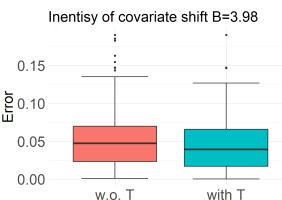 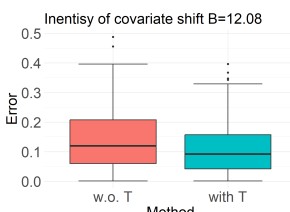 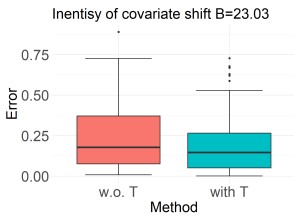

Figure 4: Effect of truncation. Each subfigure presents boxplots of errors for both untruncated and truncated estimators across different levels of covariate shift $B$. Truncation improves stability while preserving accuracy, with its impact becoming more pronounced as $B$ increases.

source distribution, we observe that larger shifts lead to higher error rates. This pattern is consistent across different settings of the function class and sampling strategies, as depicted in Figures 2(b) and (c). These findings underscore the importance of both the degree of covariate shift and the smoothness of the function in determining model performance.

**Effect of truncation.** To illustrate how truncation enhances the stability of the two-stage method, we compare the untruncated estimator from equation 1 with the truncated estimator from equation 7 using $T = \frac{1}{2}B$. The source distribution $\mathbb{P}^\circ$ is defined as a truncated normal distribution on $[0, 1]$ with mean $0.2$ and standard deviation $0.3$, denoted by $\texttt{tnorm}([0, 1], 0.2, 0.3)$. To vary the intensity of covariate shift, we consider the target distributions $\texttt{tnorm}([0, 1], \mu, 0.3)$, adjusting the mean parameter $\mu$. We set $f(x) = 1 + x^2 + \frac{1}{5}\sin(16x)$ and select $\mu = 0.4, 0.6, 0.8$, resulting in corresponding $B$ values of 3.98, 12.08, and 23.03, respectively.

The results are presented in Figure 4, showing that truncation enhances stability while preserving accuracy, with its effect becoming more significant as $B$ increases.

# 7    CONCLUSION

In this paper, we explore how to calibrate an optimal function estimator to maintain its optimality for moment estimation under covariate shift. Firstly, we establish the minimax lower bound for this problem and develop an optimal two-stage algorithm when both the source and target distributions are known. Our findings indicate that the convergence rate is influenced by three factors: the intensity of covariate shift, the sampling strategy, and the properties of the function class. Furthermore, we propose a truncated version of the estimator that ensures double robustness and provide the corresponding upper bound when the source and target distributions are unknown. This is important in practice because estimating the likelihood ratio can often be unstable.

Nonetheless, several potential extensions warrant exploration. This paper utilizes the uniformly $B$-bounded class; however, alternative choices, such as the moment-bounded class, are also feasible. Additionally, our assumptions, including the constraints on the parameters $p$ and $q$ and the bounded source p.d.f., may be extremely restrictive for some application scenarios. We leave the relaxation of these assumptions for future work. Furthermore, our analysis focuses on moment functionals. A potential direction for future research is to explore more complex functionals to investigate whether the optimality still holds given the optimal function estimator under covariate shift.

ACKNOWLEDGMENTS

This work is supported by Shanghai Science and Technology Development Funds 24YF2711700 and Fundamental Research Funds for the Central Universities 2024110586. Xin Liu is partially supported by the National Natural Science Foundation of China 12201383, Shanghai Pujiang Program 21PJC056, and Innovative Research Team of Shanghai University of Finance and Economics. Shaoli Wang is partially supported by the National Natural Science Foundation of China NSFC72271148.

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

# A APPENDIX

## A.1 PROOF OF THEOREM 1

**Theorem 1** (Minimax Lower Bound on Estimating the Moment under Covariate Shift). *Let $\mathcal{H}_n^{f,q}$ denote the class of all the estimators using $\mathcal{S} = \{(\boldsymbol{x}_i, y_i = f(\boldsymbol{x}_i))\}_{i=1}^n$ to estimate the $q$-th moment of $f$ under $\mathbb{P}^*$. For any given target p.d.f. $p^* \in \{p^* : (\cdot, \cdot, p^*) \in \mathcal{V}\}$, when $p \geq 2$, $q \leq p \leq 2q$ and $(f, p^\circ) \in \mathcal{V}_{p^*} := \{(f, p^\circ) : (f, p^\circ, p^*) \in \mathcal{V}\}$, it holds that*

$$\inf_{\hat{H}^q \in \mathcal{H}_n^{f,q}} \sup_{(f,p^\circ) \in \mathcal{V}_{p^*}} \mathbb{E}_{\mathcal{S}} \left[ \left| \hat{H}^q(\mathcal{S}) - I_f^q \right| \right] \gtrsim B \cdot \bar{b} \cdot n^{\max\{-q(\frac{s}{d} - \frac{1}{p}) - 1, -\frac{1}{2} - \frac{s}{d}\}}. \quad (2)$$

**Proof Sketch.** The primary strategy for deriving minimax lower bounds involves creating instances from $\mathcal{V}$ that are similar enough to be statistically indistinguishable, while ensuring that the corresponding $q$-th moments are as distinct as possible. In this problem, we encounter two extreme cases. The first is the rare event case, characterized by a function that has a sharp peak confined to a specific small region and is otherwise constant. Moreover, the covariate shift accentuates this peak's prominence; specifically, the target distribution assigns greater probability mass to the peak region than the source distribution. The second case involves a function that continuously oscillates around a constant. Utilizing the construction method outlined above, we derive the result based on the method of two fuzzy hypotheses.

**Lemma 1** (Method of Two Fuzzy Hypotheses (Tsybakov, 2009)). *Let $F : \Theta \to \mathbb{R}$ be some continuous functional defined on the measurable space $(\Theta, \mathcal{U})$ and taking values in $(\mathbb{R}, \mathcal{B}(\mathbb{R}))$, where $\mathcal{B}(\mathbb{R})$ denotes the Borel $\sigma$-algebra on $\mathbb{R}$. Suppose that each parameter $\theta \in \Theta$ is associated with a distribution $\mathbb{P}_\theta$, which together form a collection $\{\mathbb{P}_\theta : \theta \in \Theta\}$ of distributions.*

*For any fixed $\theta \in \Theta$, assume that our observation $\boldsymbol{X}$ is distributed as $\mathbb{P}_\theta$. Let $\hat{F}$ be an arbitrary estimator of $F(\theta)$ based on $\boldsymbol{X}$. Let $\mu_0, \mu_1$ be two prior measures on $\Theta$. Assume that there exist constants $c \in \mathbb{R}$, $\Delta \in (0, \infty)$ and $\beta_0, \beta_1 \in [0, 1]$, such that:*

$$\mu_0(\theta \in \Theta : F(\theta) \leq c - \Delta) \geq 1 - \beta_0,$$
$$\mu_1(\theta \in \Theta : F(\theta) \geq c + \Delta) \geq 1 - \beta_1.$$

*For $j \in \{0, 1\}$, we use $\mathbb{P}_j(\cdot) := \int \mathbb{P}_\theta(\cdot) \mu_j(d\theta)$ to denote the marginal distribution $\mathbb{P}_j$ associated with the prior distribution $\mu_j$. Then we have the following lower bound:*

$$\inf_{\hat{F}} \sup_{\theta \in \Theta} \mathbb{P}_\theta(|\hat{F} - F(\theta)| \geq \Delta) \geq \frac{1 - TV(\mathbb{P}_0 || \mathbb{P}_1) - \beta_0 - \beta_1}{2},$$

*where $TV(\cdot \mid \cdot)$ denotes the total variation (TV) distance.*

*Proof.* We derive the minimax lower bounds established in Theorem 1 using the method of two fuzzy hypotheses outlined above. Firstly, we introduce some notations used in our proof. Let $\mathcal{C}(\Omega)$

denote the space of all continuous function $f : \Omega \to \mathbb{R}$ and $\lfloor \cdot \rfloor$ be the rounding function. For any $s > 0$ and $f \in \mathcal{C}(\Omega)$, we define the Hölder norm $\| \cdot \|_{\mathcal{C}^s(\Omega)}$ by:

$$\|f\|_{\mathcal{C}^s(\Omega)} := \max_{|k| \leq \lfloor s \rfloor} \|D^k f\|_{\mathcal{L}^\infty(\Omega)} + \max_{|k|=\lfloor s \rfloor} \sup_{x,y \in \Omega, x \neq y} \frac{|D^k f(x) - D^k f(y)|}{\|x - y\|^{s - \lfloor s \rfloor}}.$$

The corresponding Hölder space is defined as $\mathcal{C}^s(\Omega) := \left\{ f \in \mathcal{C}(\Omega) : \|f\|_{\mathcal{C}^s(\Omega)} < \infty \right\}$. Consider the function $K_0$ defined as follows:

$$K_0(\boldsymbol{x}) := \prod_{i=1}^{d} \exp\left( -\frac{1}{1 - x_i^2} \right) \mathbb{I}(|x_i| \leq 1), \forall \boldsymbol{x} = (x_1, x_2, \cdots, x_d) \in \mathbb{R}^d,$$

then we define function $K(\boldsymbol{x}) := K_0(2\boldsymbol{x}), \forall \boldsymbol{x} \in \mathbb{R}^d$. From the construction of $K_0$ and $K$ above, we have that $K \in \mathcal{C}^\infty(\mathbb{R}^d)$ and compactly supported on $[-\frac{1}{2}, \frac{1}{2}]^d$. The function $K$ is used to model the small bump. Furthermore, we partition the domain $\Omega$ into $m^d$ small cubes $\Omega_1, \ldots, \Omega_{m^d}$, each with a side length of $m^{-1}$, where $m = (200n)^{\frac{1}{d}}$. Moreover, we use $\vec{x} := (\boldsymbol{x}_1, \ldots, \boldsymbol{x}_n)$ and $\vec{y} := (y_1, \ldots, y_n)$ to denote the two $n$-dimensional vectors formed by the sample $\mathcal{S}$. With the preliminaries established, we now present the main proof below, addressing the two different cases:

(Case I) Construct the lower bound instance: let $p^\circ(\boldsymbol{x})$ be any p.d.f. that is bounded bellow by $\underline{b}$ and above by $\bar{b}$. Since $\int_\Omega p^\circ(\boldsymbol{x})d\boldsymbol{x} = \sum_{j=1}^{m^d} \int_{\Omega_j} p^\circ(\boldsymbol{x})d\boldsymbol{x} = 1$, there exists an $i^*$ such that $\mathbb{P}(\boldsymbol{x} : \boldsymbol{x} \in \Omega_{i^*}) = \int_{\Omega_{i^*}} p^\circ(\boldsymbol{x})d\boldsymbol{x} \leq \frac{1}{m^d}$. Let $p^*(\boldsymbol{x})$ be the p.d.f. of the uniform distribution over $[\frac{i^*}{m}, \frac{i^*}{m} + (\frac{1}{B})^{\frac{1}{d}}]^d$. Consider the following two functions $g_0$ and $g_1$:

$$g_0(\boldsymbol{x}) \equiv 0 \quad (\forall \boldsymbol{x} \in \Omega),$$

$$g_1(\boldsymbol{x}) = \begin{cases} \bar{b}^{\frac{1}{q}} \cdot m^{-s+\frac{d}{p}} K(m(\boldsymbol{x} - c_{i^*})) & (\boldsymbol{x} \in \Omega_{i^*}), \\ 0 & (\text{otherwise}). \end{cases}$$

This construction is commonly used in the proof of nonparametric statistics, see Chapter 15 in Wainwright (2019). The choice of $i^*$ models the worst-case scenario of covariate shift. The functions $g_0$ and $g_1$ are in the $\mathcal{W}^{s,p}(\Omega)$, as proven in Theorem 2.1 of Blanchet et al. (2024).

Next, we compute the $q$-th moment under the target distribution $\mathbb{P}^*$. Obviously, $I_{g_0}^q = 0$, so we only need to compute $I_{g_1}^q$:

$$I_{g_1}^q = \int_\Omega B \cdot g_1^q(\boldsymbol{x})d\boldsymbol{x} = B \cdot \bar{b} \cdot \int_{\Omega_{i^*}} \left[ m^{-s+\frac{d}{p}} K(m(\boldsymbol{x} - c_1)) \right]^q d\boldsymbol{x}$$

$$= B \cdot \bar{b} \cdot m^{-q(s-\frac{d}{p})} \cdot \int_{\Omega_{i^*}} (K(y))^q \frac{1}{m^d} dy \gtrsim B \cdot \bar{b} \cdot m^{-q(s-\frac{d}{p})-d}. \tag{11}$$

Construct two discrete measures $\mu_0, \mu_1$ supported on $\{g_0, g_1\} \subset \mathcal{W}^{s,p}(\Omega)$ as bellow:

$$\mu_0(\{g_0\}) = \frac{1+\epsilon}{2}, \mu_0(\{g_1\}) = \frac{1-\epsilon}{2},$$

$$\mu_1(\{g_0\}) = \frac{1-\epsilon}{2}, \mu_1(\{g_1\}) = \frac{1+\epsilon}{2}.$$

Take $\epsilon = \frac{1}{2}, c = \Delta = \frac{1}{2}I_{g_1}^q$ and $\beta_0 = \beta_1 = \frac{1-\epsilon}{2}$, we obtain that:

$$\mu_0(f \in W^{s,p}(\Omega) : I_f^q \leq c - \Delta) = \mu_0(I_f^q \leq 0) \geq \frac{1+\epsilon}{2} = 1 - \beta_0,$$

$$\mu_1(f \in W^{s,p}(\Omega) : I_f^q \geq c + \Delta) = \mu_1(I_f^q \geq I_{g_1}^q) \geq \frac{1+\epsilon}{2} = 1 - \beta_1.$$

We proceed to compute the mixture distribution under $\mu_0$ and $\mu_1$:

$$\mathbb{P}_0(\vec{x}, \vec{y}) = \mathbb{P}(\vec{x}) \cdot \mathbb{P}_0(\vec{y} \mid \vec{x}) = \mathbb{P}(\vec{x}) \cdot \int \mathbb{P}(\vec{y} \mid \vec{x}, \theta) \mu_0(d\theta)$$

$$= \left( \prod_{i=1}^{n} p^\circ(\boldsymbol{x}_i) \right) \cdot \left( \frac{1+\epsilon}{2} \prod_{i:\boldsymbol{x}_i \in \Omega_{i*}} \delta_0(y_i) + \frac{1-\epsilon}{2} \prod_{i:\boldsymbol{x}_i \in \Omega_{i*}} \delta_{g_1(\boldsymbol{x}_i)}(y_i) \right) \cdot \prod_{j \neq i^*} \left( \prod_{i:\boldsymbol{x}_i \in \Omega_j} \delta_0(y_i) \right),$$

$$\mathbb{P}_1(\vec{x}, \vec{y}) = \mathbb{P}(\vec{x}) \cdot \mathbb{P}_1(\vec{y} \mid \vec{x}) = \mathbb{P}(\vec{x}) \cdot \int \mathbb{P}(\vec{y} \mid \vec{x}, \theta) \mu_1(d\theta)$$

$$= \left( \prod_{i=1}^{n} p^\circ(\boldsymbol{x}_i) \right) \cdot \left( \frac{1-\epsilon}{2} \prod_{i:\boldsymbol{x}_i \in \Omega_{i*}} \delta_0(y_i) + \frac{1+\epsilon}{2} \prod_{i:\boldsymbol{x}_i \in \Omega_{i*}} \delta_{g_1(\boldsymbol{x}_i)}(y_i) \right) \cdot \prod_{j \neq i^*} \left( \prod_{i:\boldsymbol{x}_i \in \Omega_j} \delta_0(y_i) \right),$$

where $\delta_a(\cdot)$ denotes the Dirac delta distribution at point $a$, *i.e.*, $\int_{-\infty}^{\infty} f(x)\delta_a(x)dx = f(a)$ for any function $f : \mathbb{R} \to \mathbb{R}$.

Then we may compute KL divergence between $\mathbb{P}_0$ and $\mathbb{P}_1$ as bellow:

$$KL(\mathbb{P}_0 \| \mathbb{P}_1) = \mathbb{E}\left[ \log(\frac{\mathbb{P}_0(\vec{x}, \vec{y})}{\mathbb{P}_1(\vec{x}, \vec{y})}) \mathbb{P}_0(\vec{x}, \vec{y}) \right] = \mathbb{E}_{\vec{x}} \mathbb{E}_{\vec{y}|\vec{x}} \left[ \log(\frac{\mathbb{P}_0(\vec{x}, \vec{y})}{\mathbb{P}_1(\vec{x}, \vec{y})}) \mathbb{P}_0(\vec{x}, \vec{y}) \right]$$

$$= \mathbb{E}_{\vec{x}} \left[ \int_{-\infty}^{\infty} \cdots \int_{-\infty}^{\infty} \log \left( \frac{\frac{1+\epsilon}{2} \prod_{i:\boldsymbol{x}_i \in \Omega_{i*}} \delta_0(y_i) + \frac{1-\epsilon}{2} \prod_{i:\boldsymbol{x}_i \in \Omega_{i*}} \delta_{g_1(\boldsymbol{x}_i)}(y_i)}{\frac{1-\epsilon}{2} \prod_{i:\boldsymbol{x}_i \in \Omega_{i*}} \delta_0(y_i) + \frac{1+\epsilon}{2} \prod_{i:\boldsymbol{x}_i \in \Omega_{i*}} \delta_{g_1(\boldsymbol{x}_i)}(y_i)} \right) \right.$$

$$\left. \cdot \left( \frac{1+\epsilon}{2} \prod_{i:\boldsymbol{x}_i \in \Omega_{i*}} \delta_0(y_i) + \frac{1-\epsilon}{2} \prod_{i:\boldsymbol{x}_i \in \Omega_{i*}} \delta_{g_1(\boldsymbol{x}_i)}(y_i) \right) \cdot \left( \prod_{j \neq i^*} \prod_{i:\boldsymbol{x}_i \in \Omega_j} \delta_0(y_i) \right) \prod_{i=1}^{n} dy_i \right]$$

$$= \mathbb{E}_{\vec{x}} \left[ \int_{-\infty}^{\infty} \cdots \int_{-\infty}^{\infty} \log \left( \frac{\frac{1+\epsilon}{2} \prod_{i:\boldsymbol{x}_i \in \Omega_{i*}} \delta_0(y_i) + \frac{1-\epsilon}{2} \prod_{i:\boldsymbol{x}_i \in \Omega_{i*}} \delta_{g_1(\boldsymbol{x}_i)}(y_i)}{\frac{1-\epsilon}{2} \prod_{i:\boldsymbol{x}_i \in \Omega_{i*}} \delta_0(y_i) + \frac{1+\epsilon}{2} \prod_{i:\boldsymbol{x}_i \in \Omega_{i*}} \delta_{g_1(\boldsymbol{x}_i)}(y_i)} \right) \right.$$

$$\left. \cdot \left( \frac{1+\epsilon}{2} \prod_{i:\boldsymbol{x}_i \in \Omega_{i*}} \delta_0(y_i) + \frac{1-\epsilon}{2} \prod_{i:\boldsymbol{x}_i \in \Omega_{i*}} \delta_{g_1(\boldsymbol{x}_i)}(y_i) \right) \prod_{i:\boldsymbol{x}_i \in \Omega_{i*}} dy_i \right]$$

$$= \left( \log \left( \frac{1+\epsilon}{1-\epsilon} \right) \frac{1+\epsilon}{2} + \log \left( \frac{1-\epsilon}{1+\epsilon} \right) \frac{1-\epsilon}{2} \right) \mathbb{P} \left( \{i : \boldsymbol{x}_i \in \Omega_{i*}\} \neq \varnothing \right)^5$$

$$= \epsilon \log \left( \frac{1+\epsilon}{1-\epsilon} \right) \mathbb{P} \left( \{i : \boldsymbol{x}_i \in \Omega_{i*}\} \neq \varnothing \right).$$

Since $\mathbb{P}(\boldsymbol{x} : \boldsymbol{x} \in \Omega_{i*}) \leq \frac{1}{m^d}$, it follows that:

$$\mathbb{P} \left( \{i : \boldsymbol{x}_i \in \Omega_{i*}\} \neq \varnothing \right) = 1 - \mathbb{P} \left( \{i : \boldsymbol{x}_i \in \Omega_{i*}\} = \varnothing \right)$$

$$= 1 - (1 - \mathbb{P}(x \in \Omega_{i*}))^n \leq 1 - (1 - \frac{1}{m^d})^n$$

$$\leq 1 - \left( \frac{1}{e}(1 - \frac{1}{m^d}) \right)^{\frac{1}{200}} \leq 1 - (2e)^{-\frac{1}{200}}.$$

Then, use the Pinkser's inequality to upper bound the TV distance between $\mathbb{P}_0$ and $\mathbb{P}_1$ as bellow:

$$TV(\mathbb{P}_0 \| \mathbb{P}_1) \leq \sqrt{\frac{1}{2} KL(\mathbb{P}_0 \| \mathbb{P}_1)} \leq \sqrt{\frac{1 - (2e)^{-\frac{1}{200}}}{2} \epsilon \log \left( \frac{1+\epsilon}{1-\epsilon} \right)} \leq \sqrt{\frac{3}{100}} \epsilon = \frac{\sqrt{3}}{10} \epsilon. \quad (12)$$

---

[5] Using the property of Dirac measure: $\int_{-\infty}^{\infty} f(x)\delta_a(x)dx = f(a)$.

Finally, combining 11 and 12, we obtain:

$$\inf_{\hat{H}^q \in \mathcal{H}_n^{f,q}} \sup_{(f,p^\circ,p^*) \in \mathcal{V}} \mathbb{E}_{\mathcal{S}} \left[ \left| \left| \hat{H}^q(\mathcal{S}) - I_f^q \right| \right| \right]$$

$$\geq \Delta \inf_{\hat{H}^q \in \mathcal{H}_n^{f,q}} \sup_{(f,p^\circ,p^*) \in \mathcal{V}} \mathbb{P}_{\mathcal{S}} \left[ \left| \hat{H}^q(\mathcal{S}) - I_f^q \right| \geq \Delta \right]$$

$$\geq \frac{1}{2} I_{g_1}^q \frac{1 - TV(\mathbb{P}_0 || \mathbb{P}_1) - \beta_0 - \beta_1}{2} \geq \frac{1}{4} \left( 1 - \frac{\sqrt{3}}{10} \right) \epsilon I_{g_1}^q$$

$$\gtrsim B \cdot \bar{b} \cdot n^{-q(\frac{s}{d} - \frac{1}{p}) - 1}, \tag{13}$$

which is the first term in the RHS of equation 2.

(Case II) We proceed to construct the second lower bound instance: let $p^\circ(\boldsymbol{x})$ and $p^*(\boldsymbol{x})$ be the p.d.f. of the uniform distribution over $[0,1]^d$. For any $1 \leq j \leq m^d$, consider $f_j$ defined as follows:

$$f_j(\boldsymbol{x}) = \begin{cases} m^{-s} K(m(\boldsymbol{x} - c_j)) & (\boldsymbol{x} \in \Omega_j), \\ 0 & \text{(otherwise)}, \end{cases}$$

we further pick two constants $\alpha, M$ satisfying $\alpha := \|K\|_{\mathcal{L}^\infty([-\frac{1}{2}, \frac{1}{2}]^d)}$ and $M = 3\alpha$. Now consider the following finite set of $2^{m^d}$ functions:

$$\mathcal{F} := \left\{ (B\bar{b})^{\frac{1}{q}} \cdot (M + \sum_{j=1}^{m^d} \eta_j f_j) : \eta_j \in \{\pm 1\}, \forall 1 \leq j \leq m^d \right\}.$$

Any function in $\mathcal{F}$ belongs to $\mathcal{W}^{s,p}(\Omega)$ as proven in Theorem 2.1 of Blanchet et al. (2024).

Next, construct two discrete measure $\mu_0, \mu_1$ supported on $\mathcal{F}$ whose p.m.f. has the following form:

$$\mu_k \left( \left\{ (B\bar{b})^{\frac{1}{q}} \cdot (M + \sum_{j=1}^{m^d} \eta_j f_j) \right\} \right) = \prod_{j=1}^{m^d} \mathbb{P}(w_j^{(k)} = \eta_j), \ k \in \{0,1\}.$$

Where $\{w_j^{(0)}\}_{j=1}^{m^d}$ and $\{w_j^{(1)}\}_{j=1}^{m^d}$ are independent and identical copies of $w_{\frac{1+\kappa}{2}}$ and $w_{\frac{1-\kappa}{2}}$ respectively, we take $\kappa = \frac{1}{3}\sqrt{\frac{2}{3n}}$. Here $w_p$ denote Radmacher random variable satisfying $\mathbb{P}(w_p = -1) = p$ and $\mathbb{P}(w_p = 1) = 1 - p$.

In order to determine the separation between two priors $\mu_0$ and $\mu_1$, we need to derive the concentration inequality of each prior first. Define $A := \int_{\Omega_j} B\bar{b} \cdot (M + f_j(\boldsymbol{x}))^q d\boldsymbol{x}$ and $C := \int_{\Omega_j} B\bar{b} \cdot (M - f_j(\boldsymbol{x}))^q d\boldsymbol{x}$, we may derive the lower bound on the quantity $\Delta' := A - C > 0$:

$$\Delta' = \int_{\Omega_j} B\bar{b} \cdot (M + f_j(\boldsymbol{x}))^q d\boldsymbol{x} - \int_{\Omega_j} B\bar{b} \cdot (M - f_j(\boldsymbol{x}))^q d\boldsymbol{x}$$

$$= B\bar{b} \cdot \int_{\Omega_j} \left( \int_{-f_j(\boldsymbol{x})}^{f_j(\boldsymbol{x})} q(M+y)^{q-1} dy \right) d\boldsymbol{x} \geq B\bar{b} \cdot \int_{\Omega_j} \left( \int_{-f_j(\boldsymbol{x})}^{f_j(\boldsymbol{x})} q(\frac{1}{2}M)^{q-1} dy \right) d\boldsymbol{x}$$

$$= B\bar{b} \cdot \frac{q}{2^{q-1}} M^{q-1} \int_{\Omega_j} \left( 2f_j(\boldsymbol{x}) \right) d\boldsymbol{x} \gtrsim B\bar{b} \cdot \int_{\Omega_j} f_j(\boldsymbol{x}) d\boldsymbol{x}$$

$$= B\bar{b} \cdot \int_{\Omega_j} m^{-s} K(m(x - c_j)) d\boldsymbol{x} = B\bar{b} \cdot m^{-s-d} \|K\|_{\mathcal{L}^1([-\frac{1}{2}, \frac{1}{2}]^d)}. \tag{14}$$

Moreover, let's set $\lambda = \frac{1}{2}$ and apply Hoeffding's Inequality to the bounded random variables $\{w_j^{(0)}\}_{j=1}^{m^d}$ and $\{w_j^{(1)}\}_{j=1}^{m^d}$ to deduce that:

$$\mathbb{P}\left( \sum_{j=1}^{m^d} w_j^{(0)} \geq -(1-\lambda)m^d \kappa \right) \leq \exp\left( -\frac{2(\lambda m^d \kappa)^2}{4m^d} \right) = \exp\left( -\frac{1}{2}\lambda^2 \kappa^2 m^d \right),$$

$$\mathbb{P}\left( \sum_{j=1}^{m^d} w_j^{(1)} \leq (1-\lambda)m^d \kappa \right) \leq \exp\left( -\frac{2(\lambda m^d \kappa)^2}{4m^d} \right) = \exp\left( -\frac{1}{2}\lambda^2 \kappa^2 m^d \right).$$

By picking $c := \frac{m^d}{2}(A + C)$, $\Delta := (1 - \lambda)\kappa m^d \Delta'$ and $\beta_0 = \beta_1 = \exp(-\frac{1}{2}\lambda^2\kappa^2 m^d)$, we may deduce that:

$$\mu_0(f \in \mathcal{W}^{s,p}(\Omega) : I_f^q \leq c - \Delta)$$

$$= \mathbb{P}\Big( \sum_{j=1}^{m^d} I^q_{(B\bar{b})^{\frac{1}{q}}(M+w_j^{(0)}f_j)} \leq \frac{1 - (1 - \lambda)\kappa}{2}m^d A + \frac{1 + (1 - \lambda)\kappa}{2}m^d C \Big)$$

$$\geq \mathbb{P}\Big( \sum_{j=1}^{m^d} w_j^{(0)} \leq -(1 - \lambda)m^d\kappa \Big) = 1 - \mathbb{P}\Big( \sum_{j=1}^{m^d} w_j^{(0)} \geq -(1 - \lambda)m^d\kappa \Big)$$

$$\geq 1 - \exp\left( -\frac{1}{2}\lambda^2\kappa^2 m^d \right) = 1 - \beta_0,$$

$$\mu_1(f \in \mathcal{W}^{s,p}(\Omega) : I_f^q \geq c + \Delta)$$

$$= \mathbb{P}\Big( \sum_{j=1}^{m^d} I^q_{(B\bar{b})^{\frac{1}{q}}(M+w_j^{(1)}f_j)} \geq \frac{1 + (1 - \lambda)\kappa}{2}m^d A + \frac{1 - (1 - \lambda)\kappa}{2}m^d C \Big)$$

$$\geq \mathbb{P}\Big( \sum_{j=1}^{m^d} w_j^{(1)} \geq (1 - \lambda)m^d\kappa \Big) = 1 - \mathbb{P}\Big( \sum_{j=1}^{m^d} w_j^{(0)} \leq (1 - \lambda)m^d\kappa \Big)$$

$$\geq 1 - \exp\left( -\frac{1}{2}\lambda^2\kappa^2 m^d \right) = 1 - \beta_1,$$

We proceed to compute the mixture distribution under $\mu_0$ and $\mu_1$:

$$\mathbb{P}_0(\vec{x}, \vec{y}) = \mathbb{P}(\vec{x}) \cdot \mathbb{P}_0(\vec{y} \mid \vec{x}) = \mathbb{P}(\vec{x}) \cdot \int \mathbb{P}(\vec{y} \mid \vec{x}, \theta)\mu_0(d\theta)$$

$$= \Big( \prod_{i=1}^n p^\circ(\boldsymbol{x}_i) \Big) \cdot \prod_{j=1}^{m^d} \Big( \frac{1+\kappa}{2} \prod_{i:\boldsymbol{x}_i \in \Omega_j} \delta_{(B\bar{b})^{\frac{1}{q}}(M-f_j(\boldsymbol{x}_i))}(y_i) + \frac{1-\kappa}{2} \prod_{i:\boldsymbol{x}_i \in \Omega_j} \delta_{(B\bar{b})^{\frac{1}{q}}(M+f_j(\boldsymbol{x}_i))}(y_i) \Big),$$

$$\mathbb{P}_1(\vec{x}, \vec{y}) = \mathbb{P}(\vec{x}) \cdot \mathbb{P}_1(\vec{y} \mid \vec{x}) = \mathbb{P}(\vec{x}) \cdot \int \mathbb{P}(\vec{y} \mid \vec{x}, \theta)\mu_1(d\theta)$$

$$= \Big( \prod_{i=1}^n p^\circ(\boldsymbol{x}_i) \Big) \cdot \prod_{j=1}^{m^d} \Big( \frac{1-\kappa}{2} \prod_{i:\boldsymbol{x}_i \in \Omega_j} \delta_{(B\bar{b})^{\frac{1}{q}}(M-f_j(\boldsymbol{x}_i))}(y_i) + \frac{1+\kappa}{2} \prod_{i:\boldsymbol{x}_i \in \Omega_j} \delta_{(B\bar{b})^{\frac{1}{q}}(M+f_j(\boldsymbol{x}_i))}(y_i) \Big).$$

Denote $\mathcal{J}_n$ as the set of all indices $j$ satisfying that $\Omega_j$ contains at least one of the points in $\{\boldsymbol{x}_i\}_{i=1}^n$:

$$\mathcal{J}_n := \mathcal{J}_n(\boldsymbol{x}_1, \cdots, \boldsymbol{x}_n) = \big\{ j : 1 \leq j \leq m^d \text{ and } \Omega_j \cap \{\boldsymbol{x}_1, \cdots, \boldsymbol{x}_n\} \neq \varnothing \big\}.$$

Since $m^d = 200n > n$, we have $|\mathcal{J}_n| \leq n$. Thus, KL divergence can be upper bounded by:

$$KL(\mathbb{P}_0\|\mathbb{P}_1) = \mathbb{E}\left[\log\left(\frac{\mathbb{P}_0(\vec{\boldsymbol{x}},\vec{y})}{\mathbb{P}_1(\vec{\boldsymbol{x}},\vec{y})}\right)\mathbb{P}_0(\vec{\boldsymbol{x}},\vec{y})\right] = \mathbb{E}_{\vec{\boldsymbol{x}}}\mathbb{E}_{\vec{y}|\vec{\boldsymbol{x}}}\left[\log\left(\frac{\mathbb{P}_0(\vec{\boldsymbol{x}},\vec{y})}{\mathbb{P}_1(\vec{\boldsymbol{x}},\vec{y})}\right)\mathbb{P}_0(\vec{\boldsymbol{x}},\vec{y})\right]$$

$$= \mathbb{E}_{\vec{\boldsymbol{x}}}\Bigg[\int_{-\infty}^{\infty}\cdots\int_{-\infty}^{\infty}\log\Big(\prod_{j=1}^{m^d}\frac{\frac{1+\kappa}{2}\prod_{i:\boldsymbol{x}_i\in\Omega_j}\delta_{(B\bar{b})^{\frac{1}{q}}(M-f_j(\boldsymbol{x}_i))}(y_i) + \frac{1-\kappa}{2}\prod_{i:\boldsymbol{x}_i\in\Omega_j}\delta_{(B\bar{b})^{\frac{1}{q}}(M+f_j(\boldsymbol{x}_i))}(y_i)}{\frac{1-\kappa}{2}\prod_{i:\boldsymbol{x}_i\in\Omega_j}\delta_{(B\bar{b})^{\frac{1}{q}}(M-f_j(\boldsymbol{x}_i))}(y_i) + \frac{1+\kappa}{2}\prod_{i:\boldsymbol{x}_i\in\Omega_j}\delta_{(B\bar{b})^{\frac{1}{q}}(M+f_j(\boldsymbol{x}_i))}(y_i)}\Big)$$

$$\cdot\prod_{j=1}^{m^d}\left(\frac{1+\kappa}{2}\prod_{i:\boldsymbol{x}_i\in\Omega_j}\delta_{(B\bar{b})^{\frac{1}{q}}(M-f_j(\boldsymbol{x}_i))}(y_i) + \frac{1-\kappa}{2}\prod_{i:\boldsymbol{x}_i\in\Omega_j}\delta_{M+f_j(\boldsymbol{x}_i)}(y_i)\right)\prod_{i=1}^{n}dy_i\Bigg]$$

$$= \mathbb{E}_{\vec{\boldsymbol{x}}}\Bigg[\sum_{j\in\mathcal{J}_n}\int_{-\infty}^{\infty}\cdots\int_{-\infty}^{\infty}\log\Big(\frac{\frac{1+\kappa}{2}\prod_{i:\boldsymbol{x}_i\in\Omega_j}\delta_{(B\bar{b})^{\frac{1}{q}}(M-f_j(\boldsymbol{x}_i))}(y_i) + \frac{1-\kappa}{2}\prod_{i:\boldsymbol{x}_i\in\Omega_j}\delta_{(B\bar{b})^{\frac{1}{q}}(M+f_j(\boldsymbol{x}_i))}(y_i)}{\frac{1-\kappa}{2}\prod_{i:\boldsymbol{x}_i\in\Omega_j}\delta_{(B\bar{b})^{\frac{1}{q}}(M-f_j(\boldsymbol{x}_i))}(y_i) + \frac{1+\kappa}{2}\prod_{i:\boldsymbol{x}_i\in\Omega_j}\delta_{(B\bar{b})^{\frac{1}{q}}(M+f_j(\boldsymbol{x}_i))}(y_i)}\Big)$$

$$\cdot\left(\frac{1+\kappa}{2}\prod_{i:\boldsymbol{x}_i\in\Omega_j}\delta_{(B\bar{b})^{\frac{1}{q}}(M-f_j(\boldsymbol{x}_i))}(y_i) + \frac{1-\kappa}{2}\prod_{i:\boldsymbol{x}_i\in\Omega_j}\delta_{(B\bar{b})^{\frac{1}{q}}(M+f_j(\boldsymbol{x}_i))}(y_i)\right)\prod_{i:\boldsymbol{x}_i\in\Omega_j}dy_i\Bigg]$$

$$= \mathbb{E}_{\vec{\boldsymbol{x}}}\left[|\mathcal{J}_n|\left(\log\left(\frac{1+\kappa}{1-\kappa}\right)\frac{1+\kappa}{2} + \log\left(\frac{1-\kappa}{1+\kappa}\right)\frac{1-\kappa}{2}\right)\right]$$

$$\leq n\kappa\log\left(\frac{1+\kappa}{1-\kappa}\right).$$

Use the Pinkser's inequality to upper bound TV distance between $\mathbb{P}_0$ and $\mathbb{P}_1$ further:

$$TV(\mathbb{P}_0\|\mathbb{P}_1) \leq \sqrt{\frac{1}{2}KL(\mathbb{P}_0\|\mathbb{P}_1)} \leq \sqrt{\frac{n\kappa}{2}\log\left(\frac{1+\kappa}{1-\kappa}\right)} \leq \sqrt{\frac{3n}{2}}\kappa = \frac{1}{3}. \tag{15}$$

Finally, let $\Delta = (1-\lambda)\kappa m^d\Delta'$ and $\beta_0 = \beta_1 = \exp(-\frac{1}{2}\lambda^2\kappa^2 m^d) = \exp(-\frac{50}{27}) < \frac{1}{6}$. Combining 14 and 15, we obtain the final lower bound:

$$\inf_{\hat{H}^q\in\mathcal{H}_n^{f,q}}\sup_{(f,p^\circ,p^*)\in\mathcal{V}}\mathbb{E}_{\{x_i\}_{i=1}^n,\{y_i\}_{i=1}^n}\left[\left|\hat{H}^q\left(\{x_i\}_{i=1}^n,\{y_i\}_{i=1}^n\right) - I_f^q\right|\right]$$

$$\geq \Delta\inf_{\hat{H}^q\in\mathcal{H}_n^{f,q}}\sup_{(f,p^\circ,p^*)\in\mathcal{V}}\mathbb{P}_{\{x_i\}_{i=1}^n,\{y_i\}_{i=1}^n}\left[\left|\hat{H}^q\left(\{x_i\}_{i=1}^n,\{y_i\}_{i=1}^n\right) - I_f^q\right| \geq \Delta\right]$$

$$\geq (1-\lambda)\kappa m^d\Delta' I_{g_1}^q\frac{1 - TV(\mathbb{P}_0\|\mathbb{P}_1) - \beta_0 - \beta_1}{2}$$

$$\gtrsim B\cdot\bar{b}\cdot n^{-\frac{s}{d}-\frac{1}{2}}. \tag{16}$$

Combing the two lower bounds established in 13 and 16 completes the proof of Theorem 1. $\qquad\square$

### A.2 Proof of Theorem 2

**Theorem 2** (Upper Bound on Moment Estimation with Smoothness). *Assume that $p \geq 2$, $q \leq p \leq 2q$ and $\frac{s}{d} > \frac{1}{p} - \frac{1}{2q}$. Let $(f,p^\circ,p^*) \in \mathcal{V}(\underline{b},\bar{b},B,s,p)$ and we have sample $\mathcal{S} = \{(\boldsymbol{x}_i,y_i = f(\boldsymbol{x}_i))\}_{i=1}^n$. If $\hat{f}_{\mathcal{S}_1}$ satisfies the Assumption 1, then the estimator $\hat{H}_B^q$ constructed in equation 3 above satisfies*

$$\mathbb{E}_{\mathcal{S}}\left[\left|\hat{H}_B^q(\mathcal{S}) - I_f^q\right|\right] \lesssim B\cdot\bar{b}\cdot n^{\max\{-q(\frac{s}{d}-\frac{1}{p})-1,-\frac{1}{2}-\frac{s}{d}\}}. \tag{5}$$

We begin with the key lemma used in the proof.

**Lemma 2** (Sobolev Embedding Theorem). *For some fixed dimension $d \in \mathbb{N}$, we have that:*

*(I) For any $s,t \in \mathbb{N}_0$ and $p,q \in \mathbb{R}$ satisfying $s > t, p < d$ and $1 \leq p < q \leq \infty$, we have $\mathcal{W}^{s,p}(\mathbb{R}^d) \subseteq \mathcal{W}^{t,q}(\mathbb{R}^d)$ when the relation $\frac{1}{p} - \frac{s}{d} = \frac{1}{q} - \frac{t}{d}$ holds. In the special case when $t = 0$, we have $\mathcal{W}^{t,q}(\mathbb{R}^d) \subseteq \mathcal{L}^q(\mathbb{R}^d)$ for any $s \in \mathbb{N}$ and $p,q \in \mathbb{R}$ satisfying $1 \leq p < q \leq \infty$ and $\frac{1}{p} - \frac{s}{d} \leq \frac{1}{q}$.*

*(II) For any $\alpha \in (0,1)$, let $\beta = \frac{d}{1-\alpha} \in (d,\infty]$. Then we have $\mathcal{C}^1(\mathbb{R}^d) \cap \mathcal{W}^{1,\beta}(\mathbb{R}^d) \subseteq \mathcal{C}^\alpha(\mathbb{R}^d)$.*

*Proof.* Firstly, decompose the mean square error $\mathbb{E}_{\mathcal{S}}\left[\left|\hat{H}_B^q(\mathcal{S}) - I_f^q\right|^2\right]$ into bias part $\left|\mathbb{E}_{\mathcal{S}}[\hat{H}_B^q(\mathcal{S})] - I_f^q\right|^2$ and variance part $\mathbb{E}_{\mathcal{S}}\left[\left|\hat{H}_B^q(\mathcal{S}) - \mathbb{E}_{\mathcal{S}}[\hat{H}_B^q(\mathcal{S})]\right|^2\right]$.

Estimator $\hat{H}_B^q(\mathcal{S})$ is unbias for $I_f^q$, so the bias part is zero:

$$
\begin{aligned}
\mathbb{E}_{\mathcal{S}}\left[\hat{H}_B^q(\mathcal{S})\right] &= \mathbb{E}_{\boldsymbol{X}\sim\mathbb{P}^*}\left[\hat{f}_{\mathcal{S}_1}^q(\boldsymbol{X})\right] + \frac{2}{n}\sum_{(\boldsymbol{x}_i,y_i)\in\mathcal{S}_2}\mathbb{E}\left[w(\boldsymbol{x}_i)\cdot(y_i^q - \hat{f}_{\mathcal{S}_1}^q(\boldsymbol{x}_i))\right] \\
&= \mathbb{E}_{\boldsymbol{X}\sim\mathbb{P}^*}\left[\hat{f}_{\mathcal{S}_1}^q(\boldsymbol{X})\right] + \int_\Omega \frac{p^*(\boldsymbol{x})}{p^\circ(\boldsymbol{x})}\cdot\left[f(\boldsymbol{x})^q - \hat{f}_{\mathcal{S}_1}^q(\boldsymbol{x})\right]\cdot p^\circ(\boldsymbol{x})d\boldsymbol{x} \\
&= \mathbb{E}_{\boldsymbol{X}\sim\mathbb{P}^*}\left[\hat{f}_{\mathcal{S}_1}^q(\boldsymbol{X})\right] + \mathbb{E}_{\boldsymbol{X}\sim\mathbb{P}^*}\left[f^q(\boldsymbol{X}) - \hat{f}_{\mathcal{S}_1}^q(\boldsymbol{X})\right] \\
&= \mathbb{E}_{\boldsymbol{X}\sim\mathbb{P}^*}[f^q(\boldsymbol{X})] = I_f^q.
\end{aligned}
$$

Using the law of total variance to decompose the variance part further:

$$
\operatorname{Var}\left[\hat{H}_B^q(\mathcal{S})\right] = \mathbb{E}\left[\operatorname{Var}\left(\hat{H}_B^q(\mathcal{S}) \mid \mathcal{S}_1\right)\right] + \operatorname{Var}\left[\mathbb{E}\left(\hat{H}_B^q(\mathcal{S}) \mid \mathcal{S}_1\right)\right].
$$

Since $\mathbb{E}\left(\hat{H}_B^q(\mathcal{S}) \mid \mathcal{S}_1\right) = I_f^q$, the second part is zero. So we only need to compute the first part.

$$
\begin{aligned}
\mathbb{E}\left[\operatorname{Var}\left(\hat{H}_B^q(\mathcal{S}) \mid \mathcal{S}_1\right)\right] &= \mathbb{E}\left[\operatorname{Var}\left(\frac{2}{n}\sum_{(\boldsymbol{x}_i,y_i)\in\mathcal{S}_2}w(\boldsymbol{x}_i)\cdot(y_i^q - \hat{f}_{\mathcal{S}_1}^q(\boldsymbol{x}_i)) \mid \mathcal{S}_1\right)\right] \\
&= \frac{2}{n}\mathbb{E}\left[\operatorname{Var}\left(w(\boldsymbol{X})\cdot(f^q(\boldsymbol{X}) - \hat{f}_{\mathcal{S}_1}^q(\boldsymbol{X})) \mid \mathcal{S}_1\right)\right] \\
&\leq \frac{2}{n}\mathbb{E}_{\mathcal{S}_1}\left[\mathbb{E}_{\mathcal{S}_2|\mathcal{S}_1}\left(w(\boldsymbol{X})\cdot(f^q(\boldsymbol{X}) - \hat{f}_{\mathcal{S}_1}^q(\boldsymbol{X}))\right)^2\right] \\
&\leq \frac{2}{n}B^2\cdot\mathbb{E}_{\mathcal{S}_1}\left[\mathbb{E}_{\mathcal{S}_2|\mathcal{S}_1}\left(f^q(\boldsymbol{X}) - \hat{f}_{\mathcal{S}_1}^q(\boldsymbol{X})\right)^2\right]. \quad (17)
\end{aligned}
$$

Following the proof in Blanchet et al. (2024), we further upper bound the expression $\mathbb{E}_{\mathcal{S}_1}\left[\mathbb{E}_{\mathcal{S}_2|\mathcal{S}_1}\left(f^q(\boldsymbol{X}) - \hat{f}_{\mathcal{S}_1}^q(\boldsymbol{X})\right)^2\right]$. Let $g_{\mathcal{S}_1} := \hat{f}_{\mathcal{S}_1} - f$ represent the difference between the estimator $\hat{f}_{\mathcal{S}_1}$ and underlying function $f$.

$$
\begin{aligned}
\mathbb{E}_{\mathcal{S}_1}\left[\mathbb{E}_{\mathcal{S}_2|\mathcal{S}_1}\left(f^q(\boldsymbol{X}) - \hat{f}_{\mathcal{S}_1}^q(\boldsymbol{X})\right)^2\right] &\leq \bar{b}\cdot\mathbb{E}_{\mathcal{S}_1}\left[\int_\Omega\left(f^q(\boldsymbol{x}) - \hat{f}_{\mathcal{S}_1}^q(\boldsymbol{x})\right)^2 d\boldsymbol{x}\right] \\
&= \bar{b}\cdot\mathbb{E}_{\mathcal{S}_1}\left[\int_\Omega\left((f(\boldsymbol{x}) + g_{\mathcal{S}_1}(\boldsymbol{x}))^q - f^q(\boldsymbol{x})^q\right)^2 d\boldsymbol{x}\right] \\
&= \bar{b}\cdot\mathbb{E}_{\mathcal{S}_1}\left[\int_\Omega\left(\int_0^{g_{\mathcal{S}_1}(\boldsymbol{x})}q(f(\boldsymbol{x}) + y)^{q-1}dy\right)^2 d\boldsymbol{x}\right] \\
&\leq \bar{b}\cdot\mathbb{E}_{\mathcal{S}_1}\left[\int_\Omega\left|\int_0^{g_{\mathcal{S}_1}(\boldsymbol{x})}1dy\right|\cdot\left|\int_0^{g_{\mathcal{S}_1}(\boldsymbol{x})}q^2(f(\boldsymbol{x}) + y)^{2q-2}dy\right|d\boldsymbol{x}\right] \quad \text{(Hölder's Inequality)} \\
&\lesssim \bar{b}\cdot\mathbb{E}_{\mathcal{S}_1}\left[\int_\Omega|g_{\mathcal{S}_1}(\boldsymbol{x})|\cdot|g_{\mathcal{S}_1}(\boldsymbol{x})|\max\left\{\left|f^{2q-2}(\boldsymbol{x})\right|,\left|g_{\mathcal{S}_1}^{2q-2}(\boldsymbol{x})\right|\right\}d\boldsymbol{x}\right] \\
&\lesssim \bar{b}\cdot\mathbb{E}_{\mathcal{S}_1}\left[\int_\Omega\left|g_{\mathcal{S}_1}^{2q}(\boldsymbol{x})\right|d\boldsymbol{x}\right] + \bar{b}\cdot\mathbb{E}_{\mathcal{S}_1}\left[\int_\Omega\left|g_{\mathcal{S}_1}^2(\boldsymbol{x})f^{2q-2}(\boldsymbol{x})\right|d\boldsymbol{x}\right]. \quad (18)
\end{aligned}
$$

Let's further upper bound the two terms in 18. For the first term, since $\frac{s}{d} > \frac{1}{p} - \frac{1}{2q}$, we may apply equation 4 from Assumption 1 by choosing $r = 2q$. This leads to the following deduction:

$$
\begin{aligned}
\mathbb{E}_{\mathcal{S}_1} \left[ \int_\Omega \left| g_{\mathcal{S}_1}^{2q}(\boldsymbol{x}) \right| d\boldsymbol{x} \right] &= \mathbb{E}_{\mathcal{S}_1} \left[ \| \hat{f}_{\mathcal{S}_1} - f \|_{\mathcal{L}^{2q}(\Omega)}^{2q} \right] \\
&\lesssim \bar{b} \cdot ((\tfrac{n}{2})^{-\frac{s}{d} + (\frac{1}{p} - \frac{1}{2q})_+})^{2q} \lesssim \bar{b} \cdot n^{2q(-\frac{s}{d} + \frac{1}{p} - \frac{1}{2q})} = \bar{b} \cdot n^{2q(\frac{1}{p} - \frac{s}{d}) - 1} \quad \text{(since } p \le 2q \text{).} \quad (19)
\end{aligned}
$$

Now, we proceed to upper bound the second term in equation 18. Here we define $p_\diamond := (\max\{\frac{1}{p} - \frac{s}{d}, 0\})^{-1}$, that is, $p_\diamond = \frac{pd}{d-sp}$ when $s < \frac{d}{p}$ and $p_\diamond = \infty$ otherwise. By the Sobolev Embedding Theorem, we have $\mathcal{W}^{s,p}(\Omega) \subseteq \mathcal{L}^{p_\diamond}(\Omega)$. There are three distinct cases to consider based on the smoothness parameter $s$.

(Case I) When $s \in (\frac{d}{p}, \infty)$, we have $p_\diamond = \infty$ which implies that $f \in \mathcal{W}^{s,p}(\Omega) \subseteq \mathcal{L}^\infty(\Omega)$. Similarly, by Assumption 1, $\hat{f}_{\mathcal{S}_1}$ is also in the Sobolev space $\mathcal{W}^{s,p}(\Omega) \subseteq \mathcal{L}^\infty(\Omega)$. Therefore, the difference $g_{\mathcal{S}_1} = \hat{f}_{\mathcal{S}_1} - f \in \mathcal{W}^{s,p}(\Omega) \subseteq \mathcal{L}^\infty(\Omega) \subseteq \mathcal{L}^2(\Omega)$. By picking $r = 2$ in 4 of Assumption 1, and utilizing the facts that $p \ge 2$ and $f \in \mathcal{L}^\infty(\Omega)$, we can deduce that:

$$
\begin{aligned}
\mathbb{E}_{\mathcal{S}_1} \left[ \int_\Omega \left| g_{\mathcal{S}_1}^2(\boldsymbol{x}) f^{2q-2}(\boldsymbol{x}) \right| d\boldsymbol{x} \right] &\lesssim \mathbb{E}_{\mathcal{S}_1} \left[ \int_\Omega \left| g_{\mathcal{S}_1}^2(\boldsymbol{x}) \right| d\boldsymbol{x} \right] \quad \text{(Hölder's Inequality)} \\
= \mathbb{E}_{\mathcal{S}_1} \left[ \| \hat{f}_{\mathcal{S}_1} - f \|_{\mathcal{L}^2(\Omega)}^2 \right] &\lesssim \bar{b} \cdot \left( n^{-\frac{s}{d} + (\frac{1}{p} - \frac{1}{2})_+} \right)^2 = \bar{b} \cdot n^{-\frac{2s}{d}}. \quad (20)
\end{aligned}
$$

(Case II) When $s \in (\frac{d(2q-p)}{p(2q-2)}, \frac{d}{p})$, it follows that

$$
p_\diamond = \frac{pd}{d-sp} > \frac{pd}{d - p\frac{d(2q-p)}{p(2q-2)}} = \frac{p(2q-2)}{p-2},
$$

which implies $f \in \mathcal{W}^{s,p}(\Omega) \subseteq \mathcal{L}^{p_\diamond}(\Omega) \subseteq \mathcal{L}^{\frac{p(2q-2)}{p-2}}(\Omega) \subseteq \mathcal{L}^p(\Omega)$. Since $\frac{p}{p-2} > 1$, it follows that $f^{2q-2} \in \mathcal{L}^{\frac{p}{p-2}}(\Omega)$. Moreover, since $\hat{f}_{\mathcal{S}_1} \in \mathcal{W}^{s,p}(\Omega) \subseteq \mathcal{L}^p(\Omega)$, we obtain $g_{\mathcal{S}_1} = \hat{f}_{\mathcal{S}_1} - f \in \mathcal{L}^p(\Omega)$. Given that $p \ge 2$, we can further deduce that $g_{\mathcal{S}_1}^2 \in \mathcal{L}^{\frac{p}{2}}(\Omega)$. We may then apply Hölder's Inequality to obtain:

$$
\begin{aligned}
\mathbb{E}_{\mathcal{S}_1} \left[ \int_\Omega \left| g_{\mathcal{S}_1}^2(\boldsymbol{x}) f^{2q-2}(\boldsymbol{x}) \right| d\boldsymbol{x} \right] &= \mathbb{E}_{\mathcal{S}_1} \left[ \| g_{\mathcal{S}_1}^2(\boldsymbol{x}) f^{2q-2}(\boldsymbol{x}) \|_{\mathcal{L}^1(\Omega)} \right] \\
\le \mathbb{E}_{\mathcal{S}_1} \left[ \| g_{\mathcal{S}_1}^2(\boldsymbol{x}) \|_{\mathcal{L}^{\frac{p}{2}}(\Omega)} \| f^{2q-2}(\boldsymbol{x}) \|_{\mathcal{L}^{\frac{p}{p-2}}(\Omega)} \right] &\quad \text{(Hölder's Inequality)} \\
\le \| f(\boldsymbol{x}) \|_{\mathcal{L}^{\frac{p(2q-2)}{p-2}}(\Omega)}^{2q-2} &\cdot \mathbb{E}_{\mathcal{S}_1} \left[ \| g_{\mathcal{S}_1}(\boldsymbol{x}) \|_{\mathcal{L}^p(\Omega)}^2 \right].
\end{aligned}
$$

Note that the function $h(t) = t^{\frac{2}{p}}$ is concave and $\frac{1}{p} \in (\frac{d-sp}{pd}, 1]$ when $p \ge 2$. Thus, by applying Jensen's inequality and choosing $r = p$ in 4 of Assumption 1, we can upper bound the last term as follows:

$$
\begin{aligned}
\mathbb{E}_{\mathcal{S}_1} \left[ \| g_{\mathcal{S}_1}(\boldsymbol{x}) \|_{\mathcal{L}^p(\Omega)}^2 \right] &= \mathbb{E}_{\mathcal{S}_1} \left[ \left( \| g_{\mathcal{S}_1}(\boldsymbol{x}) \|_{\mathcal{L}^{\frac{p}{2}}(\Omega)} \right)^{\frac{2}{p}} \right] \\
\le \mathbb{E}_{\mathcal{S}_1} \left[ \| g_{\mathcal{S}_1}(\boldsymbol{x}) \|_{\mathcal{L}^p(\Omega)}^p \right]^{\frac{2}{p}} &= \mathbb{E}_{\mathcal{S}_1} \left[ \| \hat{f}_{\mathcal{S}_1} \boldsymbol{x} - f(\boldsymbol{x}) \|_{\mathcal{L}^p(\Omega)}^p \right]^{\frac{2}{p}} \quad \text{(Jensen's inequality)} \\
\lesssim \bar{b} \cdot \left( (\tfrac{n}{2})^{-\frac{s}{d} + (\frac{1}{p} - \frac{1}{p})_+} \right)^2 &\lesssim \bar{b} \cdot n^{-\frac{2s}{d}}.
\end{aligned}
$$

This gives us the final upper bound under the assumption that $s \in (\frac{d(2q-p)}{p(2q-2)}, \frac{d}{p})$ as follows:

$$
\mathbb{E}_{\mathcal{S}_1} \left[ \int_\Omega \left| g_{\mathcal{S}_1}^2(\boldsymbol{x}) f^{2q-2}(\boldsymbol{x}) \right| d\boldsymbol{x} \right] \lesssim \bar{b} \cdot n^{-\frac{2s}{d}}. \quad (21)
$$

(Case III) When $s \in (\frac{d(2q-p)}{2pq}, \frac{d(2q-p)}{p(2q-2)})$, we have $s < \frac{d}{p}$, which implies that $p_\diamond = \frac{pd}{d-sp}$ satisfies $2q < p_\diamond < \frac{p(2q-2)}{p-2}$. Given that $p_\diamond > 2q > 2q - 2$ and $f \in \mathcal{W}^{s,p}(\Omega) \subseteq \mathcal{L}^{p_\diamond}(\Omega)$, it follows that $f^{2q-2} \in \mathcal{L}^{\frac{p_\diamond}{2q-2}}(\Omega)$. Moreover, since $p_\diamond > 2q$, we have $\frac{2p_\diamond}{p_\diamond+2-2q} < p_\diamond$. Additionally, because $p_\diamond < \frac{p(2q-2)}{p-2}$, it follows that $\frac{2p_\diamond}{p_\diamond+2-2q} > p$.

Given that both $\hat{f}_{\mathcal{S}_1}$ and $f$ are in the Sobolev space $\mathcal{W}^{s,p}(\Omega) \subseteq \mathcal{L}^{p_\diamond}(\Omega)$, it follows that $g_{\mathcal{S}_1} = \hat{f}_{\mathcal{S}_1} - f \in \mathcal{W}^{s,p}(\Omega) \subseteq \mathcal{L}^{p_\diamond}(\Omega) \subseteq \mathcal{L}^{\frac{2p_\diamond}{p_\diamond+2-2q}}(\Omega)$. Since $q \geq 1 \Rightarrow \frac{p_\diamond}{p_\diamond+2-2q} \geq 1$, we have $g_{\mathcal{S}_1}^2 \in \mathcal{L}^{\frac{p_\diamond}{p_\diamond+2-2q}}(\Omega)$. We can then apply Hölder's Inequality to obtain:

$$\mathbb{E}_{\mathcal{S}_1} \left[ \int_\Omega \left| g_{\mathcal{S}_1}^2(\boldsymbol{x}) f^{2q-2}(\boldsymbol{x}) \right| d\boldsymbol{x} \right] = \mathbb{E}_{\mathcal{S}_1} \left[ \| g_{\mathcal{S}_1}^2(\boldsymbol{x}) f^{2q-2}(\boldsymbol{x}) \|_{\mathcal{L}^1(\Omega)} \right]$$

$$\leq \mathbb{E}_{\mathcal{S}_1} \left[ \| g_{\mathcal{S}_1}^2(\boldsymbol{x}) \|_{\mathcal{L}^{\frac{2p_\diamond}{p_\diamond+2-2q}}(\Omega)} \| f^{2q-2}(\boldsymbol{x}) \|_{\mathcal{L}^{\frac{p_\diamond}{2q-2}}(\Omega)} \right] \qquad \text{(Hölder's Inequality)}$$

$$\leq \| f(\boldsymbol{x}) \|_{\mathcal{L}^{p_\diamond}(\Omega)}^{2q-2} \cdot \mathbb{E}_{\mathcal{S}_1} \left[ \| g_{\mathcal{S}_1}(\boldsymbol{x}) \|_{\mathcal{L}^{\frac{2p_\diamond}{p_\diamond+2-2q}}(\Omega)}^2 \right].$$

Note that the function $h(t) = t^{\frac{p_\diamond+2-2q}{p_\diamond}}$ is concave since $q \geq 1$. Furthermore, given the assumption $s \in (\frac{d(2q-p)}{2pq}, \frac{d(2q-p)}{p(2q-2)})$ we have $\frac{pd}{d-sp} > 2q$. This implies:

$$\frac{p_\diamond+2-2q}{2p_\diamond} = \frac{\frac{pd}{d-sp} + 2 - 2q}{2\frac{pd}{d-sp}} > \frac{2}{2\frac{pd}{d-sp}} = \frac{d-sp}{pd},$$

i.e., $\frac{p_\diamond+2-2q}{p_{[\diamond]}} \in (\frac{d-sp}{pd}, 1]$. Hence, we may apply Jensen's inequality and 4 in Assumption 1 to upper bound the last term:

$$\mathbb{E}_{\mathcal{S}_1} \left[ \| g_{\mathcal{S}_1}(\boldsymbol{x}) \|_{\mathcal{L}^{\frac{2p_\diamond}{p_\diamond+2-2q}}(\Omega)}^2 \right] = \mathbb{E}_{\mathcal{S}_1} \left[ \left( \| g_{\mathcal{S}_1}(\boldsymbol{x}) \|_{\mathcal{L}^{\frac{2p_\diamond}{p_\diamond+2-2q}}(\Omega)}^{\frac{2p_\diamond}{p_\diamond+2-2q}} \right)^{\frac{p_\diamond+2-2q}{p_\diamond}} \right]$$

$$\leq \mathbb{E}_{\mathcal{S}_1} \left[ \left( \| g_{\mathcal{S}_1}(\boldsymbol{x}) \|_{\mathcal{L}^{\frac{2p_\diamond}{p_\diamond+2-2q}}(\Omega)}^{\frac{2p_\diamond}{p_\diamond+2-2q}} \right) \right]^{\frac{p_\diamond+2-2q}{p_\diamond}} = \mathbb{E}_{\mathcal{S}_1} \left[ \left( \| \hat{f}_{\mathcal{S}_1}(\boldsymbol{x}) - f(\boldsymbol{x}) \|_{\mathcal{L}^{\frac{2p_\diamond}{p_\diamond+2-2q}}(\Omega)}^{\frac{2p_\diamond}{p_\diamond+2-2q}} \right) \right]^{\frac{p_\diamond+2-2q}{p_\diamond}}$$

$$\lesssim \bar{b} \cdot \left( (\frac{n}{2})^{-\frac{s}{d} + (\frac{1}{p} - \frac{p_\diamond+2-2q}{2p_\diamond})_+} \right)^2 \lesssim \bar{b} \cdot n^{-\frac{2s}{d} + 2(\frac{1}{p} - \frac{p_\diamond+2-2q}{2p_\diamond})_+}.$$

Recalled that we have proved that $p_\diamond \in (2q, \frac{p(2q-2)}{p-2})$, this implies that $p_\diamond(p-2) < p(2q-2) \Rightarrow 2p_\diamond > p(p_\diamond + 2 - 2q)$, i.e., $\frac{1}{p} > \frac{p_\diamond+2-2q}{2p_\diamond}$. Then we may simplify the last term above as follows:

$$-\frac{2s}{d} + 2\left( \frac{1}{p} - \frac{p_\diamond+2-2q}{2p_\diamond} \right)_+ = -\frac{2s}{d} + \frac{2}{p} - \left( 1 + \frac{2}{p_\diamond} - \frac{2q}{p_\diamond} \right) = \frac{2q}{p_\diamond} - 1 = 2q\left( \frac{1}{p} - \frac{s}{d} \right) - 1.$$

This gives us the final upper bound under the assumption that $s \in (\frac{d(2q-p)}{2pq}, \frac{d(2q-p)}{p(2q-2)})$ as follows:

$$\mathbb{E}_{\mathcal{S}_1} \left[ \int_\Omega \left| g_{\mathcal{S}_1}^2(\boldsymbol{x}) f^{2q-2}(\boldsymbol{x}) \right| d\boldsymbol{x} \right] \lesssim \bar{b} \cdot n^{2q(\frac{1}{p} - \frac{s}{d}) - 1}. \tag{22}$$

Combining the upper bounds derived in 19 20, 21 and 22 we may deduce that:

$$\mathbb{E}_{\mathcal{S}_1} \left[ \mathbb{E}_{\mathcal{S}_2 | \mathcal{S}_1} \left( f^q(\boldsymbol{X}) - \hat{f}_{\mathcal{S}_1}^q(\boldsymbol{X}) \right)^2 \right] \lesssim \bar{b}^2 \cdot \left[ n^{2q(\frac{1}{p} - \frac{s}{d}) - 1} + \max\{ n^{-\frac{2s}{d}}, n^{2q(\frac{1}{p} - \frac{s}{d}) - 1} \} \right]. \tag{23}$$

Finally, substituting 23 into 17 gives us the final upper bound:

$$\mathbb{E}_{\mathcal{S}} \left[ \left| \hat{H}_B^q(\mathcal{S}) - I_f^q \right| \right]$$

$$\leq \sqrt{\mathbb{E}_{\mathcal{S}} \left[ \left| \hat{H}_B^q(\mathcal{S}) - I_f^q \right|^2 \right]} = \sqrt{\mathbb{E} \left[ \mathrm{Var} \left( \hat{H}_B^q(\mathcal{S}) \mid \mathcal{S}_1 \right) \right]}$$

$$\lesssim B \cdot \bar{b} \cdot n^{\max\{ -q(\frac{s}{d} - \frac{1}{p}) - 1, -\frac{1}{2} - \frac{s}{d} \}}$$

$\square$

## A.3 PROOF OF THEOREM 3

**Theorem 3** (*$B$-independent Bound*). *Denote $\Omega_T^- := \{x : w(x) > T\}$ as the area where the likelihood ratio exceeds the threshold $T$, and let $\mathbb{P}(\Omega_T^-) = g(T)$ denote the probability of this event. Under the conditions in Theorem 2, the upper bound for the variance and bias of the truncated estimator 7 are given as follows:*

$$Bias = \left| \mathbb{E}(\hat{H}_T^q) - I_f^q \right| \le \bar{b} \cdot \mathbb{P}(\Omega_T^-),$$

$$Variance = \mathbb{E}\left[ \hat{H}_T^q - \mathbb{E}(\hat{H}_T^q) \right]^2 \le \bar{b}^2 \cdot T^2 \cdot r(n)^2 + \bar{b}^2 \cdot \mathbb{P}(\Omega_T^-)^2.$$

*If we further assume that $g(T) \le T^{-\alpha}$, and pick $T = \mathcal{O}(r(n)^{-\frac{1}{\alpha+1}})^6$, then we obtain the following $B$-independent bound:*

$$\mathbb{E}_{\mathcal{S}}\left[ \left| \hat{H}_T^q(\mathcal{S}) - I_f^q \right| \right] \lesssim \bar{b} \cdot r(n)^{\frac{\alpha}{\alpha+1}}, \tag{8}$$

*where $r(n) = n^{\max\{-q(\frac{s}{d} - \frac{1}{p}) - 1, -\frac{1}{2} - \frac{s}{d}\}}$ is the same order in Theorem 2.*

*Proof.* Decompose the mean square error $\mathbb{E}_{\mathcal{S}}\left[ \left| \hat{H}_T^q(\mathcal{S}) - I_f^q \right|^2 \right]$ into bias part $\left| \mathbb{E}_{\mathcal{S}}[\hat{H}_T^q(\mathcal{S})] - I_f^q \right|^2$ and variance part $\mathbb{E}_{\mathcal{S}}\left[ \left| \hat{H}_T^q(\mathcal{S}) - \mathbb{E}_{\mathcal{S}}[\hat{H}_T^q(\mathcal{S})] \right|^2 \right]$.

We first compute $\mathbb{E}\left[ \hat{H}_T^q(\mathcal{S}) \mid \mathcal{S}_1 \right]$. Recall that $\Omega_T^+ := \{x : w(x) \le T\}$ and $\Omega_T^- = \Omega / \Omega_T^+$, then we may deduce that:

$$\mathbb{E}\left[ \hat{H}_T^q(\mathcal{S}) \mid \mathcal{S}_1 \right] = \mathbb{E}_{\boldsymbol{X} \sim \mathbb{P}^*}\left[ \hat{f}_{\mathcal{S}_1}^q(\boldsymbol{X}) \right] + \int_\Omega \tau_T(w(\boldsymbol{x})) \cdot (f^q(\boldsymbol{x}) - \hat{f}_{\mathcal{S}_1}^q(\boldsymbol{x})) \cdot p^\circ(\boldsymbol{x}) d\boldsymbol{x}$$

$$= \mathbb{E}_{\boldsymbol{X} \sim \mathbb{P}^*}\left[ \hat{f}_{\mathcal{S}_1}^q(\boldsymbol{X}) \right] + \int_{\Omega_T^+} p^*(\boldsymbol{x}) \cdot (f^q(\boldsymbol{x}) - \hat{f}_{\mathcal{S}_1}^q(\boldsymbol{x})) d\boldsymbol{x} + \int_{\Omega_T^-} T \cdot p^\circ(\boldsymbol{x}) \cdot (f^q(\boldsymbol{x}) - \hat{f}_{\mathcal{S}_1}^q(\boldsymbol{x})) d\boldsymbol{x}$$

$$= \int_{\Omega_T^+} p^*(\boldsymbol{x}) \cdot f^q(\boldsymbol{x}) d\boldsymbol{x} + \int_{\Omega_T^-} p^*(\boldsymbol{x}) \cdot \hat{f}_{\mathcal{S}_1}^q(\boldsymbol{x}) d\boldsymbol{x} + \int_{\Omega_T^-} T \cdot p^\circ(\boldsymbol{x}) \cdot (f^q(\boldsymbol{x}) - \hat{f}_{\mathcal{S}_1}^q(\boldsymbol{x})) d\boldsymbol{x}. \tag{24}$$

We proceed to compute the bias term:

$$Bias = \left| \mathbb{E}\left[ \mathbb{E}[\hat{H}_T^q \mid \mathcal{S}_1] \right] - I_f^q \right|$$

$$= \left| \mathbb{E}_{\mathcal{S}_1}\left[ \int_{\Omega_T^-} p^*(\boldsymbol{x}) \hat{f}_{\mathcal{S}_1}^q(\boldsymbol{x}) d\boldsymbol{x} + \int_{\Omega_T^-} T p^\circ(\boldsymbol{x}) \cdot (f^q(\boldsymbol{x}) - \hat{f}_{\mathcal{S}_1}^q(\boldsymbol{x})) d\boldsymbol{x} - \int_{\Omega_T^-} p^*(\boldsymbol{x}) f^q(\boldsymbol{x}) d\boldsymbol{x} \right] \right|$$

$$= \mathbb{E}_{\mathcal{S}_1}\left[ \int_{\Omega_T^-} (p^*(\boldsymbol{x}) - T \cdot p^\circ(\boldsymbol{x})) \cdot (f^q(\boldsymbol{x}) - \hat{f}_{\mathcal{S}_1}^q(\boldsymbol{x})) d\boldsymbol{x} \right] \quad \text{(Since } w(\boldsymbol{x}) > T \text{ in } \Omega_T^-\text{)}$$

$$= \mathbb{E}_{\mathcal{S}_1}\left[ \int_{\Omega_T^-} p^\circ(\boldsymbol{x}) \cdot (w(\boldsymbol{x})) - T) \cdot (f^q(\boldsymbol{x}) - \hat{f}_{\mathcal{S}_1}^q(\boldsymbol{x})) d\boldsymbol{x} \right]$$

$$\le \mathbb{E}_{S_1}\left[ \int_{\Omega_T^-} p^\circ(\boldsymbol{x}) \cdot w(\boldsymbol{x}) \cdot (f^q(\boldsymbol{x}) - \hat{f}_{\mathcal{S}_1}^q(\boldsymbol{x})) d\boldsymbol{x} \right]$$

$$= \mathbb{E}_{\mathcal{S}_1}\left[ \int_{\Omega_T^-} p^*(\boldsymbol{x}) \cdot (f^q(\boldsymbol{x}) - \hat{f}_{\mathcal{S}_1}^q(\boldsymbol{x})) d\boldsymbol{x} \right]$$

$$\lesssim \mathbb{E}_{\mathcal{S}_1}\left[ \mathbb{P}(\Omega_T^-) \right] \quad \text{(Since } \hat{f}_{\mathcal{S}_1} \text{ and } f \text{ are in } \mathcal{L}^q(\mathbb{P}^*)\text{)}$$

$$\le \bar{b} \cdot \mathbb{P}(\Omega_T^-). \tag{25}$$

Using the law of total variance to decompose the variance part further:

$$\mathrm{Var}\left[ \hat{H}_T^q(\mathcal{S}) \right] = \mathbb{E}\left[ \mathrm{Var}\left( \hat{H}_T^q(\mathcal{S}) \mid \mathcal{S}_1 \right) \right] + \mathrm{Var}\left[ \mathbb{E}\left( \hat{H}_T^q(\mathcal{S}) \mid \mathcal{S}_1 \right) \right].$$

The first term can be upper bound by:

$$
\begin{aligned}
\mathbb{E}\left[\mathrm{Var}\left(\hat{H}_T^q(\mathcal{S}) \mid \mathcal{S}_1\right)\right] &= \frac{2}{n}\mathbb{E}\left[\mathrm{Var}\left(\tau_T(w(\boldsymbol{X})) \cdot (f^q(\boldsymbol{X}) - \hat{f}_{\mathcal{S}_1}^q(\boldsymbol{X})) \mid \mathcal{S}_1\right)\right] \\
&\le \frac{2}{n}\mathbb{E}_{\mathcal{S}_1}\left[\mathbb{E}_{\mathcal{S}_2 \mid \mathcal{S}_1}\left(\tau_T(w(\boldsymbol{X})) \cdot (f^q(\boldsymbol{X}) - \hat{f}_{\mathcal{S}_1}^q(\boldsymbol{X}))\right)^2\right] \\
&\le \frac{2}{n}T^2 \cdot \mathbb{E}_{\mathcal{S}_1}\left[\mathbb{E}_{\mathcal{S}_2 \mid \mathcal{S}_1}\left(f^q(\boldsymbol{X}) - \hat{f}_{\mathcal{S}_1}^q(\boldsymbol{X})\right)^2\right] \\
&\lesssim \left[T \cdot \bar{b} \cdot r(n)\right]^2.
\end{aligned}
\tag{26}
$$

Substitute equation 24 in the second term:

$$
\begin{aligned}
\mathrm{Var}\left[\mathbb{E}\left(\hat{H}_T^q(\mathcal{S}) \mid \mathcal{S}_1\right)\right] &= \mathrm{Var}\left[\int_{\Omega_T^-}(p^*(\boldsymbol{x}) - T \cdot p^\circ(\boldsymbol{x})) \cdot \hat{f}_{\mathcal{S}_1}^q(\boldsymbol{x})d\boldsymbol{x}\right] \\
&\le \mathbb{E}_{\mathcal{S}_1}\left[\int_{\Omega_T^-}(p^*(\boldsymbol{x}) - T \cdot p^\circ(\boldsymbol{x})) \cdot \hat{f}_{\mathcal{S}_1}^q(\boldsymbol{x})d\boldsymbol{x}\right]^2 \\
&\lesssim \mathbb{E}_{\mathcal{S}_1}\left[\mathbb{P}(\Omega_T^-)^2\right] \\
&\le \left[\bar{b} \cdot \mathbb{P}^\circ(\Omega_T^-)\right]^2.
\end{aligned}
\tag{27}
$$

The last term follows from the proof of Theorem 2, where $r(n) = n^{\max\{-q(\frac{s}{d} - \frac{1}{p}) - 1, -\frac{1}{2} - \frac{s}{d}\}}$.
Finally, combining the results 25, 26 and 27 above, we obtain the final upper bound:

$$
\begin{aligned}
\mathbb{E}_{\mathcal{S}}\left[\left|\hat{H}_T^q(\mathcal{S}) - I_f^q\right|\right] &\le \sqrt{\mathbb{E}_{\mathcal{S}}\left[\left|\hat{H}_T^q(\mathcal{S}) - I_f^q\right|^2\right]} = \sqrt{\left|\mathbb{E}_{\mathcal{S}}[\hat{H}_T^q(\mathcal{S})] - I_f^q\right|^2 + \mathrm{Var}\left(\hat{H}_T^q(\mathcal{S})\right)} \\
&\lesssim \sqrt{\left[\bar{b} \cdot \mathbb{P}(\Omega_T^-)\right]^2 + \left[T \cdot \bar{b} \cdot r(n)\right]^2 + \left[\bar{b} \cdot \mathbb{P}(\Omega_T^-)\right]^2}
\end{aligned}
$$

Furthermore, if $\mathbb{P}(\Omega_T^-) \le T^{-\alpha}$, by taking $T = \mathcal{O}(r(n)^{-\frac{1}{\alpha+1}})$, we obtain the following result:

$$
\mathbb{E}_{\mathcal{S}}\left[\left|\hat{H}_T^q(\mathcal{S}) - I_f^q\right|\right] \lesssim \bar{b} \cdot r(n)^{\frac{\alpha}{\alpha+1}}.
$$

$\square$

## A.4 PROOF OF THEOREM 4

**Theorem 4** (Convergence Rate of Plug-in Truncated Estimator). *Under the same assumptions in Theorem 3, the convergence rate of the plug-in truncated estimator $\tilde{H}_T^q$ is:*

$$
\mathbb{E}\left[\left|\tilde{H}_T^q(\mathcal{S} \cup \mathcal{S}') - I_f^q\right|\right] \lesssim \mathbb{E}\left[\left|\hat{H}_T^q(\mathcal{S}) - I_f^q\right|\right] + m^{-\frac{1}{2}}.
\tag{10}
$$

*Proof.* Decompose the error of $\tilde{H}_T^q$ via the intermediate estimator $\hat{H}_T^q$:

$$\mathbb{E}_{\mathcal{S} \cup \mathcal{S}'} \left[ \left| \tilde{H}_T^q - I_f^q \right| \right] = \mathbb{E}_{\mathcal{S} \cup \mathcal{S}'} \left[ \left| \tilde{H}_T^q - \hat{H}_T^q + \hat{H}_T^q - I_f^q \right| \right]$$

$$\leq \mathbb{E}_{\mathcal{S} \cup \mathcal{S}'} \left[ \left| \tilde{H}_T^q - \hat{H}_T^q \right| \right] + \mathbb{E}_{\mathcal{S} \cup \mathcal{S}'} \left[ \left| \hat{H}_T^q - I_f^q \right| \right]$$

$$\leq \underbrace{\mathbb{E}_{\mathcal{S} \cup \mathcal{S}'} \left[ \left| \frac{2}{n} \sum_{(\boldsymbol{x}_i, y_i) \in \mathcal{S}_2} (\tau_T(\hat{w}(\boldsymbol{x}_i)) - \tau_T(w(\boldsymbol{x}_i))) \cdot (y_i^q - \hat{f}_{\mathcal{S}_1}^q(\boldsymbol{x}_i)) \right| \right]}_{\text{additional error for estimating } w(\boldsymbol{x})} +$$

$$\underbrace{\mathbb{E}_{\mathcal{S} \cup \mathcal{S}'} \left[ \left| \frac{1}{m} \sum_{i=1}^{m} \hat{f}_{\mathcal{S}_1}^q(\boldsymbol{x}_i') - \mathbb{E}_{\boldsymbol{X} \sim \mathbb{P}^*}[\hat{f}_{\mathcal{S}_1}^q(\boldsymbol{X})] \right| \right]}_{\text{Monte Carlo simulation error}} + \underbrace{\mathbb{E}_{\mathcal{S} \cup \mathcal{S}'} \left[ \left| \hat{H}_T^q - I_f^q \right| \right]}_{\text{error of the case when } w(\boldsymbol{x}) \text{ is known}} \quad .$$

The first term represents the additional error due to estimating the likelihood ratio $w(\boldsymbol{x})$. The second term accounts for the Monte Carlo simulation error in estimating $\mathbb{E}_{\boldsymbol{X} \sim \mathbb{P}^*}[\hat{f}_{\mathcal{S}_1}^q(\boldsymbol{X})]$ using $m$ unlabeled data $\mathcal{S}'$ form target domain, with a convergence rate $\mathcal{O}(m^{-\frac{1}{2}})$. The third term corresponds to the estimation error of $\hat{H}_T^q$ as discussed in Theorem 3.

The convergence rates of the second and third terms have been previously addressed; thus, we only need to focus on the first term. Note that the first term can be upper bounded by:

$$\mathbb{E}_{\mathcal{S} \cup \mathcal{S}'} \left[ \left| \frac{2}{n} \sum_{(\boldsymbol{x}_i, y_i) \in \mathcal{S}_2} (\tau_T(\hat{w}(\boldsymbol{x}_i)) - \tau_T(w(\boldsymbol{x}_i))) \cdot (y_i^q - \hat{f}_{\mathcal{S}_1}^q(\boldsymbol{x}_i)) \right| \right]$$

$$\leq 2T \cdot \mathbb{E}_{\mathcal{S} \cup \mathcal{S}'} \left[ \left| f^q(\boldsymbol{x}) - \hat{f}_{\mathcal{S}_1}^q(\boldsymbol{x}) \right| \right]$$

$$\leq 2T \cdot \sqrt{\mathbb{E}_{\mathcal{S} \cup \mathcal{S}'} \left[ \left( f^q(\boldsymbol{x}) - \hat{f}_{\mathcal{S}_1}^q(\boldsymbol{x}) \right)^2 \right]},$$

which is upper bound by the variance term in Theorem 3. Thus, the convergence rate of $\tilde{H}_T^q$ is:

$$\mathbb{E} \left[ \left| \tilde{H}_T^q(\mathcal{S} \cup \mathcal{S}') - I_f^q \right| \right] \lesssim \mathbb{E} \left[ \left| \hat{H}_T^q(\mathcal{S}) - I_f^q \right| \right] + m^{-\frac{1}{2}}.$$

$$\square$$

## A.5 PROOF OF COROLLARY 1

**Corollary 1** (Double Robustness of Plug-in Truncated Estimator). *If $\mathbb{P}(\Omega_T^-) \to 0$ as $n \to \infty$, then the estimator $\tilde{H}_T^q$ exhibits double robustness. Specifically, this means that as long as either $\hat{w}(\boldsymbol{x})$ or $\hat{f}_{\mathcal{S}_1}(\boldsymbol{x})$ is consistent, the moment estimator $\tilde{H}_T^q$ is also consistent.*

*Proof.* The case when $\hat{f}_{\mathcal{S}_1}(\boldsymbol{x})$ is consistent has been discussed in Theorem 3, so we only need to consider the case when $\hat{w}(\boldsymbol{x})$ is consistent. Decompose the error as follows:

$$\mathbb{E}_{\mathcal{S} \cup \mathcal{S}'} \left[ \left| \tilde{H}_T^q - I_f^q \right| \right] = \mathbb{E} \left[ \left| \frac{1}{m} \sum_{i=1}^{m} \hat{f}_{\mathcal{S}_1}^q(\boldsymbol{x}_i') + \frac{2}{n} \sum_{(\boldsymbol{x}_i, y_i) \in \mathcal{S}_2} \tau_T(\hat{w}(\boldsymbol{x}_i)) \cdot (y_i - \hat{f}_{\mathcal{S}_1}^q(\boldsymbol{x}_i)) - I_f^q \right| \right]$$

$$\leq \mathbb{E}_{\mathcal{S} \cup \mathcal{S}'} \left[ \left| \frac{2}{n} \sum_{(\boldsymbol{x}_i, y_i) \in \mathcal{S}_2} \tau_T(\hat{w}(\boldsymbol{x}_i)) \cdot f^q(\boldsymbol{x}_i) - I_f^q \right| \right] +$$

$$\mathbb{E}_{\mathcal{S} \cup \mathcal{S}'} \left[ \left| \frac{1}{m} \sum_{i=1}^{m} \hat{f}_{\mathcal{S}_1}^q(\boldsymbol{x}_i') - \frac{2}{n} \sum_{(\boldsymbol{x}_i, y_i) \in \mathcal{S}_2} \tau_T(\hat{w}(\boldsymbol{x}_i)) \cdot \hat{f}_{\mathcal{S}_1}^q(\boldsymbol{x}_i) \right| \right] .$$

---

**Algorithm 2** Stabilized Algorithm for Unknown Likelihood Ratios

1: **Input:** labeled data: $\mathcal{S} = \{(\boldsymbol{x}_i, y_i = f(\boldsymbol{x}_i))\}_{i=1}^n$, unlabeled data: $\mathcal{S}' = \{\boldsymbol{x}_i'\}_{i=1}^m$.
2: Use $\mathcal{S}$ and $\mathcal{S}'$ to construct ancillary sample $\mathcal{S}_C = \{(\boldsymbol{x}_i, 0)\}_{i=1}^n \cup \{(\boldsymbol{x}_i', 1)\}_{i=1}^m$.
3: Use ancillary sample $\mathcal{S}_C$ to train a classifier, and then compute the likelihood ratio estimator $\hat{w}(\boldsymbol{x})$ as given in equation 29.
4: Randomly split $\mathcal{S}$ into two parts $\mathcal{S}_1 = \{(\boldsymbol{x}_i, y_i)\}_{i=1}^{\frac{n}{2}}$ and $\mathcal{S}_2 = \{(\boldsymbol{x}_i, y_i)\}_{i=\frac{n}{2}+1}^n$.
5: Train a machine learning model on $\mathcal{S}_1$ to obtain $\hat{f}_{\mathcal{S}_1}(\boldsymbol{x})$.
6: Use $\mathcal{S}'$ and $\mathcal{S}_2$ to estimate the $I_f^q$ based on equation 9.
7: **Output:** $q$-th moment under target: $I_f^q$.

---

If $\hat{w}(\boldsymbol{x})$ is consistent and $\mathbb{P}(\Omega_T^-) \to 0$ as $n \to \infty$, then $\tau_T(\hat{w}(\boldsymbol{x}))$ is a consistent estimator of $w(\boldsymbol{x})$. Let $\to_p$ denote convergence in probability. It holds that:

$$\frac{2}{n} \sum_{(\boldsymbol{x}_i, y_i) \in \mathcal{S}_2} \tau_T(\hat{w}(\boldsymbol{x}_i)) \cdot f^q(\boldsymbol{x}_i) \to_p \mathbb{E}_{\boldsymbol{X} \sim \mathbb{P}^*}[f^q(\boldsymbol{X})],$$

$$\frac{1}{m} \sum_{i=1}^m \hat{f}_{\mathcal{S}_1}^q(\boldsymbol{x}_i') \to_p \mathbb{E}_{\boldsymbol{X} \sim \mathbb{P}^*}[\hat{f}_{\mathcal{S}_1}^q(\boldsymbol{X})],$$

$$\frac{2}{n} \sum_{(\boldsymbol{x}_i, y_i) \in \mathcal{S}_2} \tau_T(\hat{w}(\boldsymbol{x}_i)) \cdot \hat{f}_{\mathcal{S}_1}^q(\boldsymbol{x}_i) \to_p \mathbb{E}_{\boldsymbol{X} \sim \mathbb{P}^*}[\hat{f}_{\mathcal{S}_1}^q(\boldsymbol{X})].$$

Therefore,

$$\tilde{H}_T^q(\mathcal{S} \cup \mathcal{S}') \to_p I_f^q.$$

$\square$

## A.6 STABILIZED ALGORITHM FOR UNKNOWN LIKELIHOOD RATIOS

The stabilized algorithm for unknown likelihood ratios proposed in Section 5 is summarised in Algorithm 1.

## A.7 ESTIMATE OF THE LIKELIHOOD RATIO

The most straightforward approach is to use the unlabeled source and target data to estimate $\hat{p}^\circ$ and $\hat{p}^*$, respectively, and then compute the likelihood ratio $\hat{w} = \frac{\hat{p}^*}{\hat{p}^\circ}$. But the intermediate step of estimating $\hat{p}^\circ$ and $\hat{p}^*$ may introduce additional errors and affect the robustness of the estimation process. Therefore, it is crucial to develop methods that allow us to estimate $w(\boldsymbol{x})$ directly from the data without relying on these intermediate density estimators. To achieve this, we introduce a latent label $Z \in \{0, 1\}$ to indicate whether a sample is from the target distribution ($Z = 1$) or the source distribution ($Z = 0$). This formulation allows us to express the likelihood ratio as:

$$w(\boldsymbol{X}) = \frac{p^*(\boldsymbol{X})}{p^\circ(\boldsymbol{X})} = \frac{\mathbb{P}(\boldsymbol{X} \mid Z = 1)}{\mathbb{P}(\boldsymbol{X} \mid Z = 0)} = \frac{\mathbb{P}(Z = 1 \mid \boldsymbol{X})}{\mathbb{P}(Z = 0 \mid \boldsymbol{X})} \cdot \frac{\mathbb{P}(Z = 0)}{\mathbb{P}(Z = 1)}, \tag{28}$$

where $e(\boldsymbol{x}) = \mathbb{P}(Z = 1 \mid \boldsymbol{X} = \boldsymbol{x})$ is known as the propensity score. With data $\{(\boldsymbol{x}_i, z_i))\}_{i=1}^n$ and $\{(\boldsymbol{x}_i', z_i))\}_{i=1}^m$, we can estimate $\mathbb{P}(Z = 1)$ and $\mathbb{P}(Z = 0)$ with $\frac{m}{m+n}$ and $\frac{n}{m+n}$, respectively. To estimate the propensity score, we can apply a classification model that distinguishes between source and target samples. The propensity score estimator, denoted as $\hat{w}(\boldsymbol{x})$, represents the estimated probability that a sample with feature $x$ belongs to the target distribution. Using the estimated propensity score, we can derive the plug in estimator of the likelihood ratio as follows:

$$\hat{w}(\boldsymbol{x}) = \frac{\hat{e}(\boldsymbol{x})}{1 - \hat{e}(\boldsymbol{x})} \cdot \frac{n}{m}. \tag{29}$$

# B  APPENDIX

## B.1  DISCUSSION ABOUT THE MOTIVATION

Generally speaking, many high-stakes areas are interested in estimating the $q$-th moment of an unknown function, as higher-order moments are often essential for capturing risk-related characteristics in real-world applications beyond the first-order moment. For example,

- in finance, investors are interested in the shape of the asset's return (which is an unknown function of the factors) distribution, especially its skewness and kurtosis. These higher-order moments help investors assess the risk characteristics of an asset, particularly extreme risks (tail risks) and asymmetric risks. Moreover, covariate shift is commonly observed among factors across different industries.

- in medical fields, we need to monitor the volatility (variance) of certain patient metrics to identify high-risk patients, which requires multiple measurements. However, measuring these indicators for rare diseases (e.g. Alzheimer's disease) is very costly. Fortunately, this metric is a function of other, more easily accessible indicators, so we can estimate this function by collecting data on those indicators. Moreover, covariate shift is commonly observed across different population.

- In classical causal inference (Ding, 2024), The average treatment effect on the treated (ATT) is defined as $\mathbb{E}\left[Y(1) - Y(0) \mid T = 1\right]$, where $Y(1)$ and $Y(0)$ represent the potential outcomes under treatment ($T = 1$) and control ($T = 0$). These potential outcomes are unknown functions of the covariate $\boldsymbol{X}$. The difficulty is to estimate $\mathbb{E}\left[Y(0) \mid T = 1\right]$ which can be written as:

$$\begin{aligned}\mathbb{E}\left[Y(0) \mid T = 1\right] =& \mathbb{E}_{\boldsymbol{X}\mid T=1}\left\{\mathbb{E}_{Y\mid\boldsymbol{X},T=1}\left[Y(0) \mid T = 1\right]\right\} \\ =& \mathbb{E}_{\boldsymbol{X}\mid T=1}\left[m_0(\boldsymbol{X})\right].\end{aligned}$$

  But we only have sample $\left\{\left(y_i^{(0)}, \boldsymbol{x}_i\right)\right\}_{i=1}^{n_0}$, where $\boldsymbol{x}_i \sim \mathbb{P}_{\boldsymbol{X}\mid T=0}$.

Due to the broad applicability of these scenarios, we have unified these questions within the framework of estimating the $q$-th moment of an unknown function under covariate shift.

## B.2  DISCUSSION ABOUT THE ADDITIONAL ASSUMPTION IN THEOREM 3

Insights into source/target distribution pairs that satisfy $g(T) \leq T^{-\alpha}$:

- Indeed, a sufficient condition for $g(T) \leq T^{-\alpha}$ is that the likelihood ratio $w(\boldsymbol{x}) = \frac{p^*(\boldsymbol{x})}{p^\circ(\boldsymbol{x})}$ exhibits a polynomial order in $\boldsymbol{x}$. Therefore, any distribution with a polynomial order p.d.f. satisfies this condition, such as the Beta and Pareto distributions.

- Additionally, we can also consider $p^\circ(\boldsymbol{x})$ with any p.d.f. that has faster rate than a polynomial, such as the truncated Gaussian on $\Omega$, provided that $w(\boldsymbol{x})$ still exhibits a polynomial order (e.g., by taking $p^*$ as a Beta distribution).

- Moreover, the benefits of truncation were also observed in certain distribution pairs that do not satisfy this condition in our experiments, such as when both the source and target distributions are truncated Gaussian. Therefore, this condition is used solely for convenience in deriving the upper bound of the truncated estimator and provides insight into changes in the convergence rate.

The estimator $\hat{H}_T^q$ in Theorem 3 may not be minimax optimal:

- The minimax lower bound $B \cdot \bar{b} \cdot r(n)$ established in Theorem 1 applies to the case where the distributions are known. Therefore, we can only assess whether an estimator is minimax optimal under the assumption that the distributions are known and belong to the same distribution class $\mathcal{V}$. However, the upper bound $\bar{b} \cdot r(n)^{\frac{\alpha}{1+\alpha}}$ in Eq. (8) is derived under this known distribution case but within a more restrictive distribution class, as we impose an additional assumption on $\mathcal{V}$, specifically $g(T) < T^{-\alpha}$. Consequently, the corresponding

minimax lower bound will be less than or equal to the bound in Theorem 1. In fact, the upper bound $\bar{b} \cdot r(n)^{\frac{\alpha}{1+\alpha}}$ in Eq. (8) may outperform the minimax lower bound $B \cdot \bar{b} \cdot r(n)$ when $B$ is large.

- When the distributions are unknown, we do not have the minimax lower bound, and therefore cannot determine whether the estimator is minimax-optimal. However, the minimax lower bound derived in the known case serves as a lower bound, which may be loose, but still provides valuable insight.

- The truncation will bring benefits when $B$ is large, as it reduces variance by introducing some bias. This is especially useful in practice because, when the likelihood ratio $w(\boldsymbol{x})$ is unknown, the estimator $\hat{w}(\boldsymbol{x})$ often involves some excessively large values. Specifically, the upper bound after truncation is $\bar{b} \cdot r(n)^{\frac{\alpha}{1+\alpha}}$ , compared to $B \cdot \bar{b} \cdot r(n)$, which is the upper bound before truncation. Note that, the new bound dose not depend on the $B > 1$, which characterize the intensity of covariate shift, but the rate in $r(n)$ decreases. This is especially beneficial when $B$ is large, as it helps mitigate high variance, thereby improving the stability of the algorithm.

When $\alpha = \infty$, the upper bound 8 matches the bound in Theorem 2 for the case of no covariate shift:

- when $\alpha = \infty$, $\mathbb{P}(\{\boldsymbol{x} : w(\boldsymbol{x}) > T\}) = 0$ for any $T > 1$, it holds that $p^*(\boldsymbol{x}) > p^\circ(\boldsymbol{x})$ almost surely. However, since both are p.d.f.s on the $\Omega$, so $p^*(\boldsymbol{x}) = p^\circ(\boldsymbol{x})$ almost surely. Thus there is no covariate shift, and the convergence rate reduce to $\bar{b}r(n)$, which matches the bound in Theorem 2 when $B = 1$ (i.e., no covariate shift).

### B.3 Construction of the Oracle in Assumption 1

When $\{\boldsymbol{x}_i\}_{i=1}^n$ are independent and uniformly distributed over $\Omega$, the optimal function estimator is constructed using the moving least squares method, as demonstrated by Krieg & Sonnleitner (2024) (when $s > \frac{d}{p}$) and Blanchet et al. (2024) (when $s \in (\frac{2dq-dp}{2pq}, \frac{d}{p})$). Thus, we only need to verify that these results hold when the sample is drawn from a distribution with a p.d.f. that is both upper and lower bounded. We need to introduce some notations first.

**Definition 2** (Covering radius)**.** *For any compact region $R \subset \mathbb{R}^d$, the diameter $R$ is defined as $diam(R) := \sup_{\boldsymbol{x}, \boldsymbol{y} \in R} \|\boldsymbol{x} - \boldsymbol{y}\|$. Moreover, for any two compact regions $R_1, R_2 \subset \mathbb{R}^d$, the distance $dist(R_1, R_2)$ between them is defined as $dist(R_1, R_2) := \|\boldsymbol{c}_{R_1} - \boldsymbol{c}_{R_2}\|_\infty$, where $\|\cdot\|_\infty$ denotes the $l_\infty$ norm in $\mathbb{R}^d$ and $\boldsymbol{c}_{R_1}, \boldsymbol{c}_{R_2}$ are the centroids of $R_1, R_2$ respectively. For any collection of $n$ data points $P := \{\boldsymbol{x}_1, \ldots, \boldsymbol{x}_n\}$, the covering radius of $P$ in $\Omega = [0, 1]^d$ is defined as follows:*

$$\rho(P, \Omega) := \sup_{\boldsymbol{y} \in \Omega} \inf_{1 \leq i \leq n} \|\boldsymbol{y} - \boldsymbol{x}_i\|. \tag{30}$$

The properties of the moving least squares estimator is summarized in the following Lemma.

**Lemma 3** (Properties of the moving least squares estimator (Theorem 4.7 in Reznikov & Saff (2016)))**.** *For any given collection of $n$ points $P = \{\boldsymbol{x}_1, \ldots, \boldsymbol{x}_n\}$ with covering radius $\rho(P, \Omega)$, there exist constants $a_1, a_2$ independent of $n$ and continuous functions $u_{\boldsymbol{x}_i} : \Omega \to \mathbb{R}(1 \leq i \leq n)$, such that*

- $\pi(\boldsymbol{y}) = \sum_{i=1}^n \pi(\boldsymbol{x}_i) u_{\boldsymbol{x}_i}(\boldsymbol{y})$ *for any $\boldsymbol{y} \in \Omega$ and any polynomial $\pi$ with $deg(\pi) \leq s - 1$.*

- $\sum_{i=1}^n |u_{\boldsymbol{x}_i}(\boldsymbol{y})| \leq a_1$ *for any $\boldsymbol{y} \in \Omega$.*

- $u_{\boldsymbol{x}}(\boldsymbol{y}) = 0$ *for any $\boldsymbol{y} \in \Omega$ and $\boldsymbol{x} \in P$ with $\|\boldsymbol{x} - \boldsymbol{y}\| \geq a_2 \rho(P, \Omega)$.*

Based on the function $u_{\boldsymbol{x}_i}(1 \leq i \leq n)$ given in Lemma 3, we define a function estimator $K_n = K_n(\{\boldsymbol{x}_i\}_{i=1}^n, \{f(\boldsymbol{x}_i)\}_{i=1}^n)$ for $f$ as

$$K_n(\{\boldsymbol{x}_i\}_{i=1}^n, \{f(\boldsymbol{x}_i)\}_{i=1}^n) := \sum_{i=1}^n f(\boldsymbol{x}_i) u_{\boldsymbol{x}_i}.$$

This estimator is derived using the moving least squares approximation, originally introduced by Wendland (2001). For further details, see Wendland (2004).

To demonstrate that the function estimator $K_n$ defined above satisfies the upper bound in Assumption 1, we need to control the covering radius $\rho(P, \Omega)$ when $P$ is drawn from a distribution with a p.d.f. that is both lower and upper bounded.

**Lemma 4** (Bound on the covering radius (Theorem 2.1 in Reznikov & Saff (2016))). *Given $P := \{\boldsymbol{x}_1, \ldots, \boldsymbol{x}_n\}$ sampled independently and identically from a distribution with a p.d.f. that is both lower and upper bounded on $\Omega = [0, 1]^d$, we have that there exist constants $c_1, c_2 > 0$ and $\alpha_0 > 0$, which are all independent of $n$, such that the following inequality holds for any $\alpha > \alpha_0$:*

$$\mathbb{P}\left(\rho(P, \Omega) \geq c_1 \left(\frac{\alpha \log n}{n}\right)^{\frac{1}{d}}\right) \lesssim n^{1 - c_2 \alpha}.$$

Building on the Lemma 3 and Lemma 4, we can use the result from Appendix E.3 in Blanchet et al. (2024) to demonstrate that the function estimator $K_n$ satisfies the upper bound specified in Assumption 1.

## C   APPENDIX

### C.1   COMPARISON OF DIFFERENT METHODS

In this subsection, we conduct experiments on both synthetic and real datasets to compare the performance of various methods, highlighting the necessity and advantages of the two-stage approach under covariate shift.

**Synthetic dataset.** In our experiment, the source distribution $\mathbb{P}^\circ$ is defined as a truncated normal distribution on $[0, 1]$ with mean $0.2$ and standard deviation $0.3$, denoted by $\mathtt{tnorm}([0, 1], 0.2, 0.3)$. To vary the intensity of covariate shift, we consider the target distributions $\mathtt{tnorm}([0, 1], \mu, 0.3)$, adjusting the mean parameter $\mu$. We set $f(x) = 1 + x^2 + \frac{1}{5} \sin(16x)$ and select $\mu = 0.2, 0.4, 0.6$, resulting in corresponding $B$ values of $1.00, 3.98$, and $12.08$, respectively.
We set $q = 2$ and train $\hat{f}_{\mathcal{S}_1}$ using linear regression. Each experiment is repeated 100 times. The performance of the following three estimators is compared:

- Monte Carlo estimator ($\mathtt{MC}$): estimate $I_f^q$ using $\frac{1}{|\mathcal{S}|} \sum_{i \in \mathcal{S}} w(\boldsymbol{x}_i) y_i^q$.

- One-stage estimator ($\mathtt{One\text{-}stage}$): train $\hat{f}_{\mathcal{S}}$ using the entire dataset $\mathcal{S}$, and then estimate $I_f^q$ by $\mathbb{E}_{\mathbf{X} \sim \mathbb{P}^*}\left[\hat{f}_{\mathcal{S}}^q(\mathbf{X})\right]$.

- Two-stage estimator ($\mathtt{Two\text{-}stage}$): estimate $I_f^q$ using the truncated estimator from equation 7, with $T$ set to $\frac{3}{4} B$.

The results are presented in Figure 3. It indicates that the Monte Carlo estimator outperforms the two-stage methods in the absence of covariate shift ($B = 1$). However, as $B$ increases, the two-stage estimator demonstrates greater accuracy and improved stability, highlighting its necessity under significant covariate shift.

**Real dataset.** We illustrate the application of the proposed two-stage estimator under covariate shift with an empirical example. Specifically, we use the airfoil dataset $\mathcal{S}$ from the UCI Machine Learning Repository, which comprises $N = 1503$ observations of a response variable $Y$ (scaled sound pressure level of NASA airfoils) and a covariate $\boldsymbol{X}$ with $d = 5$ dimensions: log frequency, angle of attack, chord length, free-stream velocity, and suction side log displacement thickness.
We conducted an experiment over 100 trials, where in each trial, we randomly partitioned the dataset $\mathcal{S}$ into three subsets: $\mathcal{S}_{train}$, $\mathcal{S}_{cls}$, and $\mathcal{S}_{test}$.

- $\mathcal{S}_{train}$, containing 50% of the data, is used to construct an estimator of $I_f^q$.

- $\mathcal{S}_{test}$, containing 30% of the data, is used to create the covariate shift and establish a ground truth for $I_f^q$. The covariate shift is introduced by sampling points from $\mathcal{S}_{test}$ with replacement, using probabilities proportional to $h(\boldsymbol{x}) = \exp\left(\boldsymbol{x}^\top \boldsymbol{\beta}\right)$, where $\boldsymbol{\beta} = (-1, 0, 0, 0, 1)^\top$. We denote the resulting sample as $\mathcal{S}_{shift}$, which is used to compute the ground truth by $\frac{1}{|\mathcal{S}_{shift}|} \sum_{(\boldsymbol{x}_i, y_i) \in \mathcal{S}_{shift}} y_i^q$.

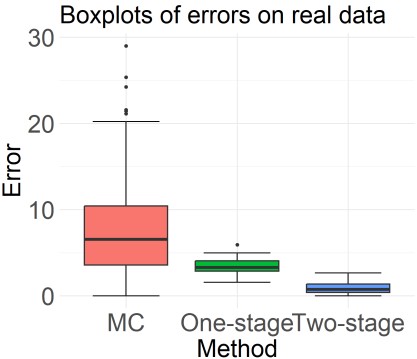

Figure 5: Boxplots of errors for the Monte Carlo estimator, one-stage estimator, and two-stage estimator on the airfoil dataset. The proposed two-stage method outperforms both the Monte Carlo estimator and the one-stage estimator under the covariate shift setting.

- $\mathcal{S}_{cls}$, comprising $20\%$ of the data, is combined with $\mathcal{S}_{shift}$ to train a classifier for estimating the likelihood ratio and obtaining the estimator $\hat{w}(\boldsymbol{x})$.

We compare the three different estimators as follows:

- Monte Carlo estimator (`MC`): estimate $I_f^q$ using $\frac{1}{|\mathcal{S}_{train}|}\sum_{(\boldsymbol{x}_i,y_i)\in\mathcal{S}_{train}}\hat{w}(\boldsymbol{x}_i)y_i^q$.

- One-stage estimator (`One-stage`): train $\hat{f}_{\mathcal{S}_{train}}$ using the entire dataset $\mathcal{S}_{train}$, and then estimate $I_f^q$ by $\frac{1}{|\mathcal{S}_{shift}|}\sum_{(\boldsymbol{x}_i,y_i)\in\mathcal{S}_{shift}}\hat{f}_{\mathcal{S}_{t}rain}^q(\boldsymbol{x}_i)$.

- Two-stage estimator (`Two-stage`): estimate $I_f^q$ using the truncated estimator from equation 7.

The results in Figure 5 demonstrate that the proposed two-stage method outperforms both the Monte Carlo estimator and the one-stage estimator under the covariate shift setting.

