# OpenReview forum: "Minimax Optimal Two-Stage Algorithm For Moment Estimation Under Covariate Shift"
_ICLR.cc/2025/Conference — ICLR 2025 Poster_

### Official Review · Reviewer_AoMc · 2024-10-30

**Soundness:** 2
**Presentation:** 3
**Contribution:** 2
**Rating:** 6
**Confidence:** 3

**Summary:**

This paper proposes a moment estimation method under covariate shift.
More specifically, the authors consider the moment estimation problem with the Sobolev class for the labeling function and with a bounded likelihood ratio of the source and target covariate distribution.

Under this setup, the paper proves a minimax lower bound on the estimation error.

Then, the authors propose a method whose upper bound of the error meets the lower bound, but with the knowledge of the target covariate distribution and the likelihood ratio.
The proposed method combines any estimate of the labeling function and the estimate with truncated likelihood-ratio weighting to construct a doubly robust estimator (Eq. (3)).

Furthermore, the paper proposes an estimator that does not use the knowledge of the target covarite distribution or the likelihood ratio but estimate them using additional unlabeled data sampled from the target distribution. Under some tail probability condition on the likelihood ratio, this estimator has an upper bound slightly worse than the minimax lower bound.

Finally, the paper presents some simulation to see the impact of the degree of covariate shift and the smoothness of the labeling function.

**Strengths:**

- The paper is nicely written. The analyses in the appendix are quite involved, but I can see the authors' efforts to make them accessible.

- The paper studies an interesting and useful problem and theory.

- The proposed doubly robust estimator is interesting (although the core idea has been already wide used in the causal inference/econometrics literature).

**Weaknesses:**

- In the minimax lower bound of Theorem 1, is the estimator $\hat{H}^q$ is not allowed to access any information about $p^*$ or $w$, unlike the proposed estimator does. This makes the comparison between the lower bound and the upper bound in Theorem 2 irrelevant.

  Indeed, the proof does not construct the two hypotheses in a way that the estimator needs to know about $p^*$ or $w$. For example, in the first part of the proof, it suffices to determine whether $p(y | x)$ is $g_0$ or $g_1$.

  I believe that in the formulation of $\mathcal{H}_n^{f,q}$, we can construct two hypothesis with a common $p(y | x)$ but different $p(x)$'s, to make any estimator unable to tell the difference.

- There is a strong assumption that there is no noise in the observations of the label: $y_i = f(x_i)$. This limits applications of the proposed method.

- The estimator in the case of an unknown likelihood ratio does not seem minimax-optimal because of the exponent $\frac{\alpha}{\alpha + 1}$ in Eq. (8).

- There is no experiment about the comparison between the proposed method and the Monte Carlo estimate using $\hat{f}^q(x_i')$ or $w(x_i) y_i^q$.

- Likewise, there is no experiment about the effect of the truncation.

**Questions:**

### Major concerns
- Is there anything I am missing in the first comment in Weaknesses (about the minimax lower bound)?

- Is there a practical application where there is no noise or uncertainty in the label?

- Does the truncation bring any benefits in theory?

- Do the authors have any result for the ablation study about the effect of truncation? That is, comparison between the proposed estimate and that without truncation.

- Any comparison between the proposed method and the Monte Carlo estimate only using $\mathbb{E}[\hat{f}^q(x_i')]$ or $\mathbb{E}[w(x_i) y_i^q]$?

### Minor issues
- Perhaps, the paper should mention that Ma et al. (2023) uses the truncation trick but in the case of unbounded likelihood ratios.

- Is the calculation of the KL divergence with the delta function mathematically sound? Because there is the logarithm function inside the integral. In particular, I could not figure out how to obtain the equality in line 718. Could the authors detail this calculation?

- What is $s$ in $f(x; s)$? Is it $k$?

- Is the $+$ at the end of Eq. (4) a typo?

- In line 644, maybe $K_0$ should be $K$ (otherwise, the support should be $[-1, 1]^d$).


---
**Edit:**
After the discussions with the authors, my concerns about the soundness has been addressed. I have adjusted my score from 5 to 6.
However, I still think the assumptions that there is no noise in $y_i$ and the density is known are strong, and the applications are very limited.

Moreover, the former assumption allows one to change the label from $y_i$ to $y_i^q$, to reduce the problem to the expectation of the mean under covariate shift. This type of problem has been already well studied in the treatment effect estimation literature, including the doubly robust estimators. The submitted paper might lack comparison with such work and discussions about its novelty.

---

> ### Author Response · Authors · 2024-11-19
> **Clarification**
>
> We appreciate the reviewer's supportive and constructive comments, which has helped us improve the quality of our paper. We are pleased that the reviewer found our paper is praised for its theoretical contributions with minimax optimal estimators, practical relevance through stabilized and doubly robust methods, and clear presentation. However, we also acknowledge the concerns raised regarding  (a) certain technical details related to the minimax lower bound, (b) the assumption of noiselessness being too strong, and (c) the absence of some ablation studies.
>
> We respectfully argue that: (a) ***the minimax lower bound is typically expected to be independent of the specific estimator and instead pertains solely to the properties of the distribution class***; (b) there are indeed numerous applications where noise is minimal or negligible, such as in importance sampling for numerical analysis. In light of (c), ***we have added two experiments*** including: a comparison between the truncated and untruncated estimators and a comparison between the Monte Carlo estimate only using $\mathbb{E}[\hat{f}^q(\boldsymbol{x}_i^{\prime})]$ or $\mathbb{E}[w(\boldsymbol{x}_i)y_i^q]$ and the proposed method. Additionally, we have incorporated your constructive suggestions into the main text. We encourage the reviewer to read the General Response first and return here for further clarifications. Below, we have addressed these questions to the best of our ability.
>
> > In the minimax lower bound of Theorem 1, is the estimator $\hat{H}^q$ is not allowed to access any information about $p^{\ast}$ or $w$, unlike the proposed estimator does. This makes the comparison between the lower bound and the upper bound in Theorem 2 irrelevant. Indeed, the proof does not construct the two hypotheses in a way that the estimator needs to know about $p^{\ast}$ or $w$. For example, in the first part of the proof, it suffices to determine whether $p(y\mid x)$ is $g_0$ or $g_1$. I believe that in the formulation of ${H}_n^{f,q}$, we can construct two hypothesis with a common $p(y \mid x)$ but different $p(x)$'s, to make any estimator unable to tell the difference.
> >
> > Is there anything I am missing in the first comment in Weaknesses (about the minimax lower bound)?
>
> We highly appreciate the reviewer's detailed and insightful feedback. However, we respectfully argue that **the lower bound is usually expected to be irrelevant to the estimator**, and instead it relates solely to the properties of distribution class. We refer to [3,4,5] for more related works. The reason is that: in the field of minimax lower bound, our goal is to use lower bound to characterize the difficulty of the estimation problem over a given class and to demonstrate that an estimator can be constructed to achieve this lower bound.
>
> In our proof, the first key step in establishing the minimax lower bound is to transform the challenging **estimation problem**  $\inf_{\hat{H}^q \in H_n^{f,q}} \sup_{(f,p^{\circ}, p^{\ast}) \in V} \mathbb{E}_{S} [|\hat{H}^q({S}) - I_f^q |]$ (i.e., the left-hand side of Eq. 2) into a **testing problem**, a widely adopted approach and nearly the only method in classical textbooks (Chapter 15 in [1], Chapter 2 in [2]). Thus, the primary strategy for deriving minimax lower bounds becomes creating instances from $V$ that are similar enough to be statistically indistinguishable, while ensuring that the corresponding $q$-th moments are as distinct as possible. Consequently, the distributional information must be **encoded in the process of constructing the testing problem**. So we try to pick out the hardest pair of source and target distribution that achieves the upper bound of likelihood ratio $B$.
>
> > There is a strong assumption that there is no noise in the observations of the label: $y_i=f(x_i)$. This limits applications of the proposed method.
> >
> > Is there a practical application where there is no noise or uncertainty in the label?
>
> We thank the reviewer for pointing it out! There are indeed many applications where there is no or less noise/uncertainty that can be ignored, such as numerical analysis. Consider using importance sampling to estimate the $q$-th moment of an unknown function. Specifically, suppose we want to estimate $ \mathbb{E}_{X}[f^q(X)] $ , where $\boldsymbol{X}\sim\mathbb{P}^{\ast}$. The pdf of $\mathbb{P^{\ast}}$ is known but difficult to sample from. We have access to $n$ random quadrature points $(x\_i)\_{i=1}^{n}$ and the corresponding function values $(y\_i)\_{i=1}^{n}$, where each ${x}_i$ is sampled from $\mathbb{P}^{\circ}$ , which is easier to sample from.
>
> We leave the analysis of the noisy case for future work. The main challenge lies in the interleaved terms of different orders between the signal and noise introduced by the expansion of the $q$-th moment, making the problem highly challenging.

---

> > ### Author Response · Authors · 2024-11-19
> > **Clarification (2)**
> >
> > > The estimator in the case of an unknown likelihood ratio does not seem minimax-optimal because of the exponent $\frac{\alpha}{\alpha+1}$ in Eq. (8).
> >
> > We thank the reviewer for the thoughtful and helpful comments. We have added the following discussion in the main text (remark in page 7) which has significantly enhanced the quality of our manuscript.
> >
> > You are correct that the estimator, when the likelihood ratio is unknown, may not be minimax-optimal. Indeed, we need to consider the two situations separately: (i) when the distributions (and thus the likelihood ratio) are known, and (ii) when the distributions are unknown.
> >
> > * The minimax lower bound $B \cdot \bar{b} \cdot r(n)$ established in Theorem 1 applies to the case where the distributions are known.  Therefore, we can only assess whether an estimator is minimax optimal under the assumption that the distributions are known and belong to the same distribution class $\mathcal{V}$. However, the upper bound $\bar{b} \cdot r(n)^{\frac{\alpha}{1+\alpha}}$ in Eq. (8) is derived under this known distribution case but within a more restrictive distribution class, as we impose an additional assumption on $\mathcal{V}$, specifically $g(T) < T^{-\alpha}$. Consequently, the corresponding minimax lower bound will be less than or equal to the bound in Theorem 1. In fact, the upper bound $\bar{b} \cdot r(n)^{\frac{\alpha}{1+\alpha}}$ in Eq. (8) may outperform the minimax lower bound $B \cdot \bar{b} \cdot r(n)$ when $B$ is large.
> > * When the distributions are unknown, we do not have the minimax lower bound, and therefore cannot determine whether the estimator is minimax-optimal. However, the minimax lower bound derived in the known case serves as a lower bound, which may be loose, but still provides valuable insight.
> >
> > > Does the truncation bring any benefits in theory?
> >
> > We sincerely appreciate the reviewer's thoughtful questions and feedback. The truncation will bring benefits when $B$ is large, as it reduces variance by introducing some bias. This is especially useful in practice because, when the likelihood ratio $w(\boldsymbol{x})$ is unknown, the estimator $\hat{w}(\boldsymbol{x})$ often involves some excessively large values. Specifically, the upper bound after truncation is $\bar{b}\cdot r(n)^{\frac{\alpha}{1+\alpha}}$ , compared to $B\cdot\bar{b}\cdot r(n)$, which is the upper bound before truncation.  Note that, the new bound dose not depend on the $B>1$, which characterize the intensity of covariate shift, but the rate in $r(n)$ decreases. This is especially beneficial when $B$ is large, as it helps mitigate high variance, thereby improving the stability of the algorithm.
> >
> > > Likewise, there is no experiment about the effect of the truncation.
> > >
> > > Do the authors have any result for the ablation study about the effect of truncation? That is, comparison between the proposed estimate and that without truncation.
> > >
> > > There is no experiment about the comparison between the proposed method and the Monte Carlo estimate using $\hat{f}^q(\boldsymbol{x}_i^{\prime})$ or $w(\boldsymbol{x}_i)y_i^q$.
> > >
> > > Any comparison between the proposed method and the Monte Carlo estimate only using $\mathbb{E}[\hat{f}^q(\boldsymbol{x}_i^{\prime})]$ or $\mathbb{E}[w(\boldsymbol{x}_i)y_i^q]$?
> >
> > We thank the reviewer for the thoughtful and helpful comments. We have added two experiments including:
> >
> > * A comparison between the truncated and untruncated estimators. It reveals that truncation enhances stability while preserving accuracy, with its effect becoming more significant as the intensity of covariate shift $B$ increases.
> > * A comparison between the Monte Carlo estimate only using $\mathbb{E}[\hat{f}^q(\boldsymbol{x}_i^{\prime})]$ or $\mathbb{E}[w(\boldsymbol{x}_i)y_i^q]$ and the proposed method. The results indicate that the Monte Carlo estimator outperforms the two-stage methods in the absence of covariate shift ($B = 1$). However, as $B$ increases, the two-stage estimator demonstrates greater accuracy and improved stability, highlighting its necessity under significant covariate shift.
> >
> > and summarized the results in the main text (Figure 3,4,5).
> >
> > > Perhaps, the paper should mention that Ma et al. (2023) uses the truncation trick but in the case of unbounded likelihood ratios.
> >
> > We thank the reviewer for the thoughtful and helpful comments. We agree that this would make our paper more comprehensive and have added the following discussion in line 142-146.
> >
> > "Among them, Ma et al. (2023) is closely related to ours which studies the covariate shift problem in the context of nonparametric regression over a reproducing kernel Hilbert space and introduces the truncation trick on the reweighted loss for the case of unbounded likelihood ratios. However, while their focus is on estimating the unknown function itself, our work is concerned with the moments of the unknown function, thereby advancing one step beyond previous studies."

---

> > > ### Author Response · Authors · 2024-11-19
> > > **Clarification (3)**
> > >
> > > > Is the calculation of the KL divergence with the delta function mathematically sound? Because there is the logarithm function inside the integral. In particular, I could not figure out how to obtain the equality in line 718. Could the authors detail this calculation?
> > >
> > > We apologize for the inconvenience. The calculation of the KL divergence with the delta function is mathematically sound, as it relies solely on the property of the Dirac measure: $\int_{-\infty}^{\infty}f(x)\delta_a(x)dx=f(a)$. In fact, the equality in line 718 uses this property.
> > >
> > > Consider now $f(y) = \log(\frac{\frac{1+\epsilon}{2}\prod\_{i:\boldsymbol{x}\_i\in\Omega\_{i^{\ast}}}\delta\_0(y\_i)+\frac{1-\epsilon}{2}\prod\_{i:\boldsymbol{x}\_i\in\Omega\_{i^{\ast}}}\delta\_{g\_1(\boldsymbol{x}\_i)}(y\_i)}{\frac{1-\epsilon}{2}\prod\_{i:\boldsymbol{x}\_i\in\Omega\_{i^{\ast}}}\delta\_0(y\_i)+\frac{1+\epsilon}{2}\prod\_{i:\boldsymbol{x}\_i\in\Omega\_{i^{\ast}}}\delta\_{g\_1(\boldsymbol{x}\_i)}(y\_i)})$ and $\delta\_{a}(y) = \prod\_{i:\boldsymbol{x}\_i\in\Omega\_{i^{\ast}}}\delta\_a(y\_i)$, where $y$ is the vector of $\{y\_i:i \in \{j:\boldsymbol{x}\_j\in\Omega\_{i^{\ast}}\}\}$ . Note that $f(0)=\log(\frac{1+\epsilon}{1-\epsilon})$ and $f(g\_1({\boldsymbol{x})})=\log(\frac{1-\epsilon}{1+\epsilon})$, where $\boldsymbol{x}$ is the vector of $\{\boldsymbol{x}\_i:i \in \{j:\boldsymbol{x}\_j\in\Omega\_{i^{\ast}}\}\}$ .
> > >
> > > Thus, the expression inside the expectation in Line 713 can be denoted as follows and it holds that:
> > >
> > > $\int\_{-\infty}^{\infty}\cdots\int\_{-\infty}^{\infty}f(y)\cdot [\frac{1+\epsilon}{2}\delta\_0(y)+\frac{1-\epsilon}{2}\delta\_{g\_1(\boldsymbol{x})}(y)]dy = \frac{1+\epsilon}{2}f(0) + \frac{1-\epsilon}{2}f(g\_1(\boldsymbol{x})) = \frac{1+\epsilon}{2}\log(\frac{1+\epsilon}{1-\epsilon}) + \frac{1-\epsilon}{2}\log(\frac{1-\epsilon}{1+\epsilon})$.
> > >
> > > The computation here relies solely on the property of the Dirac measure mentioned above. We have added a footnote in the main text (page 15). Thanks!
> > >
> > > > Some typos.
> > >
> > > We sincerely thank the reviewer for their thoroughness. We have addressed the points raised and made the necessary revisions in the main text.
> > >
> > > We appreciate the reviewer’s insightful and constructive comments, which have significantly improved our manuscript. We are eager to provide any further clarifications to facilitate the evaluation process.
> > >
> > > ## Reference
> > >
> > > [1] Wainwright, M. J. (2019). *High-dimensional statistics: A non-asymptotic viewpoint* (Vol. 48). Cambridge university press.
> > >
> > > [2] Alexandre B. Tsybakov. (2009). *Introduction to Nonparametric Estimation*. Springer.
> > >
> > > [3] Kpotufe, S., & Martinet, G. (2018, July). Marginal singularity, and the benefits of labels in covariate-shift. In *Conference On Learning Theory* (pp. 1882-1886). PMLR.
> > >
> > > [4] Pathak, R., Ma, C., & Wainwright, M. (2022, June). A new similarity measure for covariate shift with applications to nonparametric regression. In *International Conference on Machine Learning* (pp. 17517-17530). PMLR.
> > >
> > > [5] Ma, C., Pathak, R., & Wainwright, M. J. (2023). Optimally tackling covariate shift in RKHS-based nonparametric regression. *The Annals of Statistics**, **51*(2), 738-761.

---

> > ### Comment · Reviewer_AoMc · 2024-11-22
> >
> > Thank you for the detailed reply.
> >
> > I still think there is a critical issue with the minimax lower bound. Since the sample $S$ has no information about the target density $p^*$, there is no such estimator $\hat{H}(S)$ that can consistently estimate $I_q(f, p^\circ, p^*) := \mathbb{E}_{X \sim p^*}[f(X)^q]$. This is why I think the class of estimators considered in the mimimax rate is irrelevant.
> >
> > **Proof**:
> > To see this formally, we can use the results from Section 2.2 and Theorem 2.2 (i) of Tsybakov (2009) (after adapting to the problem of the submitted paper):
> > \begin{align}
> > \operatorname*{lim inf}\_{n \to \infty}
> > \inf_{\hat{H}\_n \in \mathcal{H}}
> > \sup_{(f,p^\circ, p^*) \in \mathcal{V}}
> > \mathbb{E}_{S \sim (p^\circ)^n}[\psi_n^{-1} d(\hat{H}_n(S), I^q(f, p^\circ, p^*))]
> > \ge \frac{1-\alpha}{2},
> > \end{align}
> > if there exist two hypotheses $\\{(f_0, p_0^\circ, p_0^*), (f_1, p_1^\circ, p_1^*)\\} \subseteq \mathcal{V}$ such that
> > \begin{align}
> > \operatorname{TV}((p^\circ_1)^n, (p^\circ_0)^n) \le \alpha < 1 \quad \text{(Condition 1)}
> > \end{align}
> > and
> > \begin{align}
> > d(I^q(f_0, p_0^\circ, p_0^*), I^q(f_1, p_1^\circ, p_1^*)) \ge 2 \psi_n.
> > \end{align}
> > Then, we get $\psi_n$ as the rate of the minimax lower bound.
> > Note that we need to compare the two distributions of $S$ in Condition 1.
> >
> > Now, set $q = 1$, $d=1$, $f\_0(x) = f\_1(x) = x$, $p\_0^\circ(x) = p\_1^\circ(x) = p\_0^*(x) = 1/2 \times 1[x \in [-1, 1]]$, and $p\_1^*(x) = 1[x \in [0, 1]]$. Using $d(a, b) = |a - b|$, we get
> > $$
> > \operatorname{TV}((p^\circ_1)^n, (p^\circ_0)^n) = 0 \quad \text{(no difference in $p^\circ$)}
> > $$
> > and
> > $$
> > d(I^1(f_0, p_0^\circ, p_0^*), I^1(f_1, p_1^\circ, p_1^*)) = |0 - 1/2| = 1/2,
> > $$
> > implying
> > \begin{align}
> > \operatorname*{lim inf}\_{n \to \infty}
> > \inf_{\hat{H}\_n \in \mathcal{H}}
> > \sup_{(f,p^\circ, p^*) \in \nu}
> > \mathbb{E}\_{S \sim (p^\circ)^n}[4 d(\hat{H}\_n(S), I^1(f, p^\circ, p^*))]
> > \ge \frac{1}{2},
> > \end{align}
> > so no estimator is consistent.

---

> > > ### Author Response · Authors · 2024-11-23
> > > **Further Clarifications and Corrections**
> > >
> > > We deeply apologize for the mistake and are truly grateful to the reviewer for pointing out this insightful counterexample.  We recognize that the issue stems from our proof, where $p^{\ast}$ was fixed at the outset, leaving no freedom in choosing the target distribution in the $\sup$ over $\mathcal{V}$. We have revised the statement of Theorem 1 in the main text accordingly:
> > >
> > > *Let $\mathcal{H}\_n^{f,q}$ denote  the class of all the estimators using $\mathcal{S} = (\boldsymbol{x}\_i, y_i=f(\boldsymbol{x\}_i))\_{i=1}^n$ to estimate the $q$-th moment of $f$ under $\mathbb{P}^{\ast}$. **For any given target p.d.f.** $p^{\ast} \in ( p^{\ast}:(\cdot, \cdot, p^{\ast})\in \mathcal{V} )$, when $p\geq2$, $q\leq p \leq 2q$ and $(f, p^{\circ}, p^{\ast}) \in \mathcal{V}(\underline{b}, \bar{b}, B, s, p)$, it holds that*
> > > $$
> > > \inf\_{\hat{H}^q \in \mathcal{H}\_n^{f,q}} \sup\_{(f,p^{\circ}, p^{\ast}) \in \mathcal{V}} \mathbb{E}\_{\mathcal{S}} \left[ \left| \hat{H}^q(\mathcal{S}) - I\_f^q \right| \right] \gtrsim   B \cdot \bar{b} \cdot n^{\max (-q(\frac sd - \frac 1p) -1, -\frac 12 - \frac sd)}.
> > > $$
> > >
> > >
> > > * Specifically, our proof requires fixing the target distribution $p^{\ast}$ beforehand. Thus, when constructing two instances $\theta\_0 = (f\_0,p\_0^{\circ},p\_0^{\ast})$ and  $\theta\_1 = (f\_1,p\_1^{\circ} ,p\_1^{\ast})$ within $\mathcal{V}$, it is necessary to ensure $p\_0^{\ast}=p\_1^{\ast}=p^{\ast}$. If two different target distributions are considered, the functional being estimated will differ, which does not reflect the practical scenario.
> > > * When $p^{\ast}$ is fixed, its information is inherently encoded in the intensity of covariate shift, $B$, during the construction of the corresponding source distribution.
> > >
> > > We once again apologize for our mistake and sincerely thank the reviewer for pointing it out. We remain eager to provide any further clarifications to support the evaluation process.

---

> > > > ### Comment · Reviewer_AoMc · 2024-11-24
> > > >
> > > > I agree with the authors that fixing $p^*$ would be a reasonable formulation if we want to compare it with the method using the full knowledge of $p^*$. I think the statement of the theorem could be clearer by using the restricted subset of $\mathcal{V}$ for the supremum. Something like $\sup_{(f, p^\circ) \in \mathcal{V}\_{p^*}}$ with $\mathcal{V}_{p^*} = \\{(f, p^\circ) | (f, p^\circ, p^*) \in \mathcal{V}\\}$.
> > > >
> > > > On the other hand, I disagree with this claim:
> > > > > If two different target distributions are considered, the functional being estimated will differ, which does not reflect the practical scenario.
> > > >
> > > > It is more practical to consider the class of estimators that can adapt to the change of the functional without directly knowing $p^*$ but with its observations. (And we compare it with an upper bound of such an estimator.)

---

> > > > > ### Author Response · Authors · 2024-11-25
> > > > > **Further Clarifications**
> > > > >
> > > > > We sincerely thank the reviewer once again for the insightful comment. We believe there might be a mismatch regarding the term *practical*, and we deeply apologize for any confusion it may have caused. We have made every effort to address your concern below:
> > > > >
> > > > > * By *practical*, we refer to the situation of fixing the target distribution ($p^{\ast}$, which is ***given and known***) outside the $\sup$, as you agreed. (Ma et al. (2023) used a similar technique in Theorem 2.)
> > > > > * In our understanding, the *practical* you refer to pertains the setting that the target distribution is ***given but unknown*** and needs to be estimated. In this context, we only provide the upper bound in this paper (Theorem 4).
> > > > > * We agree that the setting you mentioned is certainly more practical. However, we currently do not have a way to directly obtain such a minimax lower bound.
> > > > >
> > > > > To summarize, after the revision:
> > > > > * We provide the minimax lower bound (Theorem 1) and the corresponding upper bounds for two estimators (Theorems 2 and 3) for any ***given and known*** target distribution.
> > > > > * For any ***given but unknown*** target distribution, we provide only the upper bound (Theorem 4). The lower bound (Theorem 1) for the known setting remains valid but may be loose due to the unknown target distribution.
> > > > > * However, our work can still be seen as a step forward in this direction, shedding light on its potential and challenges.
> > > > >
> > > > >
> > > > > We sincerely thank the reviewer once again for the constructive feedback. We are more than willing to provide any further clarifications to support the evaluation process.

---

> > > > > > ### Comment · Reviewer_AoMc · 2024-11-28
> > > > > >
> > > > > > I thank the authors for their response to my questions.
> > > > > >
> > > > > > The fix for the minimax lower bound looks fine, and the additional experiments look great.
> > > > > >
> > > > > > However, I'm not convinced by the comment about the Dirac delta function. The delta function appears in the logarithm, so we cannot calculate it "solely" using the property $\int f(x) \delta_a(x) dx = f(a)$. I believe we should consider the probability mass functions to define the KL divergence between those discrete measures. In the current form, it is difficult to verify the calculations are correct.

---

> > > > > > > ### Author Response · Authors · 2024-11-29
> > > > > > > **Clarifications on the Dirac Delta Functions**
> > > > > > >
> > > > > > > We sincerely appreciate the reviewer's timely feedback and insightful comments.
> > > > > > > And we apologize for any confusion the notation *Dirac delta function* may cause.
> > > > > > > Next, we first derive the results without Dirac function, and then provide how previous notations work.
> > > > > > > The following discussions are based on Line 767 in the current version.
> > > > > > >
> > > > > > > ***(1) Derive the KL divergence without Dirac delta function $\delta_a(x)$***
> > > > > > >
> > > > > > > The derivation is based on the following three observations:
> > > > > > >
> > > > > > > * More intuitively, $\mathbb{P}_0({y}\mid {x})$ is the Bernoulli distribution that it takes value $g_0({x})$ with probability $\frac{1+\epsilon}{2}$ and $g_1({x})$ with probability $\frac{1-\epsilon}{2}$. $\mathbb{P}_1({y}\mid {x})$ is the Bernoulli distribution that it takes value $g_0({x})$ with probability $\frac{1-\epsilon}{2}$ and $g_1({x})$ with probability $\frac{1+\epsilon}{2}$.
> > > > > > >
> > > > > > > * Note that the function $g_0(\cdot)$ and $g_1(\cdot)$ is different only on $\Omega_{i^{\ast}}$. So, if ${x} \in (\Omega_{i^{\ast}}^C)^n$, then $\mathbb{P}_0({y}\mid {x}) = \mathbb{P}_1({y}\mid {x})$, which implies that $\log(\mathbb{P}_0({y}\mid {x}) / \mathbb{P}_1({y}\mid {x}))=0$. Denote event $A:= \\{ x: x \in (\Omega\_{i^{\ast}}^C)^n \\} $. Thus, If ${x}\in A$, then $KL(\mathbb{P}_0({y}\mid {x})||\mathbb{P}_1({y}\mid {x}))=0$.
> > > > > > >
> > > > > > > * If ${x}\in A^C$, then $KL(\mathbb{P}_0({y}\mid {x})||\mathbb{P}_1({y}\mid {x})) = \frac{1+\epsilon}{2} \log(\frac{1+\epsilon}{1-\epsilon}) + \frac{1-\epsilon}{2} \log(\frac{1-\epsilon}{1+\epsilon}) = \epsilon \log(\frac{1+\epsilon}{1-\epsilon}) . $
> > > > > > >
> > > > > > > Combining the observations above, we can compute the KL divergence between $\mathbb{P}_0({x},{y})$ and $\mathbb{P}_1({x},{y})$ as follows:
> > > > > > >
> > > > > > > $$
> > > > > > > KL(\mathbb{P}\_0({x},{y}) || \mathbb{P}\_1({x},{y})) = \int\_{{x}} \left[ \int\_{{y}}   \log(\frac{\mathbb{P}({x})\cdot \mathbb{P}\_0({y}\mid {x})}{\mathbb{P}({x})\cdot \mathbb{P}\_1({y}\mid {x})}) \mathbb{P}\_0({y}\mid {x}) d {y} \right] \mathbb{P}({x}) d {x}
> > > > > > > \\ = 0 \cdot \mathbb{P}(A) +  \epsilon \log(\frac{1+\epsilon}{1-\epsilon}) \mathbb{P}(A^C).
> > > > > > > $$
> > > > > > >
> > > > > > >
> > > > > > > ***(2) Derive the KL divergence using Dirac delta function $\delta_a(x)$***
> > > > > > >
> > > > > > > In the main text, we follow the notations in Blanchet et al. (2024) and use Dirac delta function during the analysis.
> > > > > > > Note that when $y = g\_0(x)$, it holds that $\mathbb{P}_0( y \mid x) = (1+\epsilon)/2$, and if $y = g\_1(x)$, it holds that $\mathbb{P}_0( y \mid x) = (1-\epsilon)/2$. Therefore, we represent it as
> > > > > > >
> > > > > > > $$\mathbb{P}_0( y \mid x) = \frac{1+\epsilon}{2} \delta_0(y) +  \frac{1-\epsilon}{2}   \delta\_{g\_1(x)}(y).$$
> > > > > > >
> > > > > > > The above formula represents the pdf of a discrete distribution.
> > > > > > > We note that in this case,  due to the property of the Dirac function that $\delta_0(y) = \prod_i \delta_0(y_i)$ and $g_0(x) = 0$, it holds that
> > > > > > > $$\mathbb{P}_0( y \mid x) = \frac{1+\epsilon}{2} \prod_i \delta_0(y_i) +  \frac{1-\epsilon}{2}  \prod_i \delta\_{g\_1(x_i)}(y_i).$$
> > > > > > > This matches our current version in Line 746.
> > > > > > > To continue the calculation in Line 767, we plug into the terms $\mathbb{P}\_0(y \mid x)$ and $\mathbb{P}\_1(y \mid x)$ using the Dirac delta function.
> > > > > > >
> > > > > > >
> > > > > > >
> > > > > > > For the log operator in Line 746, namely, $$ M(y) = \log(\frac{\frac{1+\epsilon}{2}\prod\_{i:\boldsymbol{x}\_i\in\Omega\_{i^{\ast}}}\delta\_0(y\_i)+\frac{1-\epsilon}{2}\prod\_{i:\boldsymbol{x}\_i\in\Omega\_{i^{\ast}}}\delta\_{g\_1(\boldsymbol{x}\_i)}(y\_i)}{\frac{1-\epsilon}{2}\prod\_{i:\boldsymbol{x}\_i\in\Omega\_{i^{\ast}}}\delta\_0(y\_i)+\frac{1+\epsilon}{2}\prod\_{i:\boldsymbol{x}\_i\in\Omega\_{i^{\ast}}}\delta\_{g\_1(\boldsymbol{x}\_i)}(y\_i)}).  $$
> > > > > > >
> > > > > > > The detailed analysis include: if $y = 0$, it holds that $\prod\_{i:\boldsymbol{x}\_i\in\Omega\_{i^{\ast}}} \delta\_{g\_1(\boldsymbol{x}\_i)}(y\_i) = 0$, and therefore one can derive that
> > > > > > > $$ M(0) = \log(\frac{\frac{1+\epsilon}{2}\prod\_{i:\boldsymbol{x}\_i\in\Omega\_{i^{\ast}}}\delta\_0(y\_i)}{\frac{1-\epsilon}{2}\prod\_{i:\boldsymbol{x}\_i\in\Omega\_{i^{\ast}}}\delta\_0(y\_i)}) = \log(\frac{1+\epsilon}{1-\epsilon}). $$
> > > > > > > Similarly, one can calculate that $$M(g_1(x)) = \log(\frac{1-\epsilon}{1+\epsilon}).$$
> > > > > > >
> > > > > > > To finish the derivation in Line 773, again using the above tricks, Line 770 (inside the expectation and integral) is equal to
> > > > > > > $$  \log(\frac{1+\epsilon}{1-\epsilon}) \frac{1+\epsilon}{2} \prod_i \delta_0(y_i) + \log(\frac{1-\epsilon}{1+\epsilon}) \frac{1-\epsilon}{2} \prod_i \delta\_{g\_1(\boldsymbol{x}\_i)}(y_i). $$
> > > > > > >
> > > > > > > After the integral with the fact that $\int\_{-\infty}^\infty f(x) \delta_a(x) dx = f(a)$, it is derived to be
> > > > > > > $$  \log(\frac{1+\epsilon}{1-\epsilon}) \frac{1+\epsilon}{2} + \log(\frac{1-\epsilon}{1+\epsilon}) \frac{1-\epsilon}{2}. $$
> > > > > > >
> > > > > > >
> > > > > > > We have revised our manuscript accordingly (but seems that we cannot upload the current version now). We sincerely thank the reviewer for the insightful and constructive comments, which have greatly enhanced the quality of our manuscript. We are happy to provide any further clarifications to facilitate the evaluation process.

---

> > > > > > > > ### Comment · Reviewer_AoMc · 2024-12-01
> > > > > > > >
> > > > > > > > I thank the authors for their response. I understand the derivation (1).
> > > > > > > >
> > > > > > > > I have adjusted my score from 6 to 7 and added some comments below the Question section of my review.

---

> > > > > > > > > ### Author Response · Authors · 2024-12-02
> > > > > > > > > **Thank you!**
> > > > > > > > >
> > > > > > > > > We sincerely thank the reviewer for the constructive, insightful, and supportive comments during the discussion! These comments greatly help us enhance the current manuscript.
> > > > > > > > >
> > > > > > > > > In light of the reviewer's concern about the comparison to the results in average treatment effect in causal inference, we argue that this work extends the estimation from the expectation (first order) to the $q$-th moment of an unknown function.
> > > > > > > > > This step, from estimating ***linear functional*** to ***non-linear functional***, is technically non-trivial.
> > > > > > > > >
> > > > > > > > > We sincerely thank the reviewer once again for the time and valuable advice, which have significantly improved the quality of our manuscript. We are eager to provide any further clarifications to assist in the evaluation process.

---

### Official Review · Reviewer_gBmp · 2024-11-03

**Soundness:** 3
**Presentation:** 4
**Contribution:** 4
**Rating:** 8
**Confidence:** 3

**Summary:**

The paper studies the problem of functional estimation, specifically, a moment of some function, with respect to the target distribution under covariate shift. The paper establishes minimax lower bound of this problem under RKHS assumption on the objective f, with additional constraints on source and target distribution. Furthermore, the paper proposes an estimator which attains this lower bound. The paper also introduces a practical version of the proposed estimator, and further establishes convergence rate.

**Strengths:**

The paper is beautifully written with a solid theoretical results. Starting from the optimality, it presents an idealized estimator which attains the optimality, and most importantly, it provides a practically usable estimator and establishes theoretical results for the stabilized estimator. The structure and presentation of the theoretical statements hits perfect balance between technical details and insights for readers to follow. The results are stated in a way how each step of the proposed estimator influences the final convergence rate, which really helps to gain insight on the proposed estimator.

**Weaknesses:**

No major weakness.

**Questions:**

On the decaying rate assumption on g(T) in theorem 3, it seems that this imposes further restrictions on the source/target distributions. In addition to the statements on the weight itself, is there any way to gain further insights on what type of source/target distributions pairs would satisfy this condition?

---

> ### Author Response · Authors · 2024-11-19
> **Clarification**
>
> We appreciate the reviewer's supportive and insightful comments.  We are pleased to see that the reviewer agrees that our work is praised for its theoretical contributions with minimax optimal estimators, practical relevance through stabilized and doubly robust methods, and clear, well-structured presentation. Below, we do our best to address the reviewer's questions adequately.
>
> > On the decaying rate assumption on $g(T)$ in theorem 3, it seems that this imposes further restrictions on the source/target distributions. In addition to the statements on the weight itself, is there any way to gain further insights on what type of source/target distributions pairs would satisfy this condition?
>
> We sincerely appreciate the reviewer's thoughtful questions and feedback. We have added the following discussion in the main text (remark in page 7) which has significantly enhanced the quality of our manuscript.
>
> * Indeed, a sufficient condition for $g(T) \leq T^{-\alpha}$ is that the likelihood ratio $w(\boldsymbol{x}) = \frac{p^{\ast}(\boldsymbol{x})}{p^{\circ}(\boldsymbol{x})}$ exhibits a polynomial order in $\boldsymbol{x}$. Therefore, any distribution with a polynomial order p.d.f. satisfies this condition, such as the Beta and Pareto distributions.
> * Additionally, we can also consider $p^{\circ}(\boldsymbol{x})$ with any p.d.f. that has faster rate than a polynomial, such as the truncated Gaussian on $\Omega$, provided that $w(\boldsymbol{x})$ still exhibits a polynomial order (e.g., by taking $p^{\ast}$ as a Beta distribution).
> * Moreover, the benefits of truncation were also observed in certain distribution pairs that do not satisfy this condition in our experiments, such as when both the source and target distributions are truncated Gaussian. Therefore, this condition is used solely for convenience in deriving the upper bound of the truncated estimator and provides insight into changes in the convergence rate.
>
> We sincerely thank the reviewers for their insightful and constructive comments, which have greatly enhanced the quality of our manuscript. We are happy to provide any further clarifications to facilitate the evaluation process.

---

### Official Review · Reviewer_vWi5 · 2024-11-04

**Soundness:** 3
**Presentation:** 3
**Contribution:** 2
**Rating:** 5
**Confidence:** 3

**Summary:**

This paper considers the problem of estimating the moment of an unknown function under covariate shift. Specifically, the paper aims to estimate the $q$-th moment of $f$ under target distribution $\mathbb{P}^*$ with p.d.f. $p^*(x)$, based on a random sample drawn from source distribution $\mathbb{P}^{\circ}$ with p.d.f. $p^{\circ}(x)$. Here, the unknown function $f$ is assumed to belong to the Sobolev space $\mathcal{W}^{s, p}(\Omega)$ with $\Omega \subset \mathbb{R}^d$, where $s$ indicates the degree of smoothness and $p$ specifies the integrability condition of these derivatives. The paper characterizes the impact of covariate shift on the minimax lower bound when $p^{\circ}$ and $p^*$ are known. It turns out that a constant $B \geq w(x)$, where $w(x):=\frac{p^*(x)}{p^{\circ}(x)}$, plays a central role in the established minimax lower bound. Then, the paper proposes a two-stage algorithm which attains the minimax lower bound up to a logarithmic factor for two cases: (i) $p^{\circ}$ and $p^*$ are known and (ii) $p^{\circ}$ and $p^*$ are unknown. For the latter case, the paper truncates an estimator of $w(x)$ to stablize the algorithm. The proposed method requires two models: one for estimating $f(x)$ and the other for estimating $w(x)$. The proposed estimator is doubly robust in that it will be consistent if at least one of the two estimators is consistent.

**Strengths:**

The paper is clearly laid out. Specifically, it provides the minimax lower bounds and develops an estimator that matches the lower bounds up to the log factor when both target and source distributions are known. Furthermore, a doubly robust estimator is developed when the distributions are unknown.

**Weaknesses:**

Although the paper considers an interesting problem in theme of an important topic, namely covariate shift, there are two major weaknesses.

(1) The problem of estimating the $q$-th moment of an unknown function $f$ is not well motivated. On page 1, it is stated that "This is a common scenario in many fields, such as counterfactual inference in causal inference (Ding, 2024)." However, this is not informative enough; Ding (2024) is a textbook and there is no concrete example by simply citing the textbook.

(2) In addition, the current numerical example is very artificial and has nothing to do with real applications in causal inference or any other substantive field.

**Questions:**

(1) There is a typo in the title: "Minixax" should be "Minimax".

(2) When the target distribution is unknown, it is assume that m (≫n) unlabeled samples from the target distribution. Is it necessary that m is much larger than n? This seems quite restrictive. Also, what condition is exactly needed between m and n? Is it enough to assume that $m/n \rightarrow \infty$?

---

> ### Author Response · Authors · 2024-11-19
> **Clarification**
>
> We sincerely appreciate the reviewer's valueable and insightful comments, which have significantly enhanced the quality of our paper. We are pleased that the Reviewer acknowledged our paper's theoretical contributions with minimax optimal estimators, its practical relevance through stabilized and doubly robust methods, and its clear presentation. In light of the reviewer's concerns regarding (a) our motivation for addressing the problem of estimating the $q$-th moment of an unknown function $f$ under covariate shift, as well as (b) the lack of real-world experiments, we have (a) ***provided several examples illustrating the prevalence of this problem in practical applications*** and (b) ***added an experiment using the airfoil dataset to further demonstrate the advantages of our proposed method***. Additionally, we have incorporated your constructive suggestions into the main text. Below, we do our best to address the reviewer's questions adequately. We encourage the reviewer to read the "General Response" section first and then return here for further discussion.
>
> > The problem of estimating the $q$-th moment of an unknown function $f$ is not well motivated. On page 1, it is stated that "This is a common scenario in many fields, such as counterfactual inference in causal inference (Ding, 2024)." However, this is not informative enough; Ding (2024) is a textbook and there is no concrete example by simply citing the textbook.
>
> We apologize for the inconvenience and thank the reviewer for pointing it out, as this feedback helps strengthen the motivation of our paper. Below, we provide several examples demonstrating that the problem of estimating the $q$-th moment of an unknown function $f$ is prevalent in real-world applications.
>
> Generally speaking, many high-stakes areas are interested in estimating the $q$-th moment of an unknown function, as higher-order moments are often essential for capturing risk-related characteristics in real-world applications beyond the first-order moment. For example,
>
> * in finance,  investors are interested in the shape of the asset's return (which is an unknown function of the factors) distribution, especially its skewness and kurtosis.  These higher-order moments help investors assess the risk characteristics of an asset, particularly extreme risks (tail risks) and asymmetric risks. Moreover, covariate shift is commonly observed among factors across different industries.
> * in medical fields, we need to monitor the volatility (variance) of certain patient metrics to identify high-risk patients, which requires multiple measurements. However, measuring these indicators for rare diseases (e.g. Alzheimer's disease) is very costly. Fortunately, this metric is a function of other, more easily accessible indicators, so we can estimate this function by collecting data on those indicators. Moreover, covariate shift is commonly observed across different population.
>
> Due to the broad applicability of these scenarios, we have unified these questions within the framework of estimating the $q$-th moment of an unknown function under covariate shift.
>
> > In addition, the current numerical example is very artificial and has nothing to do with real applications in causal inference or any other substantive field.
>
> We thank the reviewer for the helpful comments. We have added an experiment using the airfoil dataset from the UCI Machine Learning Repository in the main text (Figure 5). The results highlight the advantages of our proposed two-stage method over both the Monte Carlo estimator and the one-stage estimator under covariate shift.
>
> > When the target distribution is unknown, it is assume that m (≫n) unlabeled samples from the target distribution. Is it necessary that m is much larger than n? This seems quite restrictive. Also, what condition is exactly needed between m and n? Is it enough to assume that $m/n→\infty$?
>
> We apologize for any confusion this may have caused. We have added the following discussion as a footnote in the main text (footnote in page 3) which has significantly enhanced the quality of our manuscript. To clarify, $m \gg n$ is intended as an illustrative example rather than an assumption; the results in this paper are independent of the relationship between $m$ and $n$. We state $m \gg n$ because unlabeled data is generally easier to obtain than labeled data.
>
> > Some typos.
>
> We sincerely thank the reviewer for their thoroughness. We have addressed the points raised and made the necessary revisions in the main text.
>
> We appreciate the reviewer’s insightful and constructive comments, which have significantly improved our manuscript. We are eager to provide any further clarifications to facilitate the evaluation process.

---

> > ### Comment · Reviewer_vWi5 · 2024-11-26
> >
> > I appreciate the authors' responses, which provide a more detailed motivation from finance, medicine, and causal inference, along with experimental results using the airfoil dataset from the UCI Machine Learning Repository. However, these updates seem more like supplementary additions rather than central elements of the paper. For example, the airfoil dataset appears to have limited relevance to the motivations derived from finance, medicine, or causal inference. Considering this limitation alongside the authors' considerable effort, I am raising my score to a 5 (marginally below the acceptance threshold), but no higher.

---

> > > ### Author Response · Authors · 2024-11-27
> > > **Thank You!**
> > >
> > > We sincerely appreciate the reviewer's timely feedback. The motivation of this paper focuses on two key aspects: high-order moments and covariate shift. Both are critical and widely relevant in real-world applications, as demonstrated through examples in finance, medicine, and causal inference. Moreover, estimating the moments of an unknown function becomes more challenging under covariate shift compared to scenarios without covariate shift.
> > >
> > >
> > > This paper seeks to address this challenge. To reiterate, the contributions of our work are as follows:
> > > * **Efficient and robust algorithm**: We propose a two-stage estimator that is both efficient and robust for estimating the moments of an unknown function under covariate shift in practical applications.
> > > * **Theoretical guarantees**: We demonstrate that the two-stage estimator achieves minimax optimality when the target and source distributions are known and establish an upper bound for cases where they are unknown.
> > >
> > > Once again, we sincerely thank the reviewer for their valuable feedback and remain eager to provide any further clarifications to support the evaluation process.

---

### Official Review · Reviewer_PpwD · 2024-11-09

**Soundness:** 3
**Presentation:** 3
**Contribution:** 2
**Rating:** 6
**Confidence:** 3

**Summary:**

This paper addresses the problem of estimating moments of responses under covariate shifts. The authors propose a two-stage algorithm, using a doubly robust structure with weight truncation, that achieves a minimax estimation lower bound.

**Strengths:**

- The investigated problem of estimating moments under covariate shift appears fundamental, and the paper offers an approach that the authors show is minimax optimal, which seems a valuable contribution.

 - The technical results are presented and interpreted clearly.

**Weaknesses:**

**Technical Contributions**: The novelty in attaining and proving the main theorems (Theorems 1 and 2), compared to the existing literature, is unclear. In particular, the proofs appear similar to those in Blanchet et al. (2024), with the main addition being separating and upper bounding the term $w(x)$.

**Limitations of the minimax lower bound results**: If I understand correctly, the minimax lower bound result, as well as the proposed algorithm from the authors that achieve the lower bound, assumes the source and target distributions are known. However, a major difficulty of covariate shifts is to handle the unknown distributions. In this regard, the authors only show an upper rate bound and double robustness of their approach in the latter setting. So, I feel there's a gap between the claimed minimax optimality and the actual efficiency of the proposed approach, and this gap is not just the usual discrepancy between theoretical bounds and practical algorithms (as common in ML theoretical guarantees), but is about whether the claimed theory is conceptually trying to capture the fundamental difficulty of the estimation problem.

The above are my main concerns. Additionally, the following suggestions might be useful:
- It would be helpful to provide more justification for the critical Assumption 1, for instance discussing specific estimators that meet the assumption (i.e., to make the paper more self-contained instead of just referring readers to previous papers).
- Section 4.3 deserves more discussions. For example,
- It would be beneficial if the authors could draw parallels of their assumed condition on the probability concerning likelihood ratio with existing literature, possibly through examples demonstrating the probability of large shift regions and the functions $g(T)\leq T^{-\alpha}$ for classical parametric distribution classes.
- More guidance can be provided on choosing threshold $T$ in practice. For example, how much do we need to know about $\alpha$ to choose $T$ properly.
- More discussions can be provided on the implication of the power decay $\alpha$. In particular, when $\alpha = \infty$, does the convergence rate reduce to $\bar b r(n)$, matching the dependence on $n$ observed in Theorem 2?

Some typos:
- In the right-hand side of eq (4) in Assumption 1, the last “+” should be a subscript.

**Questions:**

1. Do we really need to do two-stage algorithm? For example, can $\hat f$ obtained from importance-weighted regression satisfy the upper bound? (perhaps truncating the weights in the regression?)
2. Assumption 1 requires $f$ to be sufficiently smooth. Can the upper bound still be matched if $f$ is less smooth?
3. There is not much discussion on the difference with recent minimax optimal results, e.g., [Ma et al. 2023], which also addresses estimation under nonparametric space. More detailed comparisons would help highlight the unique contributions of this work.

**Reply to authors' response:** I thank the authors for the clarifications and additional experiments. The explanation on the literature regarding known versus unknown distributions is helpful, and so is the additional experiment that shows the significance of two-stage over one-stage procedures. Regarding the relative novelty, my feeling is that using truncation on likelihood ratio to control bias-variance is, in some sense, obvious as an idea, but working it out rigorously and demonstrating its usage as the authors did requires a lot of careful work, which I appreciate. I raised my score in view of all these.

---

> ### Author Response · Authors · 2024-11-19
> **Clarifications**
>
> We sincerely appreciate the Reviewer's insightful comments, which have greatly improved the quality of our paper. We are pleased that the Reviewer found the theoretical contributions of our minimax optimal estimators. However, we also acknowledge the concerns regarding certain technical details. We respectfully argue that: (a) ***the assumption of a known distribution is widely adopted in the previous literature*** when deriving minimax lower bounds under covariate shift; (b) the minimax lower bound is still a valid (though potentially loose) lower bound for the unknown case; and (c) beyond constructing a minimax optimal estimator for the known case, as done in prior work, we also address the non-trivial challenge of implementability in the unknown case under covariate shift. Additionally, we have incorporated your constructive suggestions into the main text. We encourage the reviewer to read the General Response first and return here for further clarifications. Below, we have addressed these questions to the best of our ability.
>
> > **Technical Contributions**: The novelty in attaining and proving the main theorems (Theorems 1 and 2), compared to the existing literature, is unclear. In particular, the proofs appear similar to those in Blanchet et al. (2024), with the main addition being separating and upper bounding the term $w(x)$.
>
> We thank the reviewer for the thoughtful and helpful comments. Indeed, there are differences in constructing the extreme case for the minimax lower bound and the two-stage estimator, stemming from the covariate shift.
>
> * In the proof of the minimax lower bound, the class $\mathcal{V}$ we consider consists of two components: the function $f$ and the source/target distribution pair $(p^{\circ}, p^{\ast})$. Unlike Blanchet et al. (2024), we additionally account for the pair $(p^{\circ}, p^{\ast})$. Thus, when constructing the extreme case, we focus on the distribution pair that achieve the maximum possible covariate shift $B$.
> * The introduction of the reweighting term $w(\boldsymbol{x})$ in the two-stage estimator is intended to correct for bias under covariate shift. However, incorporating $w(\boldsymbol{x})$ can increase variance, potentially making the estimator unstable in practice. To address this issue, we employ a truncation technique to balance the variance-bias trade-off. This approach is particularly relevant in the context of covariate shift, in contrast to the methods used by Blanchet et al. (2024).
>
> > **Limitations of the minimax lower bound results**: [...] the minimax lower bound result, and the proposed algorithm from the authors that achieve the lower bound, assumes the source and target distributions are known. However, a major difficulty of covariate shifts is to handle the unknown distributions. In this regard, the authors only show an upper rate bound and double robustness of their approach in the latter setting. So, there's a gap between the claimed minimax optimality and the actual efficiency of the proposed approach, and this gap is not just the usual discrepancy between theoretical bounds and practical algorithms, but is about whether the claimed theory is conceptually trying to capture the fundamental difficulty of the estimation problem.
>
> We appreciate the reviewer's feedback and would like to address this concern in the following points:
>
> * In theoretical analysis, the case of known distributions is widely adopted in the covariate shift literature. The known distributions are used to leverage the properties of the class $\mathcal{V}$ (which characterizes the extreme case in the class). For instance, [1] derived the minimax lower bound (Theorem 1) for nonparametric classification under covariate shift, [2] established the minimax lower bound (Theorem 2) for nonparametric regression over the class of Hölder continuous functions under covariate shift, and [3] investigated the covariate shift problem in the context of nonparametric regression over a reproducing kernel Hilbert space (RKHS), establishing the corresponding minimax lower bound (Theorem 2).
> * Although this lower bound may have a gap when the distributions are unknown, it still provides a valid (though potentially loose) lower bound for the unknown case, which provides insights for this problem.
> * We respectfully argue that this paper is not a simple application of the proposed minimax bound, but introduce truncation which is not necessary without the covariate shift setting. Specifically, in addition to constructing a minimax optimal estimator for the known case, as done in [1,2,3], we also consider implementability in the unknown case. Notably, our estimator relies solely on the likelihood ratio, which encapsulates the distributional information. Thus, to ensure practical implementability when the distributions are unknown and to address instability due to estimating the likelihood ratio, we propose a truncated estimator that guarantees double robustness and present its upper bound.

---

> > ### Author Response · Authors · 2024-11-19
> > **Clarification (2)**
> >
> > > It would be helpful to provide more justification for the critical Assumption 1, for instance discussing specific estimators that meet the assumption (i.e., to make the paper more self-contained instead of just referring readers to previous papers).
> >
> > We thank the reviewer for the thoughtful and helpful comments. We have added a specific estimators that meet the assumption in Appendix B.3.
> >
> > > It would be beneficial if the authors could draw parallels of their assumed condition on the probability concerning likelihood ratio with existing literature, possibly through examples demonstrating the probability of large shift regions and the functions $g(T)≤T^{-\alpha}$ for classical parametric distribution classes.
> >
> > We sincerely appreciate the reviewer's thoughtful questions and feedback.
> >
> > * Indeed, a sufficient condition for $g(T) \leq T^{-\alpha}$ is that the likelihood ratio $w(\boldsymbol{x}) = \frac{p^{\ast}(\boldsymbol{x})}{p^{\circ}(\boldsymbol{x})}$ exhibits a polynomial order in $\boldsymbol{x}$. Therefore, any distribution with a polynomial order p.d.f. satisfies this condition, such as the Beta and Pareto distributions.
> > * Additionally, we can also consider $p^{\circ}(\boldsymbol{x})$ with any p.d.f. that has faster rate than a polynomial, such as the truncated Gaussian on $\Omega$, provided that $w(\boldsymbol{x})$ still exhibits a polynomial order (e.g., by taking $p^{\ast}$ as a Beta distribution).
> > * Moreover, the benefits of truncation were also observed in certain distribution pairs that do not satisfy this condition in our experiments, such as when both the source and target distributions are truncated Gaussian. Therefore, this condition is used solely for convenience in deriving the upper bound of the truncated estimator and provides insight into changes in the convergence rate.
> >
> > > More guidance can be provided on choosing threshold $T$ in practice. For example, how much do we need to know about $\alpha$ to choose $T$ properly.
> >
> >  We thank the reviewer for the constructive and helpful comments! We have added the following guidance in the main text (footnote in page 7) which has significantly enhanced the quality of our manuscript.
> >
> > The optimal choice of $T=\mathcal{O}(r(n)^{-\frac{1}{\alpha+1}})$ depends on the parameters that characterize the properties of the class $\mathcal{V}$, it brings insight about the relation between the choice of $T$ and the smoothness of function, intensity of covariate shift. Unfortunately, they are unknown in practice, we can only determine the appropriate truncation point through a grid search method.
> >
> > > More discussions can be provided on the implication of the power decay $\alpha$. In particular, when $\alpha=\infty$, does the convergence rate reduce to $\bar{b}r(n)$, matching the dependence on n observed in Theorem 2?
> >
> > We sincerely appreciate the reviewer’s thoughtful questions and feedback. We have added the following discussion in the main text which has significantly enhanced the quality of our manuscript.
> >
> > You are correct. Indeed, when $\alpha=\infty$, $\mathbb{P}(\{\boldsymbol{x}:w(\boldsymbol{x})>T\})=0$ for any $T>1$, it holds that $p^{\ast}(\boldsymbol{x})>p^{\circ}(\boldsymbol{x})$ almost surely. However, since both are p.d.f.s on the $\Omega$, so $p^{\ast}(\boldsymbol{x})=p^{\circ}(\boldsymbol{x})$ almost surely. Thus there is no covariate shift, and the convergence rate reduce to $\bar{b}r(n)$, which matches the bound in Theorem 2 when $B=1$ (i.e., no covariate shift).
> > > There is not much discussion on the difference with recent minimax optimal results, e.g., [Ma et al. 2023], which also addresses estimation under nonparametric space. More detailed comparisons would help highlight the unique contributions of this work.
> >
> > We thank the reviewer for the thoughtful and helpful comments. We agree that this would make our paper more comprehensive and have added the following discussion in line 142-146.
> >
> > "The work by Ma et al. (2023) is closely related to ours. It studies the covariate shift problem in the context of nonparametric regression over a reproducing kernel Hilbert space and introduces the truncation trick on the reweighted loss for the case of unbounded likelihood ratios. However, while their focus is on estimating the unknown function itself, our work is concerned with the moments of the unknown function, thereby advancing one step beyond previous studies."
> >
> > > Some typos.
> >
> > We sincerely thank the reviewer for their thoroughness. We have addressed the points raised and made the necessary revisions in the main text.
> >
> > We appreciate the reviewer’s insightful and constructive comments, which have significantly improved our manuscript. We are eager to provide any further clarifications to facilitate the evaluation process.

---

> > > ### Author Response · Authors · 2024-11-19
> > > **Clarification (3)**
> > >
> > > > Do we really need to do two-stage algorithm? For example,  can $\hat{f}$ obtained from importance-weighted regression satisfy the upper bound? (perhaps truncating the weights in the regression?)
> > >
> > > We thank the reviewer for the insightful questions. Indeed, one can get an estimator without the two-stage, but this is not efficient. Intuitively, the second stage plays a debiasing role. For any given sample size $n$, if the $\hat{f}$ trained in the first stage has a non-zero mean residual, the second stage will be more efficient.
> > >
> > > We have added an experiment comparing the Monte Carlo estimates using either $\mathbb{E}[\hat{f}^q(\boldsymbol{x}_i^{\prime})]$ or $\mathbb{E}[w(\boldsymbol{x}_i)y_i^q]$ with the proposed two-stage method. The results indicate that the Monte Carlo estimator outperforms the two-stage methods in the absence of covariate shift ($B = 1$). However, as $B$ increases, the two-stage estimator demonstrates greater accuracy and improved stability, highlighting its necessity under significant covariate shift.
> > >
> > > > Assumption 1 requires $f$ to be sufficiently smooth. Can the upper bound still be matched if $f$ is less smooth?
> > >
> > > We sincerely appreciate the reviewer's thoughtful questions and feedback. Unfortunately, the upper bound cannot  be matched given the non-smooth $f$. This is due to the large fluctuations that arise when $f$ is not sufficiently smooth (i.e. $\frac{s}{d}<\frac{1}{p}-\frac{1}{2q}$).  Theoretically, by the Sobolev Embedding Theorem, $\mathcal{W}^{s,p}$ is embedded in $\mathcal{L}^{\frac{dp}{d-sp}}$, and since $\frac{dp}{d-sp}<2q$, the function is not guaranteed to have bounded $L^{2q}$-norm. As a result, estimating the function becomes technically invalid. In fact, the reweighted truncated Monte Carlo estimator is more efficient than the two-stage estimator in non-smooth situations.
> > >
> > > We appreciate the reviewer’s insightful and constructive comments, which have significantly improved our manuscript. We are eager to provide any further clarifications to facilitate the evaluation process.
> > >
> > > ## Reference
> > >
> > > [1] Kpotufe, S., & Martinet, G. (2018, July). Marginal singularity, and the benefits of labels in covariate-shift. In *Conference On Learning Theory* (pp. 1882-1886). PMLR.
> > >
> > > [2] Pathak, R., Ma, C., & Wainwright, M. (2022, June). A new similarity measure for covariate shift with applications to nonparametric regression. In *International Conference on Machine Learning* (pp. 17517-17530). PMLR.
> > >
> > > [3] Ma, C., Pathak, R., & Wainwright, M. J. (2023). Optimally tackling covariate shift in RKHS-based nonparametric regression. *The Annals of Statistics*, 51(2), 738-761.

---

> > > > ### Author Response · Authors · 2024-12-04
> > > > **Thank You!**
> > > >
> > > > We sincerely appreciate the reviewer’s constructive, insightful, and supportive comments! These insightful comments greatly help us enhance the current manuscript.

---

### Author Response · Authors · 2024-11-19
**General Response**

We sincerely thank all the reviewers for their insightful and constructive feedback, which has significantly improved our manuscript. We are happy to find that the reviewers agree that our work is praised for its theoretical contributions with minimax optimal estimators, practical relevance through stabilized and doubly robust methods, and clear, well-structured presentation. However, they also raised concerns regarding: (a) the motivation for studying the $q$-th moment of an unknown function $f$ under covariate shift; (b) the absence of experiments comparing the Monte Carlo estimate with the proposed two-stage estimator and the truncated estimator; (c) certain technical limitations related to our minimax lower bound.

Regarding (a) mentioned by Reviewer vWi5, we thank the reviewer for the constructive comment, which helps better motivate our proposed method. Generally speaking, ***many high-stakes areas are interested in estimating the $q$-th moment of an unknown function***, as higher-order moments are often essential for capturing risk-related characteristics in real-world applications beyond the first-order moment. For example,

* in finance,  investors are interested in the shape of the asset's return (which is an unknown function of the factors) distribution, especially its skewness and kurtosis.  These higher-order moments help investors assess the risk characteristics of an asset, particularly extreme risks (tail risks) and asymmetric risks. Moreover, covariate shift is commonly observed among factors across different industries.
* in medical fields, we need to monitor the volatility (variance) of certain patient metrics to identify high-risk patients, which requires multiple measurements. However, measuring these indicators for rare diseases (e.g. Alzheimer's disease) is very costly. Fortunately, this metric is a function of other, more easily accessible indicators, so we can estimate this function by collecting data on those indicators. Moreover, covariate shift is commonly observed across different population.

Due to the broad applicability of these scenarios, we have unified these questions within the framework of estimating the $q$-th moment of an unknown function under covariate shift.

Regarding (b) mentioned by Reviewer PpwD and AoMc, we thank the reviewers for their valuable comments. We conducted several experiments, including:

* A comparison between the Monte Carlo estimate and the proposed two-stage estimator, which ***highlights the necessity of the two-stage estimator***.
* A comparison between the truncated and untruncated estimators, which ***demonstrates the benefits of truncation when the likelihood ratio exhibits high fluctuations (i.e., when $B$ is large)***,

and summarized the results in the main text (Figure 3,4,5).

---

> ### Author Response · Authors · 2024-11-19
> **General Response (2)**
>
> Regarding (c) raised by Reviewer PpwD: (i) The reviewer challenges the assumption that the source and target distributions are known when deriving the minimax lower bound, noting that this is seldom the case in practice. They argue that this assumption may create a potential gap between the claimed minimax optimality and the practical performance of our proposed approach. (ii) Additionally, the reviewer questions the technical contributions compared to [1]. We appreciate the reviewer's feedback and would like to address these concerns through the following points.
>
> For (i):
>
> * In theoretical analysis, ***the case of known distributions is widely adopted in the covariate shift literature.*** The known distributions are used to leverage the properties of the class $\mathcal{V}$ (which characterizes the extreme case in the class). For instance, [2] derived the minimax lower bound (Theorem 1) for nonparametric classification under covariate shift, [3] established the minimax lower bound (Theorem 2) for nonparametric regression over the class of Hölder continuous functions under covariate shift, and [4] investigated the covariate shift problem in the context of nonparametric regression over a reproducing kernel Hilbert space (RKHS), establishing the corresponding minimax lower bound (Theorem 2).
> * Although this lower bound may have a gap when the distributions are unknown, it still provides a valid (though potentially loose) lower bound for the unknown case, which provides insights for this problem.
> * We respectfully argue that this paper is not a simple application of the proposed minimax bound, but ***introduce truncation which is not necessary without the covariate shift setting***. Specifically, in addition to constructing a minimax optimal estimator for the known case, as done in [2,3,4], we also consider implementability in the unknown case. Notably, our estimator relies solely on the likelihood ratio, which encapsulates the distributional information. Thus, to ensure practical implementability when the distributions are unknown and to address instability due to estimating the likelihood ratio, we propose a truncated estimator that guarantees double robustness and present its upper bound.
>
> For (ii):
>
>  Indeed, there are differences in ***constructing the extreme case for the minimax lower bound and the two-stage estimator***, stemming from the covariate shift.
>
> * In the proof of the minimax lower bound, the class $\mathcal{V}$ we consider consists of two components: the function $f$ and the source/target distribution pair $(p^{\circ}, p^{\ast})$. Unlike [1], we additionally account for the pair $(p^{\circ}, p^{\ast})$. Thus, when constructing the extreme case, we focus on the distribution pair that achieve the maximum possible covariate shift $B$.
> * The introduction of the reweighting term $w(\boldsymbol{x})$ in the two-stage estimator is intended to correct for bias under covariate shift. However, incorporating $w(\boldsymbol{x})$ can increase variance, potentially making the estimator unstable in practice. To address this issue, we employ a truncation technique to balance the variance-bias trade-off. This approach is particularly relevant in the context of covariate shift, in contrast to the methods used by [1].
>
> We sincerely thank the reviewers for their insightful and constructive comments, which have significantly improved our manuscript. We are eager to provide any further clarifications to facilitate the evaluation process.
>
>
>
> ## Reference
>
> [1] Blanchet, J., Chen, H., Lu, Y., & Ying, L. (2024). When can Regression-Adjusted Control Variate Help? Rare Events, Sobolev Embedding and Minimax Optimality. *Advances in Neural Information Processing Systems*, *36*.
>
> [2] Kpotufe, S., & Martinet, G. (2018, July). Marginal singularity, and the benefits of labels in covariate-shift. In *Conference On Learning Theory* (pp. 1882-1886). PMLR.
>
> [3] Pathak, R., Ma, C., & Wainwright, M. (2022, June). A new similarity measure for covariate shift with applications to nonparametric regression. In *International Conference on Machine Learning* (pp. 17517-17530). PMLR.
>
> [4] Ma, C., Pathak, R., & Wainwright, M. J. (2023). Optimally tackling covariate shift in RKHS-based nonparametric regression. *The Annals of Statistics*, *51*(2), 738-761.

---

### Meta-Review · Area_Chair_yTpk · 2024-12-17

**Metareview:**

This paper studies the moment estimation under covariate shift by providing a two-stage procedure and attainable minimax lower bound. As the reviewers generally agreed, the paper is well-written and provide theoretically sound contribution on the interesting covariate shift settings. Despite some unclear correspondence between theory and procedure, additional explanations and insights are provided during response period to further clarify the contribution.

**Additional Comments On Reviewer Discussion:**

The original points regarding the mis-coordinate of Theorem 2 and the proposed estimation procedure, has been well-clarified in the author response period, e.g. from AoMc and PpwD. The updated manuscript also include further clarifications and effect of truncation raised by the reviewers.

---

### Decision · Program_Chairs · 2025-01-22

Accept (Poster)